# The Disparate Benefits of Deep Ensembles

**Kajetan Schweighofer** [1]   **Adrian Arnaiz-Rodriguez** [2]   **Sepp Hochreiter** [1 3]   **Nuria Oliver** [2]

## Abstract

Ensembles of Deep Neural Networks, Deep Ensembles, are widely used as a simple way to boost predictive performance. However, their impact on algorithmic fairness is not well understood yet. Algorithmic fairness examines how a model's performance varies across socially relevant groups defined by protected attributes such as age, gender, or race. In this work, we explore the interplay between the performance gains from Deep Ensembles and fairness. Our analysis reveals that they unevenly favor different groups, a phenomenon that we term the *disparate benefits* effect. We empirically investigate this effect using popular facial analysis and medical imaging datasets with protected group attributes and find that it affects multiple established group fairness metrics, including statistical parity and equal opportunity. Furthermore, we identify that the per-group differences in predictive diversity of ensemble members can explain this effect. Finally, we demonstrate that the classical Hardt post-processing method is particularly effective at mitigating the disparate benefits effect of Deep Ensembles by leveraging their better-calibrated predictive distributions.

## 1. Introduction

Deep Ensembles (Lakshminarayanan et al., 2017) have demonstrated their efficacy as a simple and robust method to improve the performance of individual Deep Neural Networks (DNNs). Their superior performance has made them a popular choice for real-world applications (Bhusal et al., 2021; Dolezal et al., 2022), including high-stakes scenarios where the impact on people's lives of Machine Learning (ML) supported decisions can be profound, such as in healthcare, education, finance or the law. In such applications,

[1]ELLIS Unit, LIT AI Lab, Institute for Machine Learning, JKU Linz, Austria [2]ELLIS Alicante, Alicante, Spain [3]NXAI GmbH, Linz, Austria. Correspondence to: Kajetan Schweighofer <schweighofer@ml.jku.at>.

*Proceedings of the 42nd International Conference on Machine Learning*, Vancouver, Canada. PMLR 267, 2025. Copyright 2025 by the author(s).

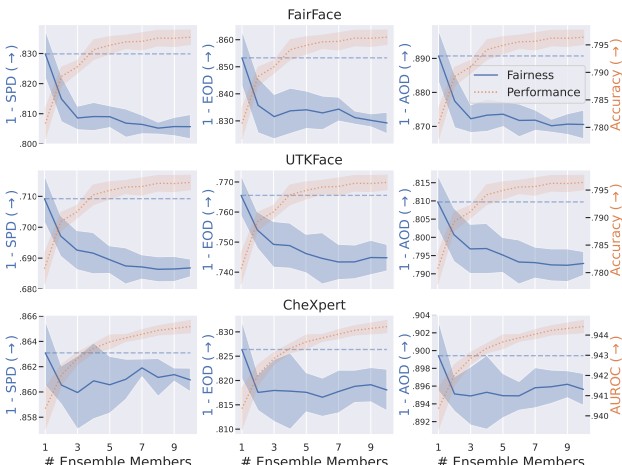

Figure 1: **Unfairness caused by the disparate benefits effect of Deep Ensembles.** Adding members to the ensemble increases its performance (Accuracy, AUROC) but decreases its fairness (1-SPD, 1-EOD, 1-AOD). We find evidence for this effect across multiple standard fairness metrics (cols) and vision datasets (rows).

it is crucial to examine how these models perform across different groups that are defined by a protected attribute (*e.g.,* gender, age, race, etc.) which is the focus of the field of Algorithmic Fairness (Barocas et al., 2023). Ensuring equitable operation of these models across protected groups is imperative, as they can significantly impact individuals and communities, potentially widening existing disparities if not adequately addressed. Although the differences in performance across protected groups (group fairness violations) of individual DNNs has been thoroughly studied (Zhang et al., 2018; Sagawa et al., 2020; Zhang et al., 2022; Arnaiz-Rodriguez & Oliver, 2024), the impact on fairness due to ensembling these networks remains underexplored.

In this paper, our aim is to fill this gap by conducting an extensive empirical study of the fairness implications of Deep Ensembles, analyzing their underlying causes, and exploring mitigation strategies. Our empirical study is based on two popular facial analysis datasets and a widely used medical imaging dataset, each with multiple binary target variables and protected group attributes. We evaluate a total of fifteen tasks across five different DNN model architectures and three standard group fairness metrics. Our analyses reveal that Deep Ensembles unevenly benefit different protected groups in what we refer to as the *disparate benefits* effect (*cf.*

Fig.1). We further investigate the causes of this disparate benefits effect and find evidence that differences in the predictive diversity of ensemble members across groups can explain why ensembling benefits groups differently. Finally, we explore potential approaches to mitigate the negative impact on fairness caused by the disparate benefits effect. We find that Deep Ensembles are more sensitive to the prediction threshold than individual models due to their improved calibration. This makes post-processing methods a suitable approach to mitigate the fairness violations. In fact, our results show that Hardt post-processing (Hardt et al., 2016) is very effective, resulting in fairer predictions while preserving the improved performance of Deep Ensembles. In sum, the main contributions of this paper are threefold:

1. We empirically analyze how the performance gains of Deep Ensembles are distributed across groups defined by protected attributes (Sec. 5). Our findings reveal that Deep Ensembles yield disparate benefits across groups, often benefiting the already advantaged group.

2. We investigate the potential causes for the disparate benefits effect (Sec. 6). Our analysis identifies per-group differences in the predictive diversity of ensemble members as a key contributing factor.

3. We evaluate approaches to mitigate unfairness due to the disparate benefits effect (Sec. 7). We find that Deep Ensembles are more sensitive to the prediction threshold due to their improved calibration. Thus, Hardt post-processing (Hardt et al., 2016) is found to be very effective, ensuring more fair predictions while preserving the improved performance of Deep Ensembles.

## 2. Related Work

**Algorithmic Fairness.** Algorithmic fairness is defined by means of various ethical and legal concepts (Barocas & Selbst, 2016; Corbett-Davies et al., 2017; Binns, 2018), resulting in diverse statistical and causal notions of equality between tasks and contexts (Kusner et al., 2017; Mehrabi et al., 2021). We focus on group fairness metrics —statistical discrimination metrics for classification (Carey & Wu, 2023)— that measure error rate differences between groups defined by protected attributes (Hardt et al., 2016; Zafar et al., 2017). Several metrics quantify group fairness by imposing independence conditions on the joint distribution of targets, predictions, and protected attributes (Barocas et al., 2023), capturing performance disparities due to varying input and target distributions among protected groups (Garg et al., 2020; Pombal et al., 2022). Consequently, a multitude of ML techniques have emerged over the past decade to promote group algorithmic fairness (Mehrabi et al., 2021) by modifying the data (pre-processing) (Kamiran & Calders, 2012; Arnaiz-Rodríguez & Oliver, 2024), the learning process (in-processing) (Agarwal et al., 2018; Jung et al., 2023);

or the model's decision rule (post-processing) (Hardt et al., 2016; Cruz & Hardt, 2024). In this paper, we focus on group algorithmic fairness and analyze the impact of Deep Ensembles on group fairness.

**Deep Ensembles.** Deep Ensembles (Lakshminarayanan et al., 2017) are known as a simple and effective method to boost the performance of DNNs and to estimate predictive uncertainty (Ovadia et al., 2019; Ashukha et al., 2020; Schweighofer et al., 2023). They mainly rely on the stochasticity of the initialization and optimization procedure for diversity (Fort et al., 2019). However, obtaining more diverse Deep Ensembles is still an active area of research (Rame & Cord, 2021; Lee et al., 2023; Pagliardini et al., 2023). Furthermore, the exact mechanisms that produce the performance improvements observed in Deep Ensembles remain an open research question (Abe et al., 2022b; Jeffares et al., 2023; Abe et al., 2024).

**Ensemble Fairness.** Prior work at the intersection of algorithmic fairness and ensembling has investigated the effect of model multiplicity (Marx et al., 2020; Coston et al., 2021; Black et al., 2022a;b; Long et al., 2023; Cooper et al., 2024), and has reported that ensembling decreases the multiplicity of predictions, thus being less arbitrary than individual models. Shallow model ensembles (*i.e.,* models that are not DNNs) have been used to improve the fairness of outcomes (Kamiran & Calders, 2012). Kenfack et al. (2021) considered a fairness-aware weighting of shallow model ensembles. Similarly, Gohar et al. (2023) investigates multiple research questions regarding ensembles of shallow models. Theoretical considerations on ensemble fairness have been investigated by Grgić-Hlača et al. (2017). Based on these insights, Bhaskaruni et al. (2019) proposed a fair ensembling strategy adopting the AdaBoost framework. However, we are not aware of any work that has investigated the impact of Deep Ensembles on group fairness metrics.

The most closely related previous work to ours is that by Ko et al. (2023), which investigates the effect of Deep Ensembles on subgroup performance and served as an inspiration for our work. However, their focus and methodology are different from ours. For most of their experiments, the group variable of interest $A$ is defined as a subset of the full target space $\mathcal{Y}$, *i.e.,* of the worst and best performing targets. In our experiments with real-world data, groups are defined by the values of a protected attribute, such as age, gender, or race. Furthermore, Ko et al. do not consider established group fairness metrics as we do, focusing instead on per-group changes in accuracy. Finally, Ko et al. conclude that Deep Ensembles have exclusively positive impact, while we show that they can negatively affect group fairness. In addition, we investigate potential causes for this effect and analyze mitigation strategies that preserve fairness while maintaining the performance gains of the ensembles.

## 3. Background

We consider the canonical setting of binary classification with inputs $\boldsymbol{x} \in \mathbb{R}^D$, targets $y \in \{0,1\}$, and group attributes $a \in \{0,1\}$ defined according to protected or sensitive variables, such as gender, age, or race. We further consider DNNs as the models to map an input $\boldsymbol{x}$ to the 1-dimensional probability simplex $\Delta^1 = \left\{ (s_0, s_1) \in \mathbb{R}^2 \mid s_0 \geq 0, s_1 \geq 0, s_0 + s_1 = 1 \right\}$. We define this mapping as $f_{\boldsymbol{w}} : \mathbb{R}^D \to \Delta^1$ for a model with parameters $\boldsymbol{w}$. The output of this mapping defines the distribution parameters of the predictive distribution of the model, denoted by $p(y \mid \boldsymbol{x}, \boldsymbol{w})$. A training dataset $\mathcal{D} = \{(\boldsymbol{x}_j, y_j)\}_{j=1}^J$ is used to determine the model parameters by minimizing the cross-entropy loss. The final prediction $\hat{y}$ is given by the argmax over the predictive distribution.

**Deep Ensembles.** Deep Ensembles (Lakshminarayanan et al., 2017) are an ensemble method that uses DNNs as the base learners. While shallow learners often aggregate predictions in the ensemble via majority voting, Deep Ensembles typically average the output distributions of individual members. Furthermore, individual models are generally trained independently on the same data using different random seeds for initialisation and training. Deep Ensembles are widely recognized as a way to perform approximate sampling from the posterior distribution $p(\boldsymbol{w} \mid \mathcal{D}) = p(\mathcal{D} \mid \boldsymbol{w})p(\boldsymbol{w})/p(\mathcal{D})$ (Wilson & Izmailov, 2020; Ashukha et al., 2020), often providing the most faithful posterior approximations (Izmailov et al., 2021). The predictive distribution of an ensemble with $N$ members is given by

$$p(y \mid \boldsymbol{x}, \mathcal{D}) = \int_W p(y \mid \boldsymbol{x}, \boldsymbol{w}) \, p(\boldsymbol{w} \mid \mathcal{D}) \, \mathrm{d}\boldsymbol{w} \qquad (1)$$

$$\approx \frac{1}{N} \sum_{n=1}^N p(y \mid \boldsymbol{x}, \boldsymbol{w}_n), \quad \boldsymbol{w}_n \sim p(\boldsymbol{w} \mid \mathcal{D})$$

Thus, it is an approximation of the posterior predictive distribution. The prediction of the Deep Ensemble equivalent to a single model is given by $\hat{y} = \arg\max p(y \mid \boldsymbol{x}, \mathcal{D})$.

**Group Fairness.** The group fairness desiderata are based on the statistical dependencies between the random variables of the predicted outcomes $\hat{Y}$, the observed outcomes $Y$ and the protected group attribute $A$. Following widespread convention, we consider binary outcomes and protected groups, with $\hat{Y} = Y = 1$ being the positive outcome and $A = 1$ the advantaged group. We focus on three well-established notions of group fairness as follows (Mehrabi et al., 2021; Caton & Haas, 2023).

First, *statistical parity* (Dwork et al., 2012; Kamishima et al., 2012), according to which fairness is achieved when the positive outcome is predicted independently of the protected group attribute. Statistical parity is also known as demographic parity. It is formally defined as

$$P(\hat{Y}=1 \mid A=1) = P(\hat{Y}=1 \mid A=0) \qquad (2)$$

Second, *equal opportunity* (Hardt et al., 2016), which defines fairness as predicting the positive outcome independently of the protected group attribute, but conditioned on the observed outcome being positive. Equal opportunity is therefore formally defined as

$$P(\hat{Y}=1 \mid A=1, Y=1) = P(\hat{Y}=1 \mid A=0, Y=1) \quad (3)$$

Third, *equalized odds* (Hardt et al., 2016), which is a stricter version of equal opportunity where the predictive independence must hold conditioned on both positive and negative observed outcomes. Equalized odds is formally defined as

$$P(\hat{Y}=1 \mid A=1, Y=y) = P(\hat{Y}=1 \mid A=0, Y=y) \quad (4)$$

$\forall y \in \{0,1\}$. These measures are particularly relevant because they operationalize antidiscrimination principles, such as disparate impact in U.S. law (Feldman et al., 2015). Statistical parity focuses on ensuring *equal* outcomes, while equal opportunity and equalized odds balance error rates to promote *equity* across groups. All operationalized notions of fairness have limitations such that it is not necessarily guaranteed that changing the model predictions to satisfy these conditions will actually lead to perfectly fair outcomes in the real world (Selbst et al., 2019; Liu et al., 2018). Furthermore, some notions of fairness can be incompatible with each other, such as statistical parity and equalized odds if $A$ and $Y$ are not independent (Chouldechova, 2017; Kleinberg et al., 2017). Nevertheless, these metrics are a meaningful and widely used tool to quantify group fairness.

## 4. Experimental Setup

**Datasets.** In our experiments, we evaluated Deep Ensembles on three different vision datasets. First, two facial analysis datasets, namely FairFace (Karkkainen & Joo, 2021) and UTKFace (Zhang et al., 2017). For these datasets, all models were trained on the training split of FairFace and evaluated on the official test split of FairFace and the full UTKFace dataset. Protected group attributes were binarized, except for `gender` which was already binary. For the attribute `age`, we defined young and old, where a person is considered old from 40 years onwards to obtain a roughly balanced age distribution. For the attribute `race`, we binarized into white vs. non-white. We trained the models using one of the attributes as the target variable and evaluated with the remaining two attributes as protected group variables for all possible combinations of the target and protected group attributes. Second, the CheXpert medical imaging dataset (Irvin et al., 2019) using the recommended targets provided by Jain et al. (2021) and protected group attributes provided by Gichoya et al. (2022). The `no finding` target was

used to train and evaluate the models. Samples without all protected group attributes have been removed. A random subset of 1/8 was split as test dataset. Protected group attributes were binarized as for the facial analysis datasets. Additional details are given in Apx. D.1.

**Models and training.** We used five different DNN architectures, namely ResNet18/34/50 (He et al., 2016), RegNet-Y 800MF (Radosavovic et al., 2020) and EfficientNetV2-S (Tan & Le, 2021) for our evaluation, due to their widespread adoption and competitive performance in vision tasks. The models that were trained on the FairFace training dataset were trained for 100 epochs using SGD with momentum of 0.9 with a batch size of 256 and learning rate of 1e-2. Furthermore, a standard combination of linear (from factor 1 to 0.1) and cosine annealing schedulers was used. The models that were trained on the CheXpert training dataset were trained for 30 epochs given that the training dataset is roughly three times the size of FairFace, resulting in a similar number of gradient steps and a similar learning rate schedule. We independently trained 10 models for 5 architectures on 4 target variables with 5 seeds. Thus, a total of 1,000 individual models were obtained for our evaluation. The results discussed in the main paper correspond to the use of ResNet50 as the model architecture. Additional results for the other model architectures are provided in Apx. F.2 and Apx. F.3.

**Performance Metrics.** We utilized accuracy as the performance metric on the FairFace and UTKFace datasets. In the case of CheXpert, we measured performance using the AUROC as established in previous work on this dataset (Zhang et al., 2022; Zong et al., 2023).

**Group Fairness Metrics.** We measured group fairness using empirical estimators for the fairness desiderata given by Eq. (2) - (4). Statistical Parity Difference (SPD) estimates the violation of the condition given by Eq. (2) and is computed as

$$PR_{A=1} - PR_{A=0} , \tag{5}$$

where $PR_{A=a}$ is the positive rate calculated on the partition of the test dataset $\mathcal{D}' = \{(\boldsymbol{x}_k, y_k, a_k)\}_{k=1}^K$ with the corresponding protected group attribute $a$. Equal Opportunity Difference (EOD) estimates the violation of the condition given by Eq. (3) and it is computed as

$$TPR_{A=1} - TPR_{A=0} , \tag{6}$$

where $TPR_{A=a}$ is the true positive rate, calculated for the respective group partitions of the test dataset. Average Odds Difference (AOD) (Bellamy et al., 2018) is an estimator of a relaxation of equalized odds (*cf.* Eq. (4)) computed as

$$\frac{1}{2}\left|TPR_{A=1} - TPR_{A=0}\right| + \frac{1}{2}\left|FPR_{A=1} - FPR_{A=0}\right| , \tag{7}$$

Table 1: **Disparate Benefits: Change in performance and fairness violations due to ensembling.** Significant differences ($\Delta$) between the Deep Ensemble (*cf.* Tab 3) and the average ensemble member (*cf.* Tab. 4) are highlighted in bold (t-test, five runs, $p < 0.05$). Gray cells denote that fairness violations are $> 0.05$ for both the Deep Ensemble and the average ensemble member.

| $\mathcal{D}'$ | Target / Group | $\Delta$ Accuracy ($\uparrow$) | $\Delta$ SPD ($\downarrow$) | $\Delta$ EOD ($\downarrow$) | $\Delta$ AOD ($\downarrow$) |
|---|---|---|---|---|---|
| FF | age / gender | $.022_{\pm.001}$ | $\mathbf{.022}_{\pm.003}$ | $\mathbf{.017}_{\pm.004}$ | $\mathbf{.017}_{\pm.003}$ |
| FF | age / race | $.022_{\pm.001}$ | $\mathbf{.009}_{\pm.003}$ | $\mathbf{.012}_{\pm.004}$ | $\mathbf{.007}_{\pm.003}$ |
| FF | gender / age | $.014_{\pm.001}$ | $-.001_{\pm.001}$ | $\mathbf{-.007}_{\pm.001}$ | $\mathbf{-.004}_{\pm.002}$ |
| FF | gender / race | $.014_{\pm.001}$ | $-.001_{\pm.001}$ | $.000_{\pm.000}$ | $-.002_{\pm.002}$ |
| FF | race / age | $.015_{\pm.001}$ | $\mathbf{-.004}_{\pm.001}$ | $\mathbf{.005}_{\pm.002}$ | $-.001_{\pm.000}$ |
| FF | race / gender | $.015_{\pm.001}$ | $.000_{\pm.002}$ | $-.008_{\pm.006}$ | $.002_{\pm.004}$ |
| UTK | age / gender | $.015_{\pm.001}$ | $\mathbf{.017}_{\pm.001}$ | $\mathbf{.015}_{\pm.002}$ | $\mathbf{.012}_{\pm.001}$ |
| UTK | age / race | $.015_{\pm.001}$ | $\mathbf{.010}_{\pm.002}$ | $\mathbf{.010}_{\pm.001}$ | $\mathbf{.004}_{\pm.002}$ |
| UTK | gender / age | $.009_{\pm.001}$ | $.001_{\pm.001}$ | $\mathbf{-.006}_{\pm.002}$ | $\mathbf{-.003}_{\pm.001}$ |
| UTK | gender / race | $.009_{\pm.001}$ | $.000_{\pm.001}$ | $.001_{\pm.002}$ | $.001_{\pm.001}$ |
| UTK | race / age | $.021_{\pm.001}$ | $\mathbf{.013}_{\pm.001}$ | $\mathbf{.007}_{\pm.002}$ | $.000_{\pm.001}$ |
| UTK | race / gender | $.021_{\pm.001}$ | $.003_{\pm.002}$ | $-.002_{\pm.003}$ | $-.002_{\pm.002}$ |

| $\mathcal{D}'$ | Group | $\Delta$ AUROC ($\uparrow$) | $\Delta$ SPD ($\downarrow$) | $\Delta$ EOD ($\downarrow$) | $\Delta$ AOD ($\downarrow$) |
|---|---|---|---|---|---|
| CX | age | $.005_{\pm.000}$ | $\mathbf{.001}_{\pm.000}$ | $\mathbf{.008}_{\pm.004}$ | $\mathbf{.003}_{\pm.001}$ |
| CX | gender | $.005_{\pm.000}$ | $.000_{\pm.001}$ | $.001_{\pm.004}$ | $.001_{\pm.002}$ |
| CX | race | $.005_{\pm.000}$ | $\mathbf{-.002}_{\pm.001}$ | $.000_{\pm.003}$ | $-.001_{\pm.002}$ |

where $FPR_{A=a}$ is the false positive rate, calculated for the respective group partitions of the test dataset. Due to our assumption that $A = 1$ is the advantaged group, all measures are consequently $\in [0, 1]$, where 0 is the most fair. More details on Eq. (5) - (7) are given in Apx. A.

## 5. The Disparate Benefits of Deep Ensembles

In this section, we study the disparate benefits effect of Deep Ensembles using the experimental setup described in Sec. 4. First, we investigate the disparate benefits effect on the FairFace (FF) test dataset. Second, we apply the same models trained on FF to the UTKFace (UTK) dataset. UTK contains facial images similar to those of FF but from a different source, representing a realistic setting for facial analysis under slight distribution shifts. Third, we investigate the disparate benefits effect on the CheXpert (CX) medical imaging dataset to assess whether the impact on fairness of Deep Ensembles also occurs in other domains than facial analysis. Our analysis examines two primary facets of the disparate benefits effect: (i) the relationship between the number of ensemble members and the changes in performance and fairness violations (Fig. 1); and (ii) for which targets and protected group attributes a statistically significant disparate benefits effect is observed (Tab. 1).

**Facial analysis (FF).** The top row of Fig. 1 shows results for FF, where models were trained on target age and evaluated under the protected group attribute gender. We find that performance increases while fairness decreases when adding ensemble members. In particular, the largest decrease in fairness occurs when the first member is added to the Deep Ensemble. Tab. 1 presents the change ($\Delta$) in performance

and fairness violations between individual models and a Deep Ensemble of 10 members for all tasks. While performance always increases for the Deep Ensemble (positive $\Delta$), fairness does not necessarily increase after ensembling. We observe a disparate benefits effect with significant changes in the fairness metrics for four out of six target / protected group combinations. This occurs primarily when individual members already exhibit substantial fairness violations (gray cells in Tab. 1). The strongest disparate benefits effect (largest absolute $\Delta$) has a negative impact, thus decreasing fairness, *i.e.* increasing fairness violations. However, there are also cases where the Deep Ensemble is a more fair classifier than individual models (negative $\Delta$).

**Facial analysis under a distribution shift (UTK).** The middle row of Fig. 1 depicts the results on the UTK dataset, with the same target and protected group as for FF. Individual ensemble members exhibit higher fairness violations than for FF, which can be explained by the distribution shift between FF and UTK. However, the magnitude and behavior of the disparate benefits effect when adding ensemble members are similar to those observed with the FF dataset. The results for all target / group combinations are presented in Tab. 1. The results for UTK are generally similar to those reported for the FF dataset. A notable exception is that the difference in SPD with target variable `race` and protected group attribute `age` is of opposite sign and larger for UTK than for FF.

**Medical imaging (CX).** The bottom row of Fig. 1 shows the results on the CX dataset with `age` as protected group attribute. The disparate benefits effect also occurs in this task, but with a smaller magnitude, which is explained by the smaller performance gains of Deep Ensembles on this dataset. Similarly as with the facial dataset, the change in fairness after adding the first ensemble member is the most pronounced. The complete results for all protected groups are presented in Tab. 1. For the protected group `age`, the disparate benefits effect occurs under all fairness metrics. Moreover, there is a significant difference in SPD for the protected group `race`, although individual models do not have substantial SPD and vice versa for EOD.

**Additional results.** We investigate the influence of the model size of the individual ensemble members in Apx. F.2. Our results show that for tasks where the disparate benefits effect occurs, it increases with model size. Furthermore, we also analyze the disparate benefits effect under different model architectures. The results and more details are given in Apx. F.3, finding that the results provided in the main paper are consistent across architectures. Finally, we also show the disparate benefits effect for heterogeneous Deep Ensembles in Apx. F.4. Complementary to our main investigation, we explore the notion of minimax fairness (Martinez et al., 2020) within our experiments in Apx. F.1.

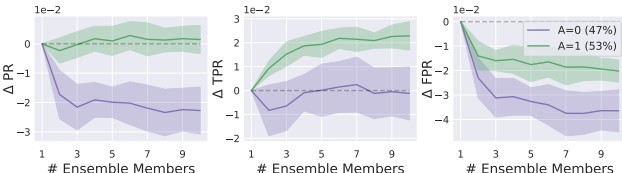

Figure 2: **Change in PR, TPR and FPR when adding members to the ensemble.** Members trained on target variable `age`, evaluated on the FF test dataset with `gender` as protected group attribute. The advantaged group $A = 1$ (male) has higher TPR and lower FPR, resulting in a net zero change in PR. The disadvantaged group $A = 0$ (female) has lower FPR and thus lower PR.

Overall, our results demonstrate that Deep Ensembles can decrease fairness. Therefore, we next investigate the reason for their disparate benefits and explore mitigation strategies.

## 6. What is the Reason for Disparate Benefits?

In this section, we investigate the potential causes of the disparate benefits effect. The considered fairness metrics (Eq. (5) - Eq. (7)) are based on the per-group PR, TPR and FPR metrics. Therefore, as a first step, we analyze how these change when adding additional ensemble members. Although this analysis offers insight into why the disparate benefits effect occurs, it does not directly explain its underlying cause. We hypothesize that the effect arises from differences in predictive diversity among ensemble members. Our empirical results support this hypothesis, indicating that a gap in the average predictive diversity between groups is a key driver of the effect. To validate this further, we designed two synthetic experiments to investigate this hypothesis in a controlled setting.

**Changes to predictions for increasing ensemble size.** We begin by examining how the metrics PR, TPR, and FPR for each group change when ensemble members are added, since the considered fairness metrics (Eq. (5) - Eq. (7)) are based on these. Fig. 2 shows these changes for the model trained on FF with `age` as target variable and `gender` as protected group, evaluated on the FF test dataset. The results show that the increase in SPD results from a decrease in the PR of the disadvantaged group when adding members to the ensemble, while the PR of the advantaged group remains stable ($\Delta \approx 0$). The TPR of the disadvantaged group stays constant, but the TPR of the advantaged group increases, so the Deep Ensemble improves in correctly predicting $Y = 1$ only for the advantaged group, resulting in a higher EOD. The FPR of both groups decreases, more so for the disadvantaged group, thus the Deep Ensemble improves in correctly predicting $Y = 0$ (since FPR is one minus the true negative rate). However, this does not offset the TPR disparity, resulting in higher AOD. The results for the remaining tasks are provided in Fig. 12 - 14 in the appendix.

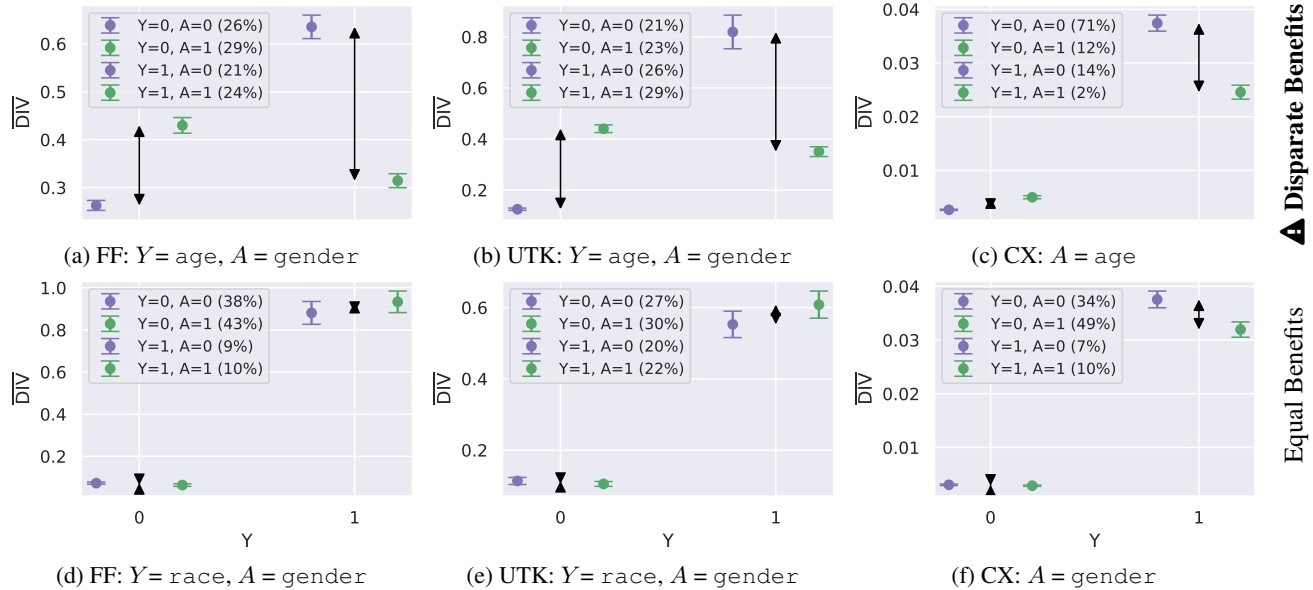

Figure 3: **Average predictive diversity ($\overline{\text{DIV}}$) per group $A$ and target $Y$.** Exemplary results for datasets FF, UTK and CX. Arrows indicate per-target group differences. Top row (a) – (c): Significant disparate benefits (*cf.* Tab. 1) occur when $\overline{\text{DIV}}$ differences between groups are large. Bottom row (d) – (f): No significant disparate benefits (*i.e.* equal benefits) occur when $\overline{\text{DIV}}$ differences are small.

**Predictive diversity of ensemble members.** The ensemble predictive distribution (Eq. (1)) is an average over the predictive distributions of its members. Therefore, the disparate benefits effect must stem from the characteristics of the predictive distributions of individual members. Previous work investigated the predictive diversity of individual members as the driving mechanism for the increase in the performance of Deep Ensembles (Abe et al., 2022b; Jeffares et al., 2023; Abe et al., 2024). Only if individual members have different predictive distributions, combining them can lead to an ensemble that performs better than the individual models. While previous work investigates predictive diversity for individual inputs $\boldsymbol{x}$, we are interested in the average predictive diversity (potentially for a given group) on the test dataset. Following from the definition of predictive diversity by Jeffares et al. (2023) (Theorem 4.3), the average predictive diversity $\overline{\text{DIV}}$ is thus given by

$$\overline{\text{DIV}} = \frac{1}{K}\sum_{k=1}^{K}\underbrace{\log\left(\frac{1}{N}\sum_{n=1}^{N}p(y=y_k \mid \boldsymbol{x}_k, \boldsymbol{w}_n)\right)}_{\text{Ensemble Log-Likelihood}} \quad (8)$$

$$- \underbrace{\frac{1}{N}\sum_{n=1}^{N}\log p(y=y_k \mid \boldsymbol{x}_k, \boldsymbol{w}_n)}_{\text{Average Member Log-Likelihood}},$$

for a test dataset $\mathcal{D}' = \{(\boldsymbol{x}_k, y_k, a_k)\}_{k=1}^{K}$, and a set of $N$ models with parameters $\{\boldsymbol{w}_n\}_{n=1}^{N}$. In Apx. C we provide further discussion about the average predictive diversity and how it arises as a natural measure of interest from a Bayesian perspective. Intuitively, the average predictive diversity $\overline{\text{DIV}}$

is a measure of how different individual ensemble members predict. Thus if there is higher $\overline{\text{DIV}}$ for one group, this group has more potential to benefit from ensembling.

Consequently, we hypothesize that differences in the average predictive diversity per group cause the disparate benefits effect. To investigate this hypothesis, we consider two sets of tasks for FF, UTK and CX, respectively: those where the disparate benefits effect occurs and those where it does not occur (*cf.* Tab. 1). The results are depicted in Fig. 3, showing the average predictive diversity $\overline{\text{DIV}}$ per combination of the target variable $Y$ and the protected group attribute $A$. In agreement with our hypothesis, tasks showing the disparate benefits effect (Fig. 3a-c) have substantial differences in average predictive diversity between groups, while tasks without the effect (Fig. 3d-f) show only minimal differences. Results on all tasks are shown in Fig. 15 - 17 in the appendix.

**Controlled experiment.** To further test our hypothesis of the differences in average predictive diversity between groups causing the disparate benefits effect, we conducted a controlled experiment. We use the FashionMNIST (Xiao et al., 2017) dataset and create a binary classification problem with two targets: "T-shirt/top" ($Y = 0$) vs "Shirt" ($Y = 1$), and two groups, $A = 0$ where the same image of the same target is concatenated twice and $A = 1$ where two different images of the same target are concatenated. This is done for both the train and test datasets. An illustration of inputs $\boldsymbol{x}$ for both targets and groups is given in Fig. 4a. Naturally, having an input consisting of two different images ($A = 1$) should lead to more diverse ensemble members, as they may learn to use the top image, the bottom image

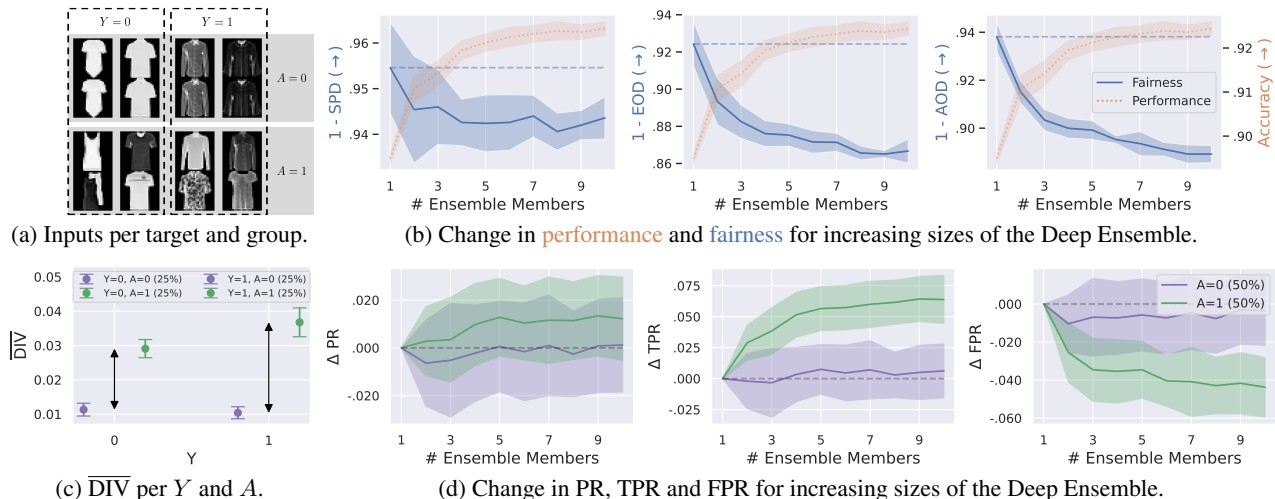

Figure 4: **Controlled experiment.** (a) Overview of the dataset, showing examplary input for different target / group combinations. (b) The performance (accuracy) increases whereas fairness (1-SPD, 1-EOD, 1-AOD) decreases when adding more members to the ensemble. (c, d) The disparate benefits effect is caused by increased PR and TPR, as well as decreased FPR for the group with higher average predictive diversity $A = 1$. For the group with smaller average predictive diversity $A = 0$, there are no significant changes in PR, TPR and FPR.

or any combination of features from both. The concatenation of two identical images ($A = 0$) does not provide additional information and therefore should not lead to an increase in diversity of the ensemble members. This intuition is experimentally confirmed by having a higher $\overline{\text{DIV}}$ for $A = 1$ (Fig. 4c). We observe the same behavior regarding the change in performance, fairness (Fig. 4b) and PR, TPR and FPR (Fig. 4d) as for the real-world datasets that we investigate throughout the rest of the paper. In sum, the synthetic dataset (Fig. 4a) enforces more predictive diversity for one group (Fig. 4c), leading to the disparate benefits effect (Fig. 4b, d).

**Controlling for predictive diversity.** Next, we want to alter the level of predictive diversity and further investigate its relationship with the observed changes in fairness metrics. We used a similar setup as in the previous controlled experiment, *i.e.*, the FashionMNIST dataset and the same targets. However, we used a different methodology to define the groups, in order to vary the level of predictive diversity in the advantaged group $A = 1$. We define an input $x$ for the disadvantaged group $A = 0$ as the original image concatenated with uniform random noise of the same size (each pixel is drawn independently). Furthermore, we define an input for the advantaged group $A = 1$ as the original image concatenated with a linear interpolation between a different image of the same target and uniform random noise. The linear interpolation coefficient is $\alpha$, where $\alpha = 0$ results in solely uniform random noise ($A = 0$ and $A = 1$ are equivalent then) and $\alpha = 1$ results in two concatenated images from the same label. Thus, for $\alpha = 1$, $A = 1$ is equivalent to how it was defined in the previous controlled experiment. An illustration of inputs $x$ for both targets and groups under different values of $\alpha$ is given in Fig. 5.

The results are shown in Fig. 6. In order to summarize the average predictive diversity into a single number, we calculate a diversity score as $|\overline{\text{DIV}}_{Y=1,A=1} - \overline{\text{DIV}}_{Y=1,A=0}| + |\overline{\text{DIV}}_{Y=0,A=1} - \overline{\text{DIV}}_{Y=0,A=0}|$. Intuitively speaking, this is the sum of the lengths of the arrows in Fig. 3, 4c (also in Fig. 15, 16, 17), shown in the rightmost plot in Fig. 6. We observe that for increasing $\alpha$, the diversity score increases. Furthermore, we find that the changes ($\Delta$) in accuracy, SPD, EOD and AOD due to ensembling increase as well, being highly correlated with the average predictive diversity. We provide the absolute accuracies, SPDs, EODs and AODs for individual ensemble members, the Deep Ensemble and the differences between those in Tab. 2 in the Appendix. In sum, for this second controlled experiment, we find that the higher the predictive diversity per group, the stronger the disparate benefits effect.

# 7. Mitigating the Unfairness caused by the Disparate Benefits Effect

In this section, we investigate strategies to mitigate the unfairness due to the disparate benefits effect in the cases where fairness decreases due to ensembling. We focus on interventions that can be applied to trained ensemble members, thus operate in a post-processing manner. This allows to leverage the existing ensemble members as opposed to pre- and in-processing methods that would require expensive re-training of individual members.

First, we analyze whether it would be possible to non-uniformly weight ensemble members to attain a better trade-off between performance and fairness violations in the Deep Ensemble. Second, we examine the characteristics of the predictive distribution of the Deep Ensemble. We find that

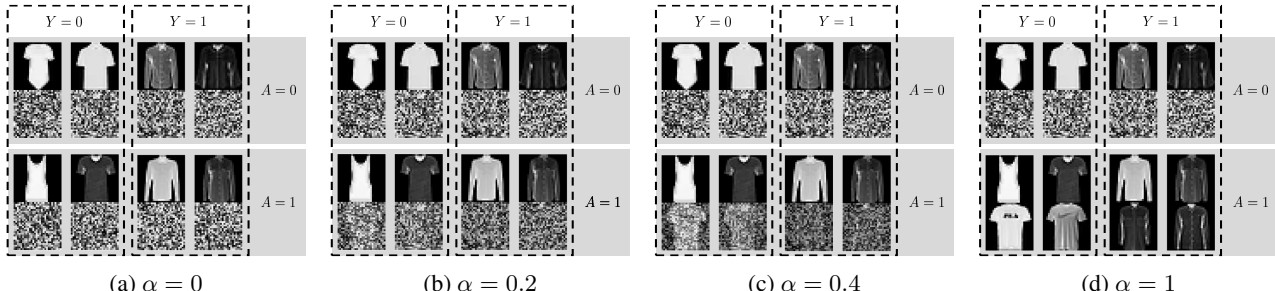

(a) $\alpha = 0$      (b) $\alpha = 0.2$      (c) $\alpha = 0.4$      (d) $\alpha = 1$

Figure 5: **Controlling for predictive diversity.** Inputs per target $Y$ and group $A$ for different levels of linear interpolation factor $\alpha$.

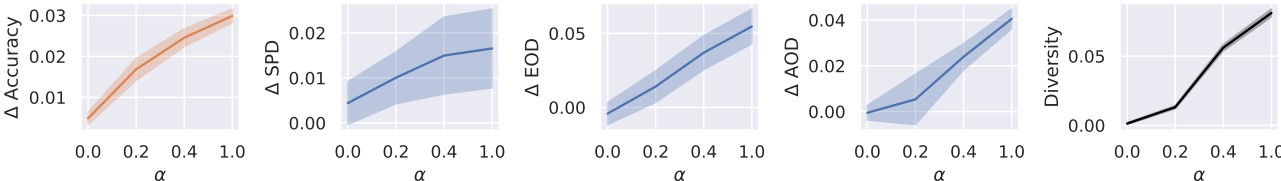

Figure 6: Change ($\Delta$) in accuracy, SPD, EOD and AOD due to ensembling, as well as the diversity score for different levels of linear interpolation factor $\alpha$. The disparate benefits effect is stronger for experimental conditions with higher average predictive diversity.

Deep Ensembles are more calibrated than individual members on our considered tasks and consequently more sensitive to the selected prediction threshold. Inspired by this finding, we investigate a group-dependent threshold optimization approach (Hardt et al., 2016), often simply referred to as Hardt post-processing (HPP) in the algorithmic fairness literature, to mitigate the negative impact of the disparate benefits effect of Deep Ensembles. The results show that HPP is highly effective in ensuring fairer predictions while maintaining the enhanced performance of Deep Ensembles.

**Weighting the ensemble members.** We analyze whether it is possible to improve the performance / fairness trade-off of Deep Ensembles by assigning different weights to each ensemble member, as opposed to the standard uniform weights reflected in Eq. (1). Although the results, shown in Fig. 28 in the appendix, suggest the possibility of better trade-offs, developing a method that systematically identifies the optimal weights to achieve significantly improved outcomes remains challenging. Specifically, we tried two approaches: (i) selecting the best weighting on the validation set and (ii) weighting the individual ensemble members proportional to their fairness violations. Both methods lead to ensembles that are on average in between the performance and fairness violations of the Deep Ensemble with standard uniform weighting and individual models, with high variance. A detailed discussion is provided in Apx. F.5.

**Better calibration leads to more sensitivity to the prediction threshold.** Next, we analyze the predictive distribution of Deep Ensembles to identify mechanisms to mitigate the additional unfairness caused by the disparate benefits effect. Deep Ensembles are known to be better calibrated than individual models because they average over individual predictive distributions (Ovadia et al., 2019; Seligmann et al., 2023). We empirically validate this finding on our considered datasets by evaluating the Expected Calibration Error (ECE) (Naeini et al., 2015). The results are given in Fig. 7a, showing that Deep Ensembles are indeed more calibrated (lower ECE) than individual members for all considered datasets with all available targets $Y$. Being more calibrated means that the predicted probabilities correspond better to the actual outcomes. Importantly, better calibration increases the sensitivity to the prediction threshold, as even slight changes can significantly impact predictions (Cohen & Goldszmidt, 2004). Representative results are shown in Fig. 7b. For Deep Ensembles (Fig. 7b, left), there are clear optimal values for prediction thresholds for each group (dashed lines) that are consistent across runs. For individual members (Fig. 7b, right), there is no clear optimal value. Any threshold between 0.2 and 0.8 leads to similar accuracy and the optimal value has a high variance between runs. The complete results and analysis are provided in Apx. F.6.

**Hardt Post-Processing (HPP).** The sensitivity of Deep Ensembles to the selected threshold suggests that group-specific threshold optimization could be an effective unfairness mitigation strategy. A commonly used approach for this purpose in the algorithmic fairness literature is Hardt post-processing (HPP) (Hardt et al., 2016). As a post-processing method, HPP can be applied to the Deep Ensembles predictive distribution without changing how individual models are trained. Moreover, HPP was shown to be Pareto dominant in addressing equalized odds fairness constraints compared to other fairness interventions (Cruz & Hardt, 2024), while adding minimal computational overhead. Notably, although HPP is a classical method to improve the fairness of ML models, it hasn't been investigated for Deep Ensembles.

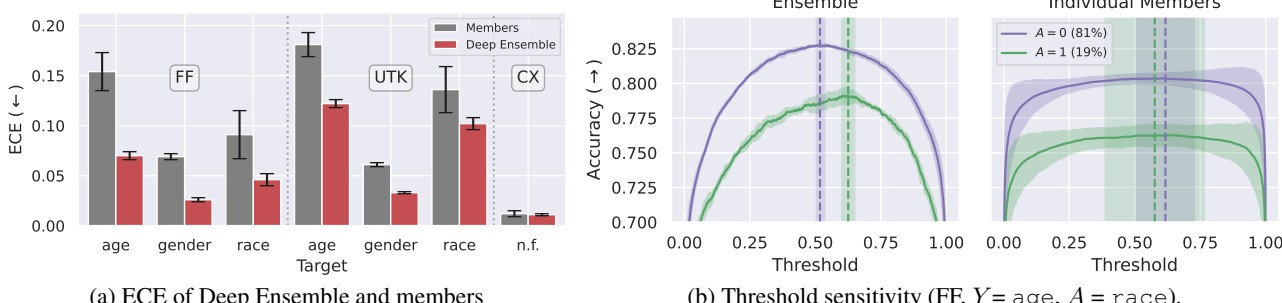

(a) ECE of Deep Ensemble and members      (b) Threshold sensitivity (FF, $Y$ = age, $A$ = race).

Figure 7: **Calibration & prediction threshold.** (a) Deep Ensembles are more calibrated than individual members, thus have lower ECE for all considered tasks. (b) As a result, they are more sensitive to the selection of the prediction threshold.

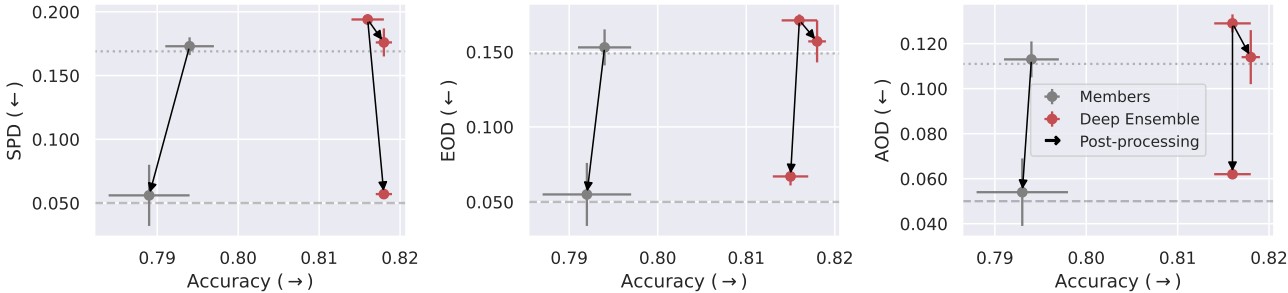

Figure 8: **Impact of applying HPP on the individual members and the Deep Ensemble on FF.** Models are trained on target variable age, evaluated using protected attribute gender. Dotted lines indicate average fairness violation of individual members on the validation set, dashed line indicates 0.05 fairness violation. Arrows denote applying HPP to the Deep Ensemble (red) or the individual members (grey) with those two target fairness violations. The Deep Ensemble maintains or improves in accuracy while also improving in fairness.

We apply HPP to the Deep Ensembles considering each of the three group fairness metrics (SPD, EOD and AOD) with the aim of satisfying the fairness desiderata given by Eq. (2) - Eq. (4). Representative results for the FF dataset with target variable age and protected group attribute gender are depicted in Fig. 8. The complete results for all tasks are given in Tab. 5 - 19 in the appendix. As seen in Fig. 8, after applying HPP, the Deep Ensemble (red dots) attains the same level of fairness violation (y-axis) as individual ensemble members (gray dots) exhibit on average, without sacrificing any performance (x-axis). This is achieved by setting the desired fairness violation for HPP to the average violation of the individual members on a validation set (dotted line). Noteworthy, the Deep Ensemble's accuracy even increases slightly when optimizing the decision thresholds through HPP to values different from 0.5, which is the implicit threshold when using the argmax. Furthermore, we compare the Deep Ensemble and individual ensemble members after applying HPP with a target fairness violation of 0.05 (dashed line). Here, the performance of the Deep Ensemble remains robust, even as the performance of individual members declines.

## 8. Conclusion

In this work, we reveal the disparate benefits effect of Deep Ensembles, analyze its underlying reason, and explore mit-

igation strategies. Specifically, we demonstrated the existence of the effect in experiments on three vision datasets, considering 15 different tasks and five model architectures. We investigated potential causes for this effect, our findings suggesting differences in the predictive diversity of the ensemble members as an explanation. Finally, we evaluated different approaches to mitigate the additional unfairness due to the disparate benefits effect. We find that Deep Ensembles are better calibrated than the individual ensemble members and thus more sensitive to the prediction threshold. Consequently, the classical HPP method is particularly suited to this setting, as it leverages the better-calibrated predictive distribution of the Deep Ensemble. Remarkably, HPP is thus very effective in mitigating unfairness for Deep Ensembles while preserving their superior performance.

The main limitations of our study are that we focus on vision tasks, and hence on ensembles of convolutional DNNs. Furthermore, the three considered group fairness metrics, while widely used, are not sufficient to guarantee fair outcomes, as fairness can't be reduced to satisfying any single metric alone. In future work, we thus plan to explore other notions of fairness, such as individual fairness, and extend our analysis to other types of models and datasets, for instance, in the language domain. Furthermore, we intend to investigate the disparate benefits effect of Deep Ensembles where pre- or in-processing fairness interventions have been applied to individual ensemble members.

## Impact Statement

Our study unveils the disparate benefits effect of Deep Ensembles, which potentially causes socially harmful predictions. Although we investigate its origin and explore a way to mitigate it, this intervention alone can not guarantee fair outcomes. Researchers and practitioners should keep in mind that the fairness of predictions of any ML model applied in the real world can't be reduced to any single metric and must be carefully assessed depending on the application.

## Acknowledgements

We thank Julien Colin, Vihang Patil and Aditya Gulati for their constructive feedback on our work. Furthermore, we thank André Cruz for his help with adapting the `error-parity` package we use for our post-processing experiments.

The ELLIS Unit Linz, the LIT AI Lab, the Institute for Machine Learning, are supported by the Federal State Upper Austria. We thank the projects FWF AIRI FG 9-N (10.55776/FG9), AI4GreenHeatingGrids (FFG- 899943), Stars4Waters (HORIZON-CL6-2021-CLIMATE-01-01), FWF Bilateral Artificial Intelligence (10.55776/COE12). We thank NXAI GmbH, Audi AG, Silicon Austria Labs (SAL), Merck Healthcare KGaA, GLS (Univ. Waterloo), TÜV Holding GmbH, Software Competence Center Hagenberg GmbH, dSPACE GmbH, TRUMPF SE + Co. KG.

The ELLIS Unit Alicante Foundation acknowledges support from Intel corporation, a nominal grant received at the ELLIS Unit Alicante Foundation from the Regional Government of Valencia in Spain (Convenio Singular signed with Generalitat Valenciana, Conselleria de Innovación, Industria, Comercio y Turismo, Dirección General de Innovación) and a grant by the Banc Sabadell Foundation. In addition, this work is partially funded by the European Union EU - HE ELIAS – Grant Agreement 101120237. Views and opinions expressed are however those of the author(s) only and do not necessarily reflect those of the European Union or the European Health and Digital Executive Agency (HaDEA).

Kajetan Schweighofer acknowledges travel support from ELISE (GA no 951847).

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

# A. Details on Computing Group Fairness Metrics

Group fairness metrics, as previously discussed, are based on assumptions related to the independence of the prediction with respect to the protected attribute and the target. For completeness, we present below how to estimate the metrics given in Eq. (5) - Eq. (7) with samples. We start by defining the number of correct (TP, TN) and wrong decissions (FP, FN) of a model:

$$\text{TP} := \sum_{k=1}^{K} \mathbb{1}[f(\boldsymbol{x}_k) > t] \, \mathbb{1}[y_k = 1], \qquad \text{TN} := \sum_{k=1}^{K} \mathbb{1}[f(\boldsymbol{x}_k) < t] \, \mathbb{1}[y_k = 0]$$

$$\text{FP} := \sum_{k=1}^{K} \mathbb{1}[f(\boldsymbol{x}_k) > t] \, \mathbb{1}[y_k = 0], \qquad \text{FN} := \sum_{k=1}^{K} \mathbb{1}[f(\boldsymbol{x}_k) < t] \, \mathbb{1}[y_k = 1].$$

Here, $\mathcal{D}' = \{(\boldsymbol{x}_k, y_k, a_k)\}_{k=1}^{K}$ is the test dataset; a datapoint $(\boldsymbol{x}_k, y_k, a_k)$ consists of input features, observed outcome and protected group attribute; $f(\boldsymbol{x}_k)$ is the model's predicted value for $\boldsymbol{x}_k$; and $t$ is the classification threshold. To compute these metrics for a specific value $a$ of protected group attribute $A$ (*e.g.*, male for `gender`), we add the term $\mathbb{1}[a_k = a]$ to each computation, resulting in group-specific true positives $\text{TP}_{A=a}$, true negatives $\text{TN}_{A=a}$, false positives $\text{FP}_{A=a}$, and false negatives $\text{FN}_{A=a}$.

Once all these buliding blocks are computed, the group-specific Positive Rate ($\text{PR}_{A=a}$) is given by

$$\text{PR}_{A=a} \;=\; P(\hat{Y} = 1 \mid A = a) \;\approx\; \frac{\text{TP}_{A=a} \;+\; \text{FP}_{A=a}}{\text{TP}_{A=a} \;+\; \text{FP}_{A=a} \;+\; \text{TN}_{A=a} \;+\; \text{FN}_{A=a}}\,.$$

Finally, equal opportunity and equalized odds depend on the *conditional* true/false negative/positive rates, depending on the values of the protected group attribute $A$ and are calculated as:

$$\text{TPR}_{A=a} \;=\; P(\hat{Y} = 1 \mid Y = 1, A = a) \;\approx\; \frac{\text{TP}_{A=a}}{\text{TP}_{A=a} \;+\; \text{FN}_{A=a}}$$

$$\text{TNR}_{A=a} \;=\; P(\hat{Y} = 0 \mid Y = 0, A = a) \;\approx\; \frac{\text{TN}_{A=a}}{\text{FP}_{A=a} \;+\; \text{TN}_{A=a}}$$

$$\text{FPR}_{A=a} \;=\; P(\hat{Y} = 1 \mid Y = 0, A = a) \;\approx\; \frac{\text{FP}_{A=a}}{\text{FP}_{A=a} \;+\; \text{TN}_{A=a}}$$

$$\text{FNR}_{A=a} \;=\; P(\hat{Y} = 0 \mid Y = 1, A = a) \;\approx\; \frac{\text{FN}_{A=a}}{\text{TP}_{A=a} \;+\; \text{FN}_{A=a}}$$

## A.1. Group fairness metrics as a factorization of $P(Y, \hat{Y} \mid A)$.

In order to analyze the trade-offs and connections between different statistical group fairness metrics, a common approach is to use the factorization of $P(Y, \hat{Y} \mid A)$, which offers a clear intuition of the incompatibilities between some of them. Then, all the introduced metrics are related as per:

$$
\begin{array}{ccccccc}
P(\hat{Y} \mid Y, A = 1) & \times & P(Y \mid A = 1) & = & P(Y \mid \hat{Y}, A = 1) & \times & P(\hat{Y} \mid A = 1) \\
\| & & \| & & \| & & \| \\
\underbrace{P(\hat{Y} \mid Y, A = 0)}_{\substack{\text{Separation} \\ \hat{Y} \perp A \mid Y \\ \textit{e.g.} \text{ AOD, EOD}}} & \times & \underbrace{P(Y \mid A = 0)}_{\substack{\text{Prevalence Eq.} \\ Y \perp A}} & = & \underbrace{P(Y \mid \hat{Y}, A = 0)}_{\substack{\text{Sufficiency} \\ Y \perp A \mid \hat{Y}}} & \times & \underbrace{P(\hat{Y} \mid A = 0)}_{\substack{\text{Independence} \\ \hat{Y} \perp A \\ \textit{e.g.} \text{ SPD}}}
\end{array}
\tag{9}
$$

For instance, it suggests that, if the target prevalence is different across groups and the model is perfectly calibrated (sufficiency), then separation and independence conditions cannot be satisfied simultaneously.

# B. Biases and Group Unfairness

Biases induced by datasets have been studied in Pombal et al. (2022). They consider the joint distribution $P(X, Y, A)$. Generally there is a bias under a distribution shift with $P^*(X, Y, A) \neq P(X, Y, A)$, where the distribution after the shift $P^*$

the model is applied on is different to the distribution $P$ the training data was sampled from. Furthermore, Pombal et al. (2022) consider biases in the training data distribution. A bias arises if

$$P(X, Y) \neq P(X, Y \mid A) , \tag{10}$$

as well as if $P(A)$ is not a uniform distribution. Note that $P(X, Y \mid A)$ can be factorized into

$$
\begin{aligned}
P(X, Y \mid A) &= P(X \mid Y, A)\, P(X \mid A) \\
&= P(Y \mid X, A)\, P(Y \mid A) .
\end{aligned} \tag{11}
$$

Different parts of the factorization in Eq. (11) can lead to unfairness:

- $P(Y) \neq P(Y \mid A)$ corresponds to a *prevalence disparity*, i.e., the class probability depends on the protected attribute. This imbalance is not present in FairFace dataset since it has been specifically curated to avoid this problem (Karkkainen & Joo, 2021). However, we observe it in the UTKFace and CheXpert datasets.

- $P(X \mid Y) \neq P(X \mid Y, A)$ reflects a *group-wise disparity of the class-conditional distribution*, and indicates that the feature space is distributed differently depending on the protected attribute, which is undesirable, since the likelihood of $p(\mathcal{D} \mid \boldsymbol{w})$ could vary across protected groups, leading to potentially different per-group error rates and hence unfairness. The experimental results in Fig. (3) illustrate differences in the likelihood of the dataset for different $(A, Y)$.

- $P(Y \mid X) \neq P(Y \mid X, A)$ represents *noisy targets*. In this case, the distribution of $Y$ given $X$ depends on the protected group attribute. The classification experiments in Tab. 1, Fig. 1 and Fig. 2 analyze metrics related to $P(Y \mid X, A)$ and the resulting accuracy and fairness violations.

## C. Bayesian Perspective on the Average Predictive Diversity

In this section, we motivate the average predictive diversity $\overline{\mathrm{DIV}}$ (*cf.* Eq. (8)) from a Bayesian perspective. Given are a training dataset $\mathcal{D} = \{(\boldsymbol{x}_j, y_j)\}_{j=1}^{J}$ as well as a test dataset $\mathcal{D}' = \{(\boldsymbol{x}_k, y_k)\}_{k=1}^{K}$; the protected attribute is omitted for brevity in this section. Furthermore, we are given a prior distribution $p(\boldsymbol{w})$ on the model parameters.

**Marginal Likelihood.**  Through Bayes' rule, we obtain a posterior distribution over the model parameters given the training dataset $p(\boldsymbol{w} \mid \mathcal{D}) = p(\mathcal{D} \mid \boldsymbol{w})p(\boldsymbol{w})/p(\mathcal{D})$. Recall that the marginal likelihood is given by $p(\mathcal{D}) = \int_W p(\mathcal{D} \mid \boldsymbol{w})p(\boldsymbol{w})\mathrm{d}\boldsymbol{w}$, *i.e.,* the expected likelihood on the dataset over all models according to their prior distribution. Intuitively, the marginal likelihood thus measures how well possible models represent the given dataset.

The disparate benefits effect occurs on a test dataset $\mathcal{D}'$. Consequently, we are interested in the marginal likelihood under the test dataset $p(\mathcal{D}')$. For the test dataset $\mathcal{D}'$, the posterior distribution given the training dataset $p(\boldsymbol{w} \mid \mathcal{D})$ is the new prior distribution $p(\boldsymbol{w})$. The marginal likelihood under the test dataset is thus given by

$$p(\mathcal{D}') = \int_W \prod_{k=1}^{K} p(y = y_k \mid \boldsymbol{x}_k, \boldsymbol{w})\, p(\boldsymbol{w})\, \mathrm{d}\boldsymbol{w} \approx \frac{1}{N} \sum_{n=1}^{N} \prod_{k=1}^{K} p(y = y_k \mid \boldsymbol{x}_k, \boldsymbol{w}_n) , \tag{12}$$

with $\boldsymbol{w}_n$ drawn according to $p(\boldsymbol{w}) = p(\boldsymbol{w} \mid \mathcal{D})$. In practice, the set of model parameters $\{\boldsymbol{w}_n\}_{n=1}^{N}$ obtained from the training of the Deep Ensemble is used to approximate the integral.

**Likelihood Ratio.**  If the likelihood under the posterior predictive distribution

$$\bar{p}(\mathcal{D}') = \prod_{k=1}^{K} \int_W p(y = y_k \mid \boldsymbol{x}_k, \boldsymbol{w})\, p(\boldsymbol{w} \mid \mathcal{D})\, \mathrm{d}\boldsymbol{w} \approx \prod_{k=1}^{K} \frac{1}{N} \sum_{n=1}^{N} p(y = y_k \mid \boldsymbol{x}_k, \boldsymbol{w}_n) , \tag{13}$$

again with $\boldsymbol{w}_n$ drawn according to $p(\boldsymbol{w}) = p(\boldsymbol{w} \mid \mathcal{D})$, does not differ from the marginal likelihood, there is no difference between predicting with a single model sampled according to the posterior and predicting with the ensemble of all sampled models. Thus, we investigate the likelihood ratio $\bar{p}(\mathcal{D}')/p(\mathcal{D}')$ as a natural measure of diversity in the predictions of the models that make up the ensemble.

For practical purposes, it is more convinient to work with log-likelihoods rather than likelihoods, as the products in Eq. (13) and Eq. (13) become sums. Therefore, we consider the logarithm of the likelihood ratio, leading to

$$\log\left(\frac{\bar{p}(\mathcal{D}')}{p(\mathcal{D}')}\right) = \log\bar{p}(\mathcal{D}') - \log p(\mathcal{D}') . \tag{14}$$

Inserting Eq. (12) and Eq. (13) into Eq. (14) we obtain

$$\log\left(\frac{\bar{p}(\mathcal{D}')}{p(\mathcal{D}')}\right) \approx \sum_{k=1}^{K}\log\left(\frac{1}{N}\sum_{n=1}^{N}p(y=y_k \mid \boldsymbol{x}_k, \boldsymbol{w}_n)\right) - \frac{1}{N}\sum_{n=1}^{N}\log p(y=y_k \mid \boldsymbol{x}_k, \boldsymbol{w}_n) \tag{15}$$
$$= K\,\overline{\text{DIV}},$$

with $\overline{\text{DIV}}$ as defined in Eq. (8), which is what we wanted to show. Eq. (15) is $\sum_{k=1}^{K}\text{DIV}$, with the predictive diversity DIV given by Theorem 4.3 in Jeffares et al. (2023). To mitigate the impact of different dataset sizes, it is common practice to divide log-likelihoods by the number of datapoints in the dataset $K$ when comparing between datasets of different sizes. Doing so for the logarithm of the likelihood ratio, $1/K\log\left(\bar{p}(\mathcal{D}')/p(\mathcal{D}')\right)$ is an approximation of the Jensen gap (Eq. (5) in Abe et al. (2022a) and Eq. (3) in Abe et al. (2024)) with $K$ samples in the dataset $\mathcal{D}'$.

# D. Details of the Experimental Setup

The code to reproduce our experiments is available at https://github.com/ml-jku/disparate-benefits.

## D.1. Datasets

We conducted all our experiments on facial analysis and medical imaging datasets. In the following, we provide details about the datasets.

**Facial Analysis.** We used two widely used facial analysis datasets, FairFace[1] (Karkkainen & Joo, 2021) (License: CC BY 4.0) and UTKFace[2] (Zhang et al., 2017) (License: research only, not commercial). FairFace was created for advancing research in fairness, accountability and transparency in computer vision as it addresses the lack of diversity in existing face datasets used for research purposes. The FairFace dataset comprises 108,501 facial images collected from publicly available sources, such as Flickr and Google Images, and covers a diverse range of demographics, including various ethnicities, ages, genders, and skin tones. The dataset includes annotations for gender, age, and ethnicity. UTKFace contains over 20,000 facial images of individuals collected from the publicly available datasets UTKinect (Xia et al., 2012) and FGNET (Lanitis et al., 2002), as well as images scraped from the internet. It includes annotations for three demographic attributes: age, gender, and ethnicity.

**Medical Imaging.** We used the medical imaing dataset CheXpert [3] (Irvin et al., 2019) (License: Stanford University Dataset Research Use Agreement). It consists of a large publicly available dataset of 224,316 chest X-rays along with associated radiologist-labeled annotations for the presence or absence of 14 different thoracic pathologies. It is designed to address the challenges of class imbalance and target noise commonly encountered in medical image classification tasks. CheXpert has become a widely used benchmark dataset in the field of medical imaging and has been instrumental in advancing research on automated chest radiograph interpretation, particularly in the context of deep learning approaches. We use the recommended targets provided by Jain et al. (2021) (visualCheXbert targets) and group attributes provided by Gichoya et al. (2022)[4].

## D.2. Models and Training

We used the ResNet18/24/50, RegNet-Y 800MF and EfficientNetV2-S implementations of Pytorch (Paszke et al., 2019). Hyperparameters as reported in the main paper were the result of an initial manual tuning on the respective validation

---

[1]Obtained from https://github.com/joojs/fairface using the [Padding=0.25] version.

[2]Obtained from https://www.kaggle.com/datasets/abhikjha/utk-face-cropped as the download link on the original source https://susanqq.github.io/UTKFace does no longer work.

[3]Obtained from https://stanfordaimi.azurewebsites.net/datasets/8cbd9ed4-2eb9-4565-affc-111cf4f7ebe2, user account required.

[4]Obtained from https://stanfordaimi.azurewebsites.net/datasets/192ada7c-4d43-466e-b8bb-b81992bb80cf, user account required.

Table 2: **Results for controlled experiments.** Performance and fairness violations of individual ensemble members, the Deep Ensemble as well as the change in performance and fairness violation due to ensembling. Gray cells denote the results of the controlled experiment in Sec. 6 in the main paper.

| Individual Ensemble Members | | | | |
|---|---|---|---|---|
| Setting | Accuracy ($\uparrow$) | SPD ($\downarrow$) | EOD ($\downarrow$) | AOD ($\downarrow$) |
| Main Paper (Fig. 4a) | $0.894_{\pm 0.005}$ | $0.048_{\pm 0.011}$ | $0.080_{\pm 0.016}$ | $0.064_{\pm 0.008}$ |
| $\alpha = 0.0$ (Fig. 5a) | $0.844_{\pm 0.005}$ | $0.029_{\pm 0.005}$ | $0.015_{\pm 0.009}$ | $0.016_{\pm 0.006}$ |
| $\alpha = 0.2$ (Fig. 5b) | $0.860_{\pm 0.005}$ | $0.024_{\pm 0.016}$ | $0.033_{\pm 0.018}$ | $0.038_{\pm 0.009}$ |
| $\alpha = 0.4$ (Fig. 5c) | $0.871_{\pm 0.005}$ | $0.039_{\pm 0.014}$ | $0.069_{\pm 0.016}$ | $0.060_{\pm 0.008}$ |
| $\alpha = 1.0$ (Fig. 5d) | $0.880_{\pm 0.006}$ | $0.041_{\pm 0.024}$ | $0.079_{\pm 0.027}$ | $0.068_{\pm 0.009}$ |
| **Deep Ensemble** | | | | |
| Setting | Accuracy ($\uparrow$) | SPD ($\downarrow$) | EOD ($\downarrow$) | AOD ($\downarrow$) |
| Main Paper (Fig. 4a) | $0.924_{\pm 0.002}$ | $0.057_{\pm 0.005}$ | $0.133_{\pm 0.007}$ | $0.111_{\pm 0.004}$ |
| $\alpha = 0.0$ (Fig. 5a) | $0.849_{\pm 0.003}$ | $0.033_{\pm 0.004}$ | $0.010_{\pm 0.005}$ | $0.015_{\pm 0.002}$ |
| $\alpha = 0.2$ (Fig. 5b) | $0.876_{\pm 0.002}$ | $0.034_{\pm 0.010}$ | $0.047_{\pm 0.017}$ | $0.043_{\pm 0.010}$ |
| $\alpha = 0.4$ (Fig. 5c) | $0.896_{\pm 0.002}$ | $0.054_{\pm 0.008}$ | $0.105_{\pm 0.013}$ | $0.084_{\pm 0.006}$ |
| $\alpha = 1.0$ (Fig. 5d) | $0.910_{\pm 0.003}$ | $0.058_{\pm 0.017}$ | $0.133_{\pm 0.021}$ | $0.108_{\pm 0.005}$ |
| **Difference ($\Delta$) between Deep Ensemble and individual members** | | | | |
| Setting | $\Delta$ Accuracy ($\uparrow$) | $\Delta$ SPD ($\downarrow$) | $\Delta$ EOD ($\downarrow$) | $\Delta$ AOD ($\downarrow$) |
| Main Paper (Fig. 4a) | $0.030_{\pm 0.002}$ | $0.009_{\pm 0.005}$ | $0.054_{\pm 0.007}$ | $0.047_{\pm 0.004}$ |
| $\alpha = 0.0$ (Fig. 5a) | $0.005_{\pm 0.001}$ | $0.004_{\pm 0.005}$ | $-0.004_{\pm 0.007}$ | $-0.001_{\pm 0.003}$ |
| $\alpha = 0.2$ (Fig. 5b) | $0.017_{\pm 0.003}$ | $0.010_{\pm 0.006}$ | $0.014_{\pm 0.011}$ | $0.005_{\pm 0.011}$ |
| $\alpha = 0.4$ (Fig. 5c) | $0.025_{\pm 0.002}$ | $0.015_{\pm 0.009}$ | $0.037_{\pm 0.011}$ | $0.024_{\pm 0.006}$ |
| $\alpha = 1.0$ (Fig. 5d) | $0.030_{\pm 0.002}$ | $0.017_{\pm 0.009}$ | $0.055_{\pm 0.012}$ | $0.040_{\pm 0.004}$ |

sets, but mostly align with commonly utilized hyperparameters for classical image datasets such as CIFAR10. The raw performance on the task was not of extreme importance, but is comparable to previous studies on the same datasets with similar network architectures (Karkkainen & Joo, 2021; Zhang et al., 2022; Zong et al., 2023).

### D.3. Computational Cost

For training the models, we utilized a mixture of P100, RTX 3090, A40 and A100 GPUs, depending on availablility in our cluster. Training a single model took around 3 hours on average over all considered model architectures and datasets, resulting in 3,000 GPU-hours. Evaluating these models on the test datasets accounted for approximately 150 additional GPU-hours.

## E. Complete Experimental Results

The experimental results included in the main paper describe a subset of all the considered tasks. In this section, we provide the results of the complete set, along with additional supporting tables and figures.

**Detailed Results on Controlled Experiments.** The detailed results for the controlled experiments in the main paper (Sec. 6) are provided in Tab. 2. We report the absolute accuracies, SPDs, EODs and AODs for individual ensemble members, the Deep Ensemble and the differences between those.

**Performance and fairness violation of Deep Ensemble and individual members.** Tab. 4 and Tab. 3 contain the performance and fairness violations of individual ensemble members and the resulting Deep Ensemble, respectively.

**The disparate benefits effect of Deep Ensembles.** Fig. 9 - 11 depict the change in performance and fairness violations when adding individual ensemble members for all considered tasks.

Table 3: Performance and fairness violations of Deep Ensembles (10 members). Statistics are obtained from five independent runs.

| $\mathcal{D}'$ | Target / Group | Accuracy ($\uparrow$) | SPD ($\downarrow$) | EOD ($\downarrow$) | AOD ($\downarrow$) |
|---|---|---|---|---|---|
| FF | age / gender | $0.812_{\pm0.007}$ | $0.190_{\pm0.009}$ | $0.165_{\pm0.010}$ | $0.126_{\pm0.008}$ |
| FF | age / race | $0.812_{\pm0.007}$ | $0.112_{\pm0.008}$ | $0.063_{\pm0.011}$ | $0.075_{\pm0.008}$ |
| FF | gender / age | $0.909_{\pm0.004}$ | $0.142_{\pm0.003}$ | $0.109_{\pm0.005}$ | $0.065_{\pm0.004}$ |
| FF | gender / race | $0.909_{\pm0.004}$ | $0.009_{\pm0.003}$ | $0.003_{\pm0.004}$ | $0.004_{\pm0.003}$ |
| FF | race / age | $0.885_{\pm0.004}$ | $0.035_{\pm0.003}$ | $0.038_{\pm0.014}$ | $0.025_{\pm0.006}$ |
| FF | race / gender | $0.885_{\pm0.004}$ | $0.005_{\pm0.004}$ | $0.025_{\pm0.010}$ | $0.015_{\pm0.005}$ |
| UTK | age / gender | $0.793_{\pm0.005}$ | $0.309_{\pm0.009}$ | $0.252_{\pm0.009}$ | $0.204_{\pm0.008}$ |
| UTK | age / race | $0.793_{\pm0.005}$ | $0.214_{\pm0.006}$ | $0.188_{\pm0.007}$ | $0.106_{\pm0.005}$ |
| UTK | gender / age | $0.923_{\pm0.003}$ | $0.180_{\pm0.003}$ | $0.083_{\pm0.004}$ | $0.054_{\pm0.002}$ |
| UTK | gender / race | $0.923_{\pm0.003}$ | $0.002_{\pm0.002}$ | $0.023_{\pm0.003}$ | $0.029_{\pm0.002}$ |
| UTK | race / age | $0.840_{\pm0.006}$ | $0.129_{\pm0.004}$ | $0.079_{\pm0.008}$ | $0.044_{\pm0.005}$ |
| UTK | race / gender | $0.840_{\pm0.006}$ | $0.010_{\pm0.004}$ | $0.024_{\pm0.008}$ | $0.014_{\pm0.004}$ |
| $\mathcal{D}'$ | Group | AUROC ($\uparrow$) | SPD ($\downarrow$) | EOD ($\downarrow$) | AOD ($\downarrow$) |
| CX | age | $0.943_{\pm0.001}$ | $0.139_{\pm0.002}$ | $0.181_{\pm0.006}$ | $0.104_{\pm0.003}$ |
| CX | gender | $0.943_{\pm0.001}$ | $0.000_{\pm0.001}$ | $0.024_{\pm0.006}$ | $0.014_{\pm0.003}$ |
| CX | race | $0.943_{\pm0.001}$ | $0.040_{\pm0.001}$ | $0.092_{\pm0.005}$ | $0.048_{\pm0.002}$ |

**Changes in PR, TPR and FPR.** Fig. 12 - 14 display the change in PR, TPR and FPR per group when adding individual ensemble members for all considered tasks.

**Difference in average predictive diversity.** Fig. 15 - 17 depict the differences in average predictive diversity per target and protected group.

**Hardt post-processing (HPP).** Tab. 5 - 19 contain the results of mitigating unfairness by means of HPP (Hardt et al., 2016) on all considered tasks. HPP was either applied with the threshold set to the average fairness violation of the individual ensemble members on the validation set (val) or to 0.05. Note that for some tasks, the original fairness violation of both the Deep Ensemble and its members was already lower than 0.05, where HPP leads to an increase in unfairness up to the desired threshold. Experiments on FairFace and CheXpert use the respective validation sets to learn the group dependent thresholds in HPP. For experiments on UTKFace, the FairFace validation set was used to learn the thresholds, as it was designed to emulate a real-world distribution shift scenario. Also for UTKFace, the same conclusions as for the FairFace experiments described in the main paper hold, *i.e.*, while HPP is very effective to mitigate unfairness in the Deep Ensembles, the desired fairness violation (0.05) is not reached due to the distribution shift. Note that the balanced accuracy was used as the performance metric for CheXpert, because the metric investigated in the main paper, the AUROC, does not consider selecting a threshold.

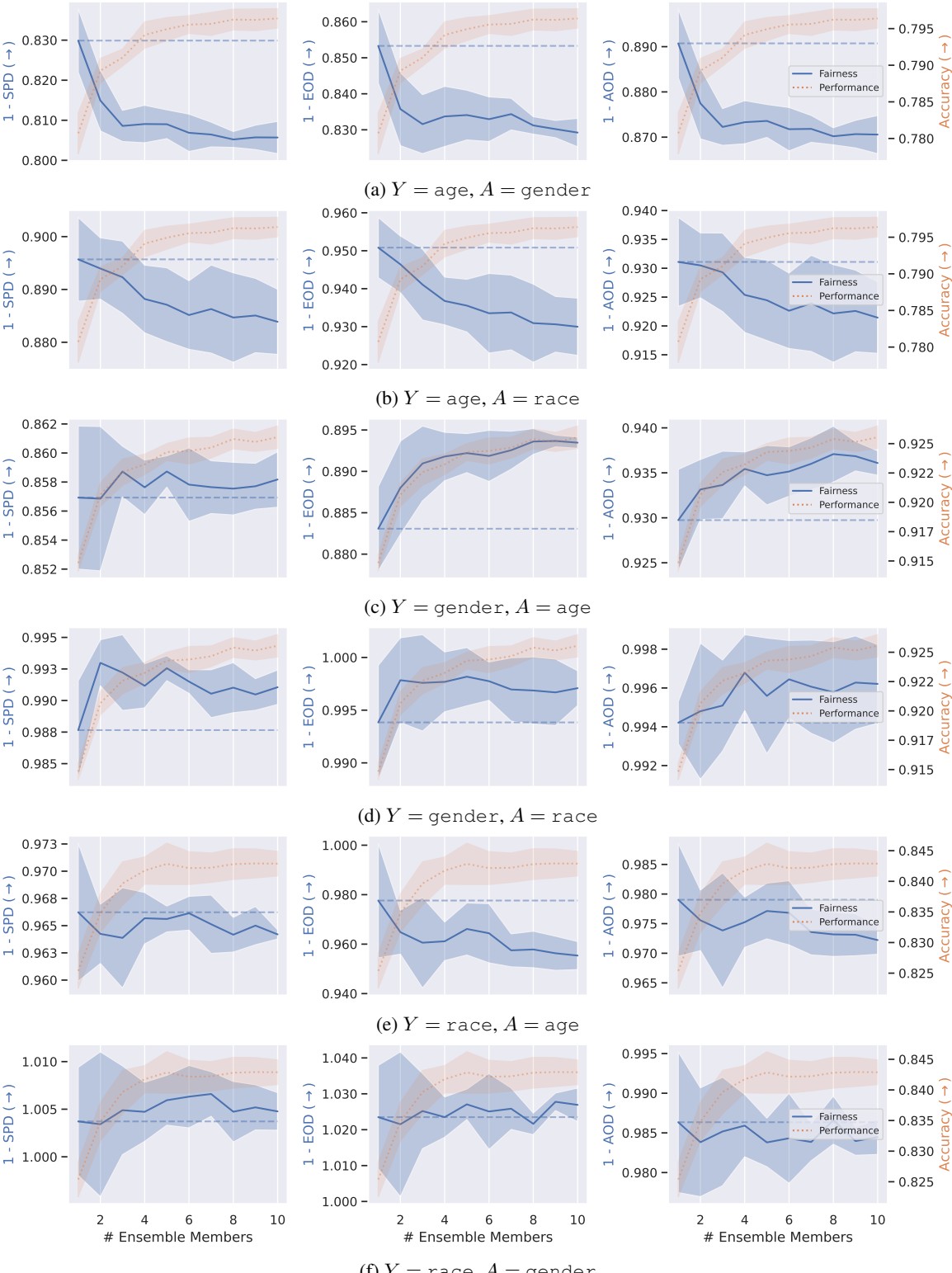

Figure 9: The disparate benefits effect of Deep Ensembles. The performance increases, but also the fairness changes, often decreasing, when adding more members to the ensemble. Models are trained and evaluated on the FF dataset. Statistics are computed based on five independent runs.

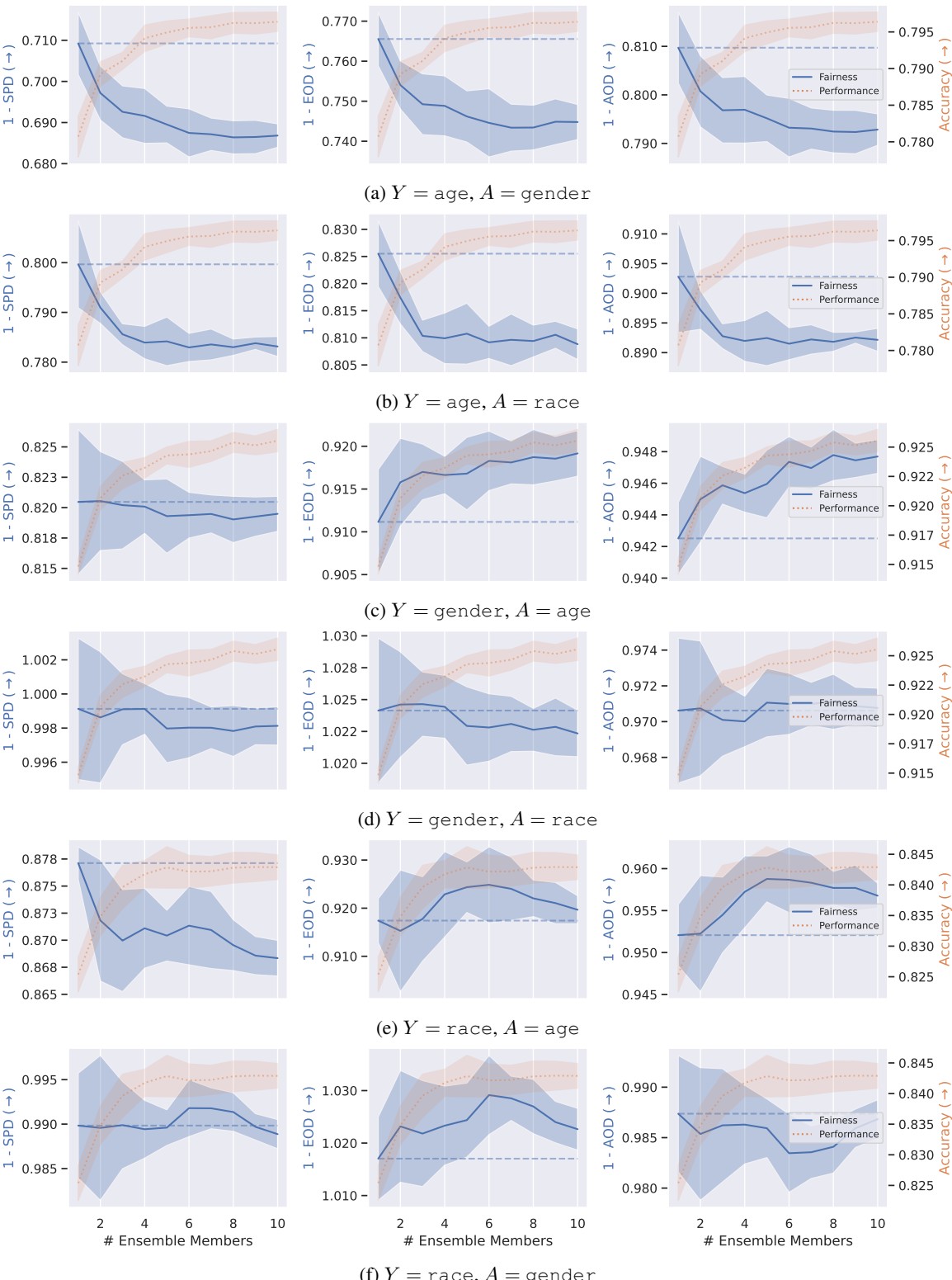

(a) $Y = \texttt{age}, A = \texttt{gender}$

(b) $Y = \texttt{age}, A = \texttt{race}$

(c) $Y = \texttt{gender}, A = \texttt{age}$

(d) $Y = \texttt{gender}, A = \texttt{race}$

(e) $Y = \texttt{race}, A = \texttt{age}$

(f) $Y = \texttt{race}, A = \texttt{gender}$

Figure 10: The disparate benefits effect of Deep Ensembles. The performance increases, but also the fairness changes, often decreasing, when adding more members to the ensemble. Models are trained on FF and evaluated on the UTK dataset. Statistics are computed based on five independent runs.

Table 4: Performance and fairness violations of individual members. Statistics are obtained from five independent runs.

| $\mathcal{D}'$ | Target / Group | Accuracy ($\uparrow$) | SPD ($\downarrow$) | EOD ($\downarrow$) | AOD ($\downarrow$) |
|---|---|---|---|---|---|
| FF | age / gender | $0.794_{\pm 0.001}$ | $0.173_{\pm 0.001}$ | $0.153_{\pm 0.002}$ | $0.113_{\pm 0.001}$ |
| FF | age / race | $0.794_{\pm 0.001}$ | $0.107_{\pm 0.004}$ | $0.058_{\pm 0.004}$ | $0.072_{\pm 0.004}$ |
| FF | gender / age | $0.899_{\pm 0.001}$ | $0.142_{\pm 0.001}$ | $0.114_{\pm 0.001}$ | $0.068_{\pm 0.001}$ |
| FF | gender / race | $0.899_{\pm 0.001}$ | $0.010_{\pm 0.001}$ | $0.003_{\pm 0.001}$ | $0.006_{\pm 0.001}$ |
| FF | race / age | $0.873_{\pm 0.000}$ | $0.040_{\pm 0.001}$ | $0.040_{\pm 0.005}$ | $0.029_{\pm 0.002}$ |
| FF | race / gender | $0.873_{\pm 0.000}$ | $0.004_{\pm 0.002}$ | $0.019_{\pm 0.003}$ | $0.013_{\pm 0.002}$ |
| UTK | age / gender | $0.782_{\pm 0.001}$ | $0.296_{\pm 0.003}$ | $0.240_{\pm 0.003}$ | $0.195_{\pm 0.003}$ |
| UTK | age / race | $0.782_{\pm 0.001}$ | $0.207_{\pm 0.002}$ | $0.182_{\pm 0.003}$ | $0.104_{\pm 0.002}$ |
| UTK | gender / age | $0.916_{\pm 0.001}$ | $0.180_{\pm 0.002}$ | $0.087_{\pm 0.003}$ | $0.056_{\pm 0.001}$ |
| UTK | gender / race | $0.916_{\pm 0.001}$ | $0.002_{\pm 0.001}$ | $0.023_{\pm 0.002}$ | $0.028_{\pm 0.001}$ |
| UTK | race / age | $0.822_{\pm 0.002}$ | $0.118_{\pm 0.001}$ | $0.073_{\pm 0.002}$ | $0.043_{\pm 0.001}$ |
| UTK | race / gender | $0.822_{\pm 0.002}$ | $0.008_{\pm 0.001}$ | $0.021_{\pm 0.002}$ | $0.015_{\pm 0.001}$ |
| $\mathcal{D}'$ | Group | AUROC ($\uparrow$) | SPD ($\downarrow$) | EOD ($\downarrow$) | AOD ($\downarrow$) |
| CX | age | $0.940_{\pm 0.000}$ | $0.138_{\pm 0.001}$ | $0.174_{\pm 0.003}$ | $0.101_{\pm 0.001}$ |
| CX | gender | $0.940_{\pm 0.000}$ | $0.000_{\pm 0.001}$ | $0.024_{\pm 0.003}$ | $0.014_{\pm 0.001}$ |
| CX | race | $0.940_{\pm 0.000}$ | $0.041_{\pm 0.000}$ | $0.091_{\pm 0.003}$ | $0.049_{\pm 0.001}$ |

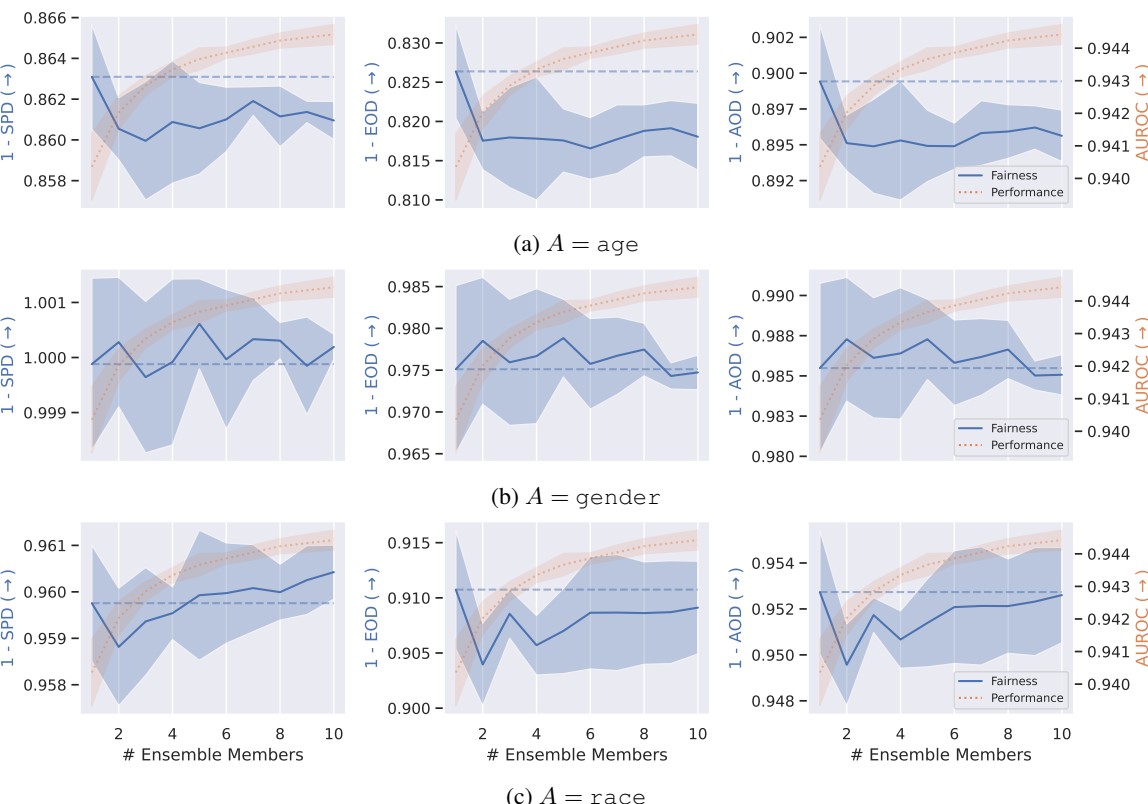

Figure 11: The disparate benefits effect of Deep Ensembles. The performance increases, but also the fairness changes, often decreasing, when adding more members to the ensemble. Models are trained and evaluated on the CX dataset. Statistics are computed based on five independent runs.

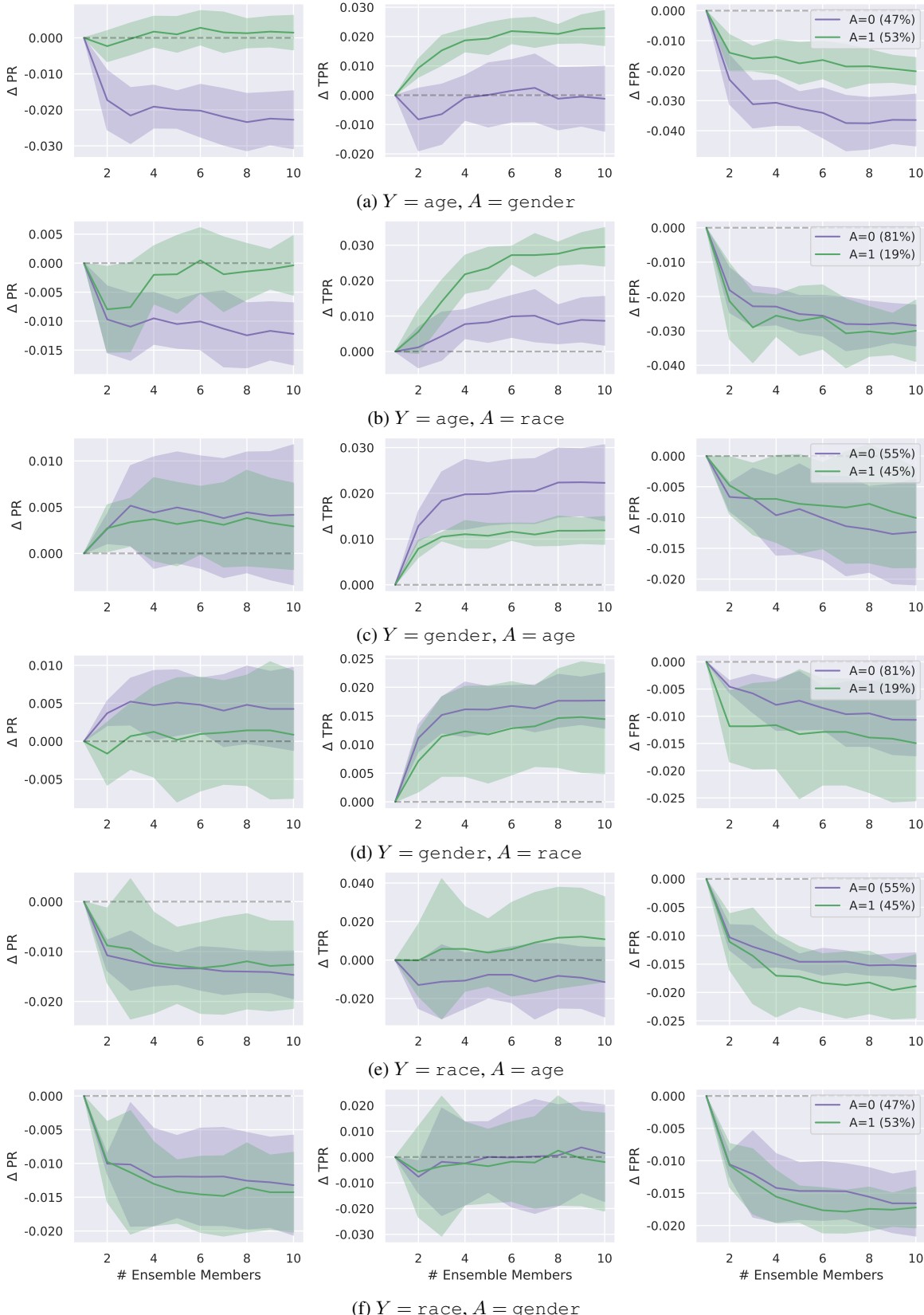

Figure 12: Changes in PR, TPR and FPR for a Deep Ensemble (10 members) on the FF dataset. Statistics are computed based on five independent runs.

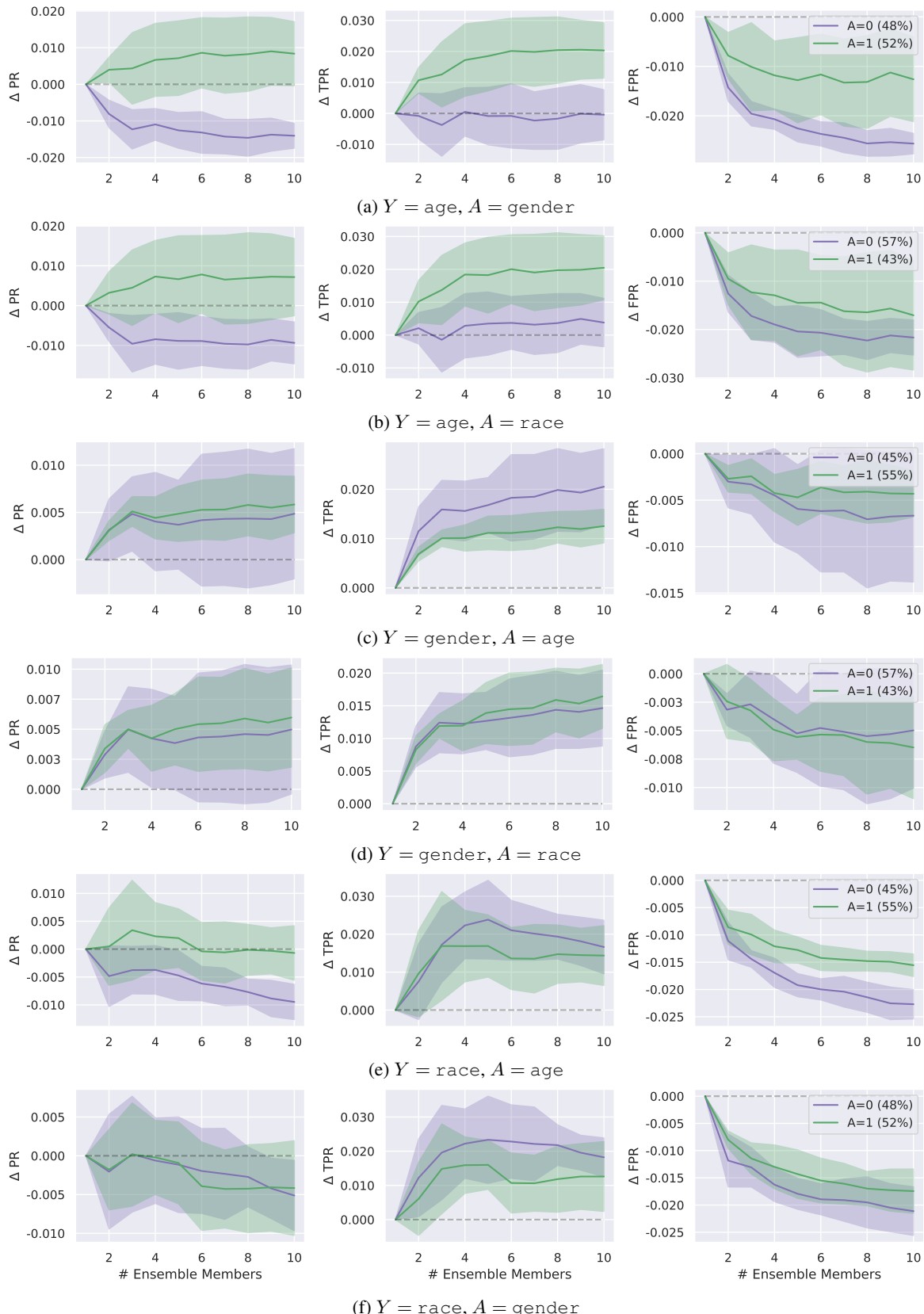

Figure 13: Changes in PR, TPR and FPR for a Deep Ensemble (10 members) on the UTK dataset. Statistics are computed based on five independent runs.

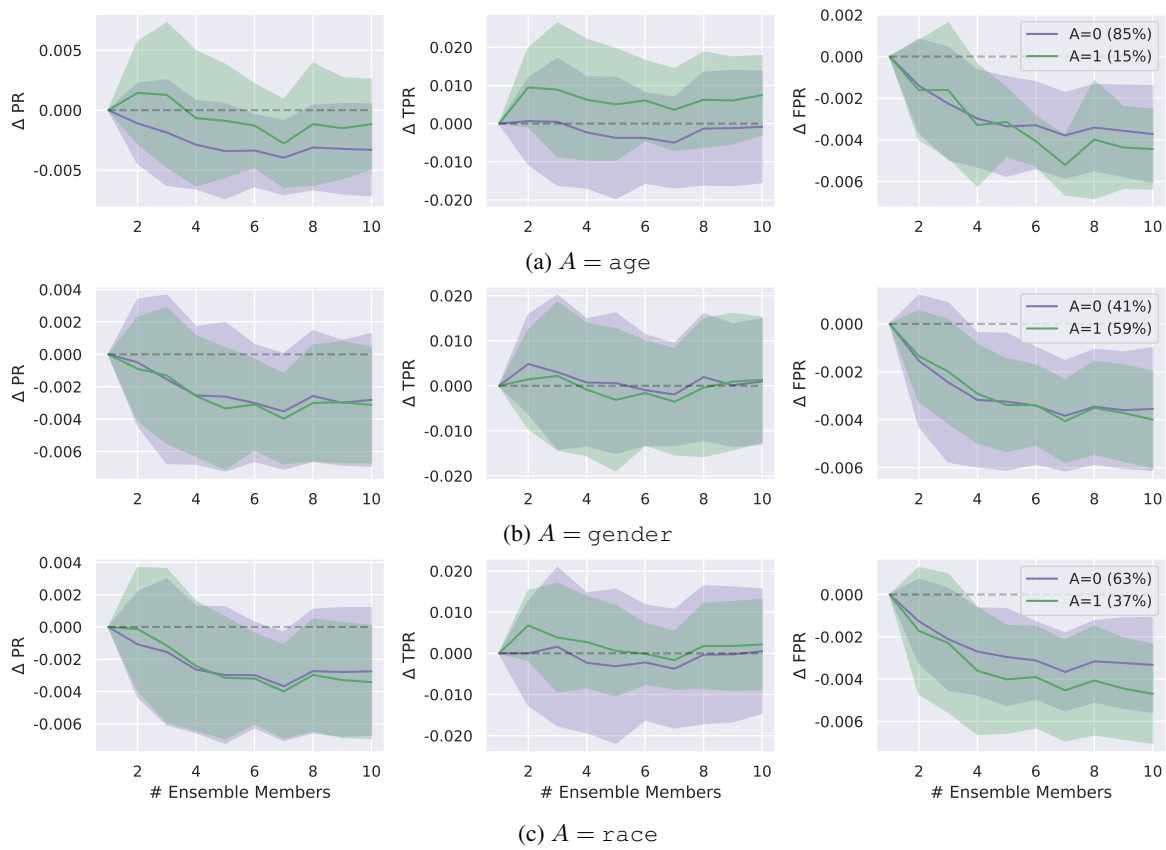

Figure 14: Changes in PR, TPR and FPR for a Deep Ensemble (10 members) on the CX dataset. Statistics are computed based on five independent runs.

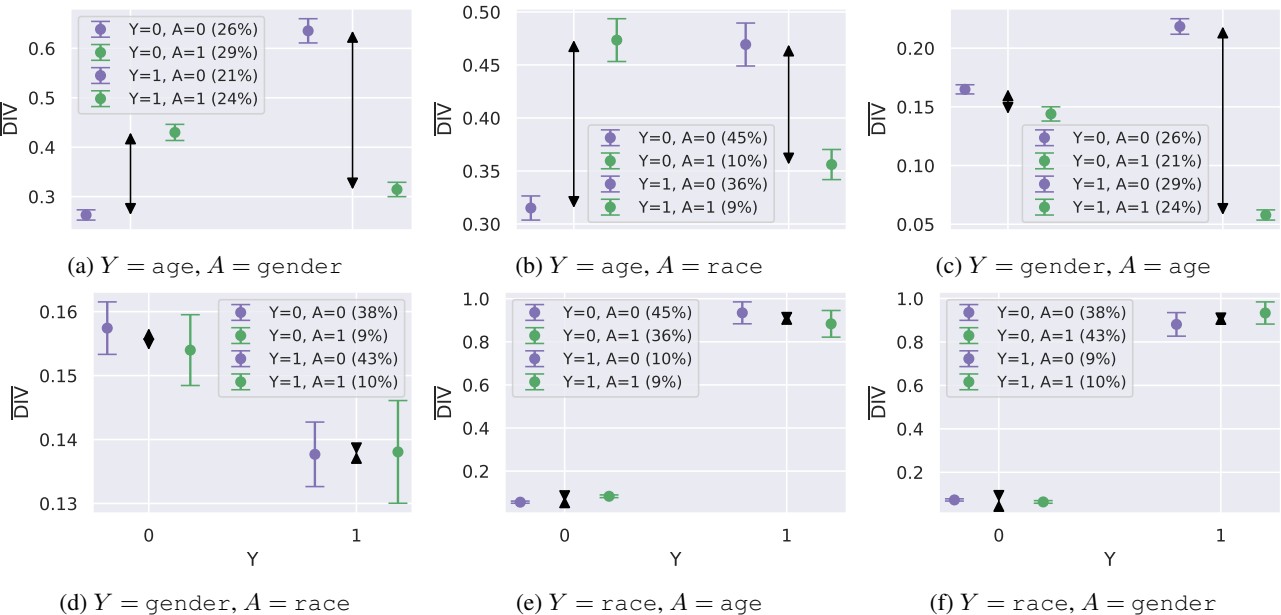

Figure 15: Average predictive diversity $\overline{\text{DIV}}$ for each value of the protected attribute $A$ and target variable $Y$ on the FF dataset. Statistics are obtained from five independent runs.

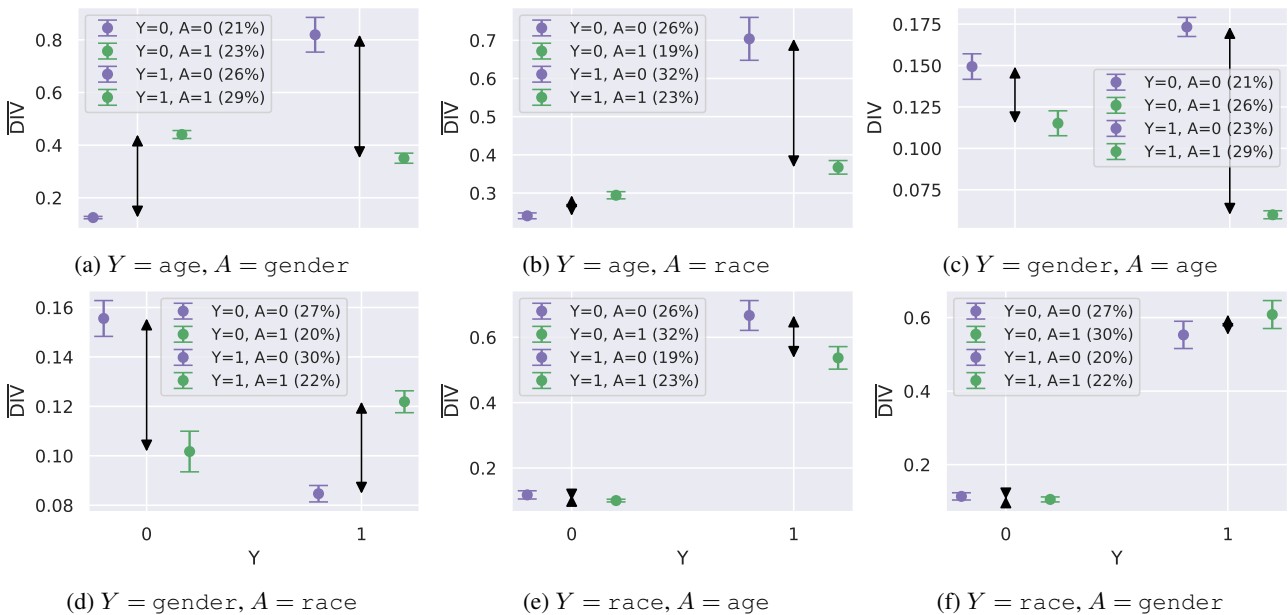

Figure 16: Average predictive diversity $\overline{\text{DIV}}$ for each value of the protected attribute $A$ and target variable $Y$ on the UTK dataset. Statistics are obtained from five independent runs.

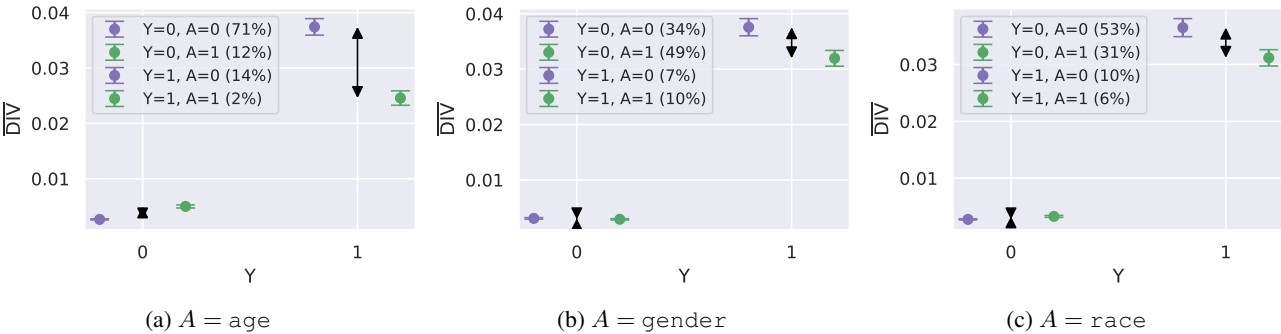

Figure 17: Average predictive diversity $\overline{\text{DIV}}$ for each value of the protected attribute $A$ and target variable $Y$ on the CX dataset. Statistics are obtained from five independent runs.

Table 5: HPP results (accuracy and fairness violation metrics) on FF. Models are trained on target variable `age`, evaluated using protected attribute `gender`. Statistics are obtained from five independent runs, and additionally over all individual ensemble members if applicable.

| Before HPP | Acc (↑) | SPD (↓) | Acc (↑) | EOD (↓) | Acc (↑) | AOD (↓) |
|---|---|---|---|---|---|---|
| Members | $0.794_{\pm.003}$ | $0.173_{\pm.007}$ | $0.794_{\pm.003}$ | $0.153_{\pm.012}$ | $0.794_{\pm.003}$ | $0.113_{\pm.008}$ |
| Deep Ensemble | $0.816_{\pm.002}$ | $0.194_{\pm.004}$ | $0.816_{\pm.002}$ | $0.171_{\pm.004}$ | $0.816_{\pm.002}$ | $0.129_{\pm.004}$ |
| After HPP | HPP-SPD (↓) | | HPP-EOD (↓) | | HPP-AOD (↓) | |
| Deep Ens. (val) | $0.818_{\pm.001}$ | $0.176_{\pm.011}$ | $0.818_{\pm.001}$ | $0.157_{\pm.014}$ | $0.818_{\pm.001}$ | $0.114_{\pm.012}$ |
| Deep Ens. (0.05) | $0.818_{\pm.001}$ | $0.057_{\pm.003}$ | $0.815_{\pm.002}$ | $0.067_{\pm.006}$ | $0.816_{\pm.002}$ | $0.062_{\pm.002}$ |
| Members (0.05) | $0.789_{\pm.005}$ | $0.056_{\pm.024}$ | $0.792_{\pm.005}$ | $0.055_{\pm.021}$ | $0.793_{\pm.005}$ | $0.054_{\pm.015}$ |

Table 6: HPP results results (accuracy and fairness violation metrics) on FF. Models are trained on target variable `age`, evaluated using protected attribute `race`. Statistics are obtained from five independent runs, and additionally over all individual ensemble members if applicable.

| Before HPP | Acc (↑) | SPD (↓) | Acc (↑) | EOD (↓) | Acc (↑) | AOD (↓) |
|---|---|---|---|---|---|---|
| Members | $0.794_{\pm.003}$ | $0.107_{\pm.007}$ | $0.794_{\pm.003}$ | $0.058_{\pm.011}$ | $0.794_{\pm.003}$ | $0.072_{\pm.007}$ |
| Deep Ensemble | $0.816_{\pm.001}$ | $0.116_{\pm.006}$ | $0.816_{\pm.001}$ | $0.070_{\pm.008}$ | $0.816_{\pm.001}$ | $0.079_{\pm.006}$ |
| After HPP | HPP-SPD (↓) | | HPP-EOD (↓) | | HPP-AOD (↓) | |
| Deep Ens. (val) | $0.818_{\pm.001}$ | $0.070_{\pm.011}$ | $0.818_{\pm.001}$ | $0.041_{\pm.006}$ | $0.818_{\pm.001}$ | $0.032_{\pm.012}$ |
| Deep Ens. (0.05) | $0.818_{\pm.001}$ | $0.063_{\pm.007}$ | $0.818_{\pm.001}$ | $0.033_{\pm.013}$ | $0.818_{\pm.001}$ | $0.032_{\pm.011}$ |
| Members (0.05) | $0.795_{\pm.004}$ | $0.061_{\pm.015}$ | $0.795_{\pm.004}$ | $0.049_{\pm.028}$ | $0.795_{\pm.004}$ | $0.054_{\pm.018}$ |

Table 7: HPP results (accuracy and fairness violation metrics) on FF. Models are trained on target variable `gender`, evaluated using protected attribute `age`. Statistics are obtained from five independent runs, and additionally over all individual ensemble members if applicable.

| Before HPP | Acc (↑) | SPD (↓) | Acc (↑) | EOD (↓) | Acc (↑) | AOD (↓) |
|---|---|---|---|---|---|---|
| Members | $0.899_{\pm.003}$ | $0.142_{\pm.005}$ | $0.899_{\pm.003}$ | $0.114_{\pm.007}$ | $0.899_{\pm.003}$ | $0.068_{\pm.005}$ |
| Deep Ensemble | $0.913_{\pm.001}$ | $0.142_{\pm.002}$ | $0.913_{\pm.001}$ | $0.107_{\pm.001}$ | $0.913_{\pm.001}$ | $0.064_{\pm.001}$ |
| After HPP | HPP-SPD (↓) | | HPP-EOD (↓) | | HPP-AOD (↓) | |
| Deep Ens. (val) | $0.913_{\pm.001}$ | $0.116_{\pm.015}$ | $0.913_{\pm.001}$ | $0.084_{\pm.015}$ | $0.913_{\pm.001}$ | $0.067_{\pm.001}$ |
| Deep Ens. (0.05) | $0.911_{\pm.001}$ | $0.055_{\pm.003}$ | $0.913_{\pm.001}$ | $0.054_{\pm.003}$ | $0.913_{\pm.001}$ | $0.067_{\pm.001}$ |
| Members (0.05) | $0.894_{\pm.004}$ | $0.048_{\pm.016}$ | $0.897_{\pm.004}$ | $0.048_{\pm.013}$ | $0.898_{\pm.003}$ | $0.072_{\pm.005}$ |

Table 8: HPP results (accuracy and fairness violation metrics) on FF. Models are trained on target variable `gender`, evaluated using protected attribute `race`. Statistics are obtained from five independent runs, and additionally over all individual ensemble members if applicable.

| Before HPP | Acc (↑) | SPD (↓) | Acc (↑) | EOD (↓) | Acc (↑) | AOD (↓) |
|---|---|---|---|---|---|---|
| Members | $0.899_{\pm.003}$ | $0.010_{\pm.004}$ | $0.899_{\pm.003}$ | $0.003_{\pm.005}$ | $0.899_{\pm.003}$ | $0.006_{\pm.003}$ |
| Deep Ensemble | $0.913_{\pm.001}$ | $0.009_{\pm.001}$ | $0.913_{\pm.001}$ | $0.003_{\pm.002}$ | $0.913_{\pm.001}$ | $0.004_{\pm.002}$ |
| After HPP | HPP-SPD (↓) | | HPP-EOD (↓) | | HPP-AOD (↓) | |
| Deep Ens. (val) | $0.912_{\pm.001}$ | $0.037_{\pm.005}$ | $0.912_{\pm.001}$ | $0.004_{\pm.007}$ | $0.912_{\pm.001}$ | $0.007_{\pm.001}$ |
| Deep Ens. (0.05) | $0.912_{\pm.001}$ | $0.009_{\pm.002}$ | $0.912_{\pm.001}$ | $0.024_{\pm.007}$ | $0.912_{\pm.001}$ | $0.032_{\pm.008}$ |
| Members (0.05) | $0.898_{\pm.003}$ | $0.002_{\pm.013}$ | $0.898_{\pm.003}$ | $0.007_{\pm.019}$ | $0.898_{\pm.003}$ | $0.017_{\pm.012}$ |

Table 9: HPP results (accuracy and fairness violation metrics) on FF. Models are trained on target variable `race`, evaluated using protected attribute `age`. Statistics are obtained from five independent runs, and additionally over all individual ensemble members if applicable.

| Before HPP | Acc (↑) | SPD (↓) | Acc (↑) | EOD (↓) | Acc (↑) | AOD (↓) |
|---|---|---|---|---|---|---|
| Members | $0.873_{\pm.002}$ | $0.040_{\pm.006}$ | $0.873_{\pm.002}$ | $0.040_{\pm.017}$ | $0.873_{\pm.002}$ | $0.029_{\pm.009}$ |
| Ensemble | $0.888_{\pm.001}$ | $0.036_{\pm.000}$ | $0.888_{\pm.001}$ | $0.045_{\pm.006}$ | $0.888_{\pm.001}$ | $0.028_{\pm.002}$ |
| After HPP | HPP-SPD (↓) | | HPP-EOD (↓) | | HPP-AOD (↓) | |
| Deep Ens. (val) | $0.887_{\pm.001}$ | $0.040_{\pm.003}$ | $0.888_{\pm.001}$ | $0.030_{\pm.011}$ | $0.887_{\pm.001}$ | $0.025_{\pm.006}$ |
| Deep Ens. (0.05) | $0.888_{\pm.001}$ | $0.052_{\pm.004}$ | $0.888_{\pm.001}$ | $0.054_{\pm.006}$ | $0.887_{\pm.001}$ | $0.052_{\pm.011}$ |
| Members (0.05) | $0.873_{\pm.004}$ | $0.030_{\pm.024}$ | $0.873_{\pm.004}$ | $0.018_{\pm.050}$ | $0.874_{\pm.004}$ | $0.038_{\pm.030}$ |

Table 10: HPP results (accuracy and fairness violation metrics) on FF. Models are trained on target variable `race`, evaluated using protected attribute `gender`. Statistics are obtained from five independent runs, and additionally over all individual ensemble members if applicable.

| Before HPP | Acc (↑) | SPD (↓) | Acc (↑) | EOD (↓) | Acc (↑) | AOD (↓) |
|---|---|---|---|---|---|---|
| Members | $0.873_{\pm.002}$ | $0.004_{\pm.005}$ | $0.873_{\pm.002}$ | $0.019_{\pm.016}$ | $0.873_{\pm.002}$ | $0.013_{\pm.006}$ |
| Ensemble | $0.888_{\pm.001}$ | $0.005_{\pm.002}$ | $0.888_{\pm.001}$ | $0.027_{\pm.005}$ | $0.888_{\pm.001}$ | $0.016_{\pm.002}$ |
| After HPP | HPP-SPD (↓) | | HPP-EOD (↓) | | HPP-AOD (↓) | |
| Deep Ens. (val) | $0.888_{\pm.001}$ | $0.012_{\pm.003}$ | $0.888_{\pm.001}$ | $0.005_{\pm.007}$ | $0.888_{\pm.001}$ | $0.016_{\pm.005}$ |
| Deep Ens. (0.05) | $0.888_{\pm.002}$ | $0.013_{\pm.010}$ | $0.888_{\pm.002}$ | $0.017_{\pm.022}$ | $0.888_{\pm.002}$ | $0.019_{\pm.004}$ |
| Members (0.05) | $0.873_{\pm.004}$ | $0.004_{\pm.025}$ | $0.873_{\pm.004}$ | $0.003_{\pm.044}$ | $0.873_{\pm.004}$ | $0.029_{\pm.027}$ |

Table 11: HPP results (accuracy and fairness violation metrics) on UTK. Models are trained on target variable `age`, evaluated using protected attribute `gender`. Statistics are obtained from five independent runs, and additionally over all individual ensemble members if applicable.

| Before HPP | Acc (↑) | SPD (↓) | Acc (↑) | EOD (↓) | Acc (↑) | AOD (↓) |
|---|---|---|---|---|---|---|
| Members | $0.782_{\pm.004}$ | $0.296_{\pm.008}$ | $0.782_{\pm.004}$ | $0.240_{\pm.012}$ | $0.782_{\pm.004}$ | $0.195_{\pm.008}$ |
| Ensemble | $0.796_{\pm.001}$ | $0.313_{\pm.003}$ | $0.796_{\pm.001}$ | $0.255_{\pm.004}$ | $0.796_{\pm.001}$ | $0.207_{\pm.003}$ |
| After HPP | HPP-SPD (↓) | | HPP-EOD (↓) | | HPP-AOD (↓) | |
| Deep Ens. (val) | $0.796_{\pm.002}$ | $0.299_{\pm.008}$ | $0.796_{\pm.002}$ | $0.245_{\pm.011}$ | $0.795_{\pm.002}$ | $0.194_{\pm.010}$ |
| Deep Ens. (0.05) | $0.795_{\pm.004}$ | $0.211_{\pm.005}$ | $0.796_{\pm.003}$ | $0.175_{\pm.007}$ | $0.797_{\pm.004}$ | $0.155_{\pm.006}$ |
| Members (0.05) | $0.777_{\pm.004}$ | $0.202_{\pm.021}$ | $0.778_{\pm.004}$ | $0.163_{\pm.018}$ | $0.778_{\pm.004}$ | $0.145_{\pm.013}$ |

Table 12: HPP results (accuracy and fairness violation metrics) on UTK. Models are trained on target variable `age`, evaluated using protected attribute `race`. Statistics are obtained from five independent runs, and additionally over all individual ensemble members if applicable.

| Before HPP | Acc (↑) | SPD (↓) | Acc (↑) | EOD (↓) | Acc (↑) | AOD (↓) |
|---|---|---|---|---|---|---|
| Members | $0.782_{\pm.004}$ | $0.207_{\pm.007}$ | $0.782_{\pm.004}$ | $0.182_{\pm.009}$ | $0.782_{\pm.004}$ | $0.104_{\pm.007}$ |
| Deep Ensemble | $0.796_{\pm.001}$ | $0.217_{\pm.002}$ | $0.796_{\pm.001}$ | $0.191_{\pm.003}$ | $0.796_{\pm.001}$ | $0.108_{\pm.002}$ |
| After HPP | HPP-SPD (↓) | | HPP-EOD (↓) | | HPP-AOD (↓) | |
| Deep Ens. (val) | $0.791_{\pm.001}$ | $0.188_{\pm.008}$ | $0.792_{\pm.001}$ | $0.168_{\pm.006}$ | $0.791_{\pm.001}$ | $0.085_{\pm.004}$ |
| Deep Ens. (0.05) | $0.791_{\pm.001}$ | $0.183_{\pm.005}$ | $0.791_{\pm.001}$ | $0.163_{\pm.010}$ | $0.791_{\pm.001}$ | $0.085_{\pm.004}$ |
| Members (0.05) | $0.774_{\pm.005}$ | $0.173_{\pm.011}$ | $0.777_{\pm.005}$ | $0.176_{\pm.021}$ | $0.777_{\pm.005}$ | $0.092_{\pm.011}$ |

Table 13: HPP results (accuracy and fairness violation metrics) on UTK. Models are trained on target variable `gender`, evaluated using protected attribute `age`. Statistics are obtained from five independent runs, and additionally over all individual ensemble members if applicable.

| Before HPP | Acc ($\uparrow$) | SPD ($\downarrow$) | Acc ($\uparrow$) | EOD ($\downarrow$) | Acc ($\uparrow$) | AOD ($\downarrow$) |
|---|---|---|---|---|---|---|
| Members | $0.916_{\pm.002}$ | $0.180_{\pm.005}$ | $0.916_{\pm.002}$ | $0.087_{\pm.007}$ | $0.916_{\pm.002}$ | $0.056_{\pm.003}$ |
| Deep Ensemble | $0.926_{\pm.001}$ | $0.181_{\pm.001}$ | $0.926_{\pm.001}$ | $0.081_{\pm.003}$ | $0.926_{\pm.001}$ | $0.052_{\pm.001}$ |
| After HPP | | HPP-SPD ($\downarrow$) | | HPP-EOD ($\downarrow$) | | HPP-AOD ($\downarrow$) |
| Deep Ens. (val) | $0.925_{\pm.001}$ | $0.161_{\pm.011}$ | $0.925_{\pm.001}$ | $0.060_{\pm.013}$ | $0.925_{\pm.001}$ | $0.051_{\pm.002}$ |
| Deep Ens. (0.05) | $0.920_{\pm.001}$ | $0.117_{\pm.001}$ | $0.923_{\pm.001}$ | $0.037_{\pm.002}$ | $0.925_{\pm.001}$ | $0.051_{\pm.002}$ |
| Members (0.05) | $0.910_{\pm.001}$ | $0.111_{\pm.011}$ | $0.911_{\pm.001}$ | $0.034_{\pm.011}$ | $0.914_{\pm.001}$ | $0.057_{\pm.003}$ |

Table 14: HPP results (accuracy and fairness violation metrics) on UTK. Models are trained on target variable `gender`, evaluated using protected attribute `race`. Statistics are obtained from five independent runs, and additionally over all individual ensemble members if applicable.

| Before HPP | Acc ($\uparrow$) | SPD ($\downarrow$) | Acc ($\uparrow$) | EOD ($\downarrow$) | Acc ($\uparrow$) | AOD ($\downarrow$) |
|---|---|---|---|---|---|---|
| Members | $0.916_{\pm.002}$ | $0.002_{\pm.003}$ | $0.916_{\pm.002}$ | $0.023_{\pm.004}$ | $0.916_{\pm.002}$ | $0.028_{\pm.003}$ |
| Deep Ensemble | $0.926_{\pm.001}$ | $0.002_{\pm.001}$ | $0.926_{\pm.001}$ | $0.022_{\pm.002}$ | $0.926_{\pm.001}$ | $0.029_{\pm.001}$ |
| After HPP | | HPP-SPD ($\downarrow$) | | HPP-EOD ($\downarrow$) | | HPP-AOD ($\downarrow$) |
| Deep Ens. (val) | $0.926_{\pm.001}$ | $0.021_{\pm.003}$ | $0.924_{\pm.001}$ | $0.029_{\pm.006}$ | $0.925_{\pm.001}$ | $0.034_{\pm.003}$ |
| Deep Ens. (0.05) | $0.924_{\pm.001}$ | $0.012_{\pm.001}$ | $0.923_{\pm.002}$ | $0.049_{\pm.007}$ | $0.922_{\pm.002}$ | $0.053_{\pm.005}$ |
| Members (0.05) | $0.914_{\pm.002}$ | $0.006_{\pm.010}$ | $0.914_{\pm.002}$ | $0.035_{\pm.015}$ | $0.914_{\pm.002}$ | $0.039_{\pm.013}$ |

Table 15: HPP results (accuracy and fairness violation metrics) on UTK. Models are trained on target variable `race`, evaluated using protected attribute `age`. Statistics are obtained from five independent runs, and additionally over all individual ensemble members if applicable.

| Before HPP | Acc ($\uparrow$) | SPD ($\downarrow$) | Acc ($\uparrow$) | EOD ($\downarrow$) | Acc ($\uparrow$) | AOD ($\downarrow$) |
|---|---|---|---|---|---|---|
| Members | $0.822_{\pm.006}$ | $0.118_{\pm.009}$ | $0.822_{\pm.006}$ | $0.073_{\pm.016}$ | $0.822_{\pm.006}$ | $0.043_{\pm.007}$ |
| Deep Ensemble | $0.843_{\pm.002}$ | $0.132_{\pm.002}$ | $0.843_{\pm.002}$ | $0.080_{\pm.003}$ | $0.843_{\pm.002}$ | $0.043_{\pm.002}$ |
| After HPP | | HPP-SPD ($\downarrow$) | | HPP-EOD ($\downarrow$) | | HPP-AOD ($\downarrow$) |
| Deep Ens. (val) | $0.857_{\pm.001}$ | $0.131_{\pm.006}$ | $0.858_{\pm.002}$ | $0.066_{\pm.008}$ | $0.858_{\pm.002}$ | $0.042_{\pm.003}$ |
| Deep Ens. (0.05) | $0.856_{\pm.002}$ | $0.149_{\pm.007}$ | $0.858_{\pm.002}$ | $0.086_{\pm.005}$ | $0.857_{\pm.003}$ | $0.057_{\pm.006}$ |
| Members (0.05) | $0.816_{\pm.014}$ | $0.118_{\pm.038}$ | $0.816_{\pm.015}$ | $0.078_{\pm.047}$ | $0.817_{\pm.014}$ | $0.055_{\pm.029}$ |

Table 16: HPP results (accuracy and fairness violation metrics) on UTK. Models are trained on target variable `race`, evaluated using protected attribute `gender`. Statistics are obtained from five independent runs, and additionally over all individual ensemble members if applicable.

| Before HPP | Acc ($\uparrow$) | SPD ($\downarrow$) | Acc ($\uparrow$) | EOD ($\downarrow$) | Acc ($\uparrow$) | AOD ($\downarrow$) |
|---|---|---|---|---|---|---|
| Members | $0.822_{\pm.006}$ | $0.008_{\pm.010}$ | $0.822_{\pm.006}$ | $0.021_{\pm.019}$ | $0.822_{\pm.006}$ | $0.015_{\pm.010}$ |
| Ensemble | $0.843_{\pm.002}$ | $0.011_{\pm.002}$ | $0.843_{\pm.002}$ | $0.023_{\pm.004}$ | $0.843_{\pm.002}$ | $0.013_{\pm.002}$ |
| After HPP | | HPP-SPD ($\downarrow$) | | HPP-EOD ($\downarrow$) | | HPP-AOD ($\downarrow$) |
| Deep Ens. (val) | $0.858_{\pm.003}$ | $0.039_{\pm.002}$ | $0.858_{\pm.001}$ | $0.000_{\pm.006}$ | $0.859_{\pm.003}$ | $0.016_{\pm.008}$ |
| Deep Ens. (0.05) | $0.859_{\pm.002}$ | $0.038_{\pm.013}$ | $0.859_{\pm.002}$ | $0.019_{\pm.019}$ | $0.859_{\pm.002}$ | $0.019_{\pm.006}$ |
| Members (0.05) | $0.816_{\pm.014}$ | $0.009_{\pm.044}$ | $0.816_{\pm.015}$ | $0.002_{\pm.049}$ | $0.816_{\pm.014}$ | $0.030_{\pm.032}$ |

Table 17: HPP results (balanced accuracy and fairness violation metrics) on CX. Models are evaluated using protected attribute `age`. Statistics are obtained from five independent runs, and additionally over all individual ensemble members if applicable.

| Before HPP | BAcc ($\uparrow$) | SPD ($\downarrow$) | BAcc ($\uparrow$) | EOD ($\downarrow$) | BAcc ($\uparrow$) | AOD ($\downarrow$) |
|---|---|---|---|---|---|---|
| Members | $0.783_{\pm.008}$ | $0.138_{\pm.004}$ | $0.783_{\pm.008}$ | $0.174_{\pm.010}$ | $0.783_{\pm.008}$ | $0.101_{\pm.006}$ |
| Deep Ensemble | $0.786_{\pm.004}$ | $0.139_{\pm.001}$ | $0.786_{\pm.004}$ | $0.182_{\pm.004}$ | $0.786_{\pm.004}$ | $0.104_{\pm.002}$ |
| After HPP | HPP-SPD ($\downarrow$) | | HPP-EOD ($\downarrow$) | | HPP-AOD ($\downarrow$) | |
| Deep Ens. (val) | $0.801_{\pm.004}$ | $0.122_{\pm.019}$ | $0.800_{\pm.004}$ | $0.125_{\pm.048}$ | $0.800_{\pm.005}$ | $0.073_{\pm.030}$ |
| Deep Ens. (0.05) | $0.788_{\pm.004}$ | $0.057_{\pm.002}$ | $0.798_{\pm.005}$ | $0.052_{\pm.007}$ | $0.800_{\pm.005}$ | $0.038_{\pm.010}$ |
| Members (0.05) | $0.782_{\pm.010}$ | $0.060_{\pm.005}$ | $0.789_{\pm.010}$ | $0.063_{\pm.015}$ | $0.790_{\pm.011}$ | $0.049_{\pm.009}$ |

Table 18: HPP results (balanced accuracy and fairness violation metrics) on CX. Models are evaluated using protected attribute `gender`. Statistics are obtained from five independent runs, and additionally over all individual ensemble members if applicable.

| Before HPP | BAcc ($\uparrow$) | SPD ($\downarrow$) | BAcc ($\uparrow$) | EOD ($\downarrow$) | BAcc ($\uparrow$) | AOD ($\downarrow$) |
|---|---|---|---|---|---|---|
| Members | $0.783_{\pm.008}$ | $0.000_{\pm.002}$ | $0.783_{\pm.008}$ | $0.024_{\pm.010}$ | $0.783_{\pm.008}$ | $0.014_{\pm.005}$ |
| Deep Ensemble | $0.786_{\pm.004}$ | $0.000_{\pm.000}$ | $0.786_{\pm.004}$ | $0.025_{\pm.002}$ | $0.786_{\pm.004}$ | $0.015_{\pm.001}$ |
| After HPP | HPP-SPD ($\downarrow$) | | HPP-EOD ($\downarrow$) | | HPP-AOD ($\downarrow$) | |
| Deep Ens. (val) | $0.801_{\pm.006}$ | $0.002_{\pm.001}$ | $0.798_{\pm.007}$ | $0.005_{\pm.020}$ | $0.798_{\pm.007}$ | $0.014_{\pm.005}$ |
| Deep Ens. (0.05) | $0.796_{\pm.005}$ | $0.001_{\pm.014}$ | $0.798_{\pm.006}$ | $0.009_{\pm.024}$ | $0.796_{\pm.005}$ | $0.020_{\pm.013}$ |
| Members (0.05) | $0.792_{\pm.012}$ | $0.001_{\pm.013}$ | $0.792_{\pm.012}$ | $0.018_{\pm.027}$ | $0.792_{\pm.012}$ | $0.022_{\pm.013}$ |

Table 19: HPP results (balanced accuracy and fairness violation metrics) on CX. Models are evaluated using protected attribute `race`. Statistics are obtained from five independent runs, and additionally over all individual ensemble members if applicable.

| Before HPP | BAcc ($\uparrow$) | SPD ($\downarrow$) | BAcc ($\uparrow$) | EOD ($\downarrow$) | BAcc ($\uparrow$) | AOD ($\downarrow$) |
|---|---|---|---|---|---|---|
| Members | $0.783_{\pm.008}$ | $0.041_{\pm.002}$ | $0.783_{\pm.008}$ | $0.091_{\pm.008}$ | $0.783_{\pm.008}$ | $0.049_{\pm.004}$ |
| Deep Ensemble | $0.786_{\pm.004}$ | $0.040_{\pm.001}$ | $0.786_{\pm.004}$ | $0.091_{\pm.004}$ | $0.786_{\pm.004}$ | $0.047_{\pm.002}$ |
| After HPP | HPP-SPD ($\downarrow$) | | HPP-EOD ($\downarrow$) | | HPP-AOD ($\downarrow$) | |
| Deep Ens. (val) | $0.801_{\pm.007}$ | $0.037_{\pm.002}$ | $0.802_{\pm.007}$ | $0.083_{\pm.010}$ | $0.802_{\pm.007}$ | $0.044_{\pm.006}$ |
| Deep Ens. (0.05) | $0.802_{\pm.007}$ | $0.039_{\pm.004}$ | $0.802_{\pm.008}$ | $0.078_{\pm.008}$ | $0.799_{\pm.004}$ | $0.053_{\pm.013}$ |
| Members (0.05) | $0.793_{\pm.011}$ | $0.038_{\pm.006}$ | $0.793_{\pm.011}$ | $0.073_{\pm.019}$ | $0.793_{\pm.011}$ | $0.047_{\pm.016}$ |

# F. Additional Investigations

This section presents additional investigations that are complementary to those presented in the main section of the manuscript. First, we analyze the complementary notion of min-max fairness. Second, we investigate how the disparate benefits effect behaves for different model sizes of the individual ensemble members. Third, we conduct the same investigation on different model architectures. Fourth, we study whether the disparate benefits effect also occurs for heterogeneous Deep Ensembles composed of members with different model architectures. Fifth, we report an alternative approach to mitigate the negative impact on fairness due to Deep Ensembling by means of weighting individual members differently in the ensemble. Finally, we study the calibration of the Deep Ensemble and its individual members and the resulting sensitivity of their threshold used to make the prediction.

## F.1. Minimax Fairness

The notions of group fairness discussed throughout the paper (Eq. (2) - (4)) control for the gap between group characteristics such as their PR, TPR or FPR. Another notion often considered in recent work is minimax fairness (Martinez et al., 2020; Diana et al., 2021; Zietlow et al., 2022), where the characteristics of the worst group are of importance. For instance (Zietlow et al., 2022) showed, that the accuracy and TPR of both the minority and majority group decrease when using standard in-processing interventions in facial analysis tasks similar to FF and UTK in our experiments. Therefore, we investigate the minimax fairness impact of Deep Ensembles. Specifically, we discuss the TPR, FPR and accuracy.

The results for TPR and FPR are given in Fig. 12 - 14. We observe, that for none of the considered tasks, there as a significant negative change of the TPR due to ensembling. Similarly, we find that for none of the considered tasks, there is a significant positive change of the FPR due to ensembling, which is desired as a better classifier should have a lower FPR. The results for accuracy are given in Fig. 18 - 20. We find, that the accuracies of both groups significantly increase for all considered tasks. In sum, while we find that Deep Ensembles have a disparate benefits effect, where one group benefits more than the other, thus increases unfairness w.r.t. disparity based group fairness metrics, the predictive performances of both groups increase thus improve fairness under a minimax fairness perspective.

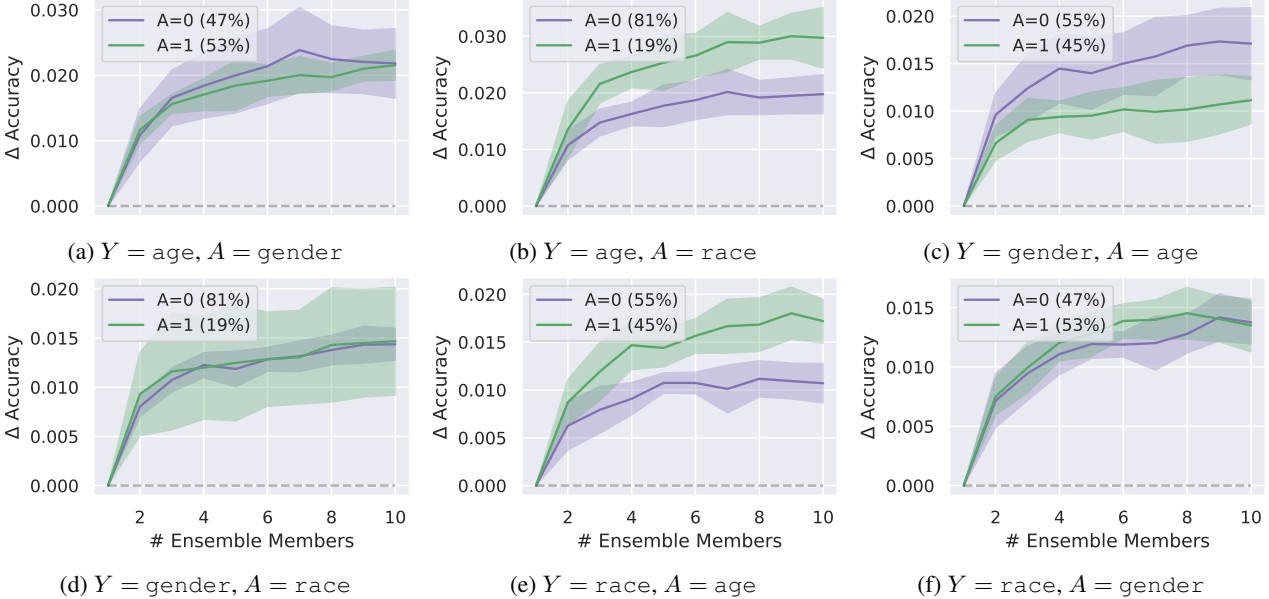

Figure 18: Change in Accuracy for a Deep Ensemble (10 members) on the FF dataset, Statistics are computed based on five independent runs.

## F.2. Model Size

The experiments in the main paper were conducted using ResNet50 models. In this section we investigate whether the size of the models plays a major role in determining the existence and strength of the disparate benefits effect. The results are

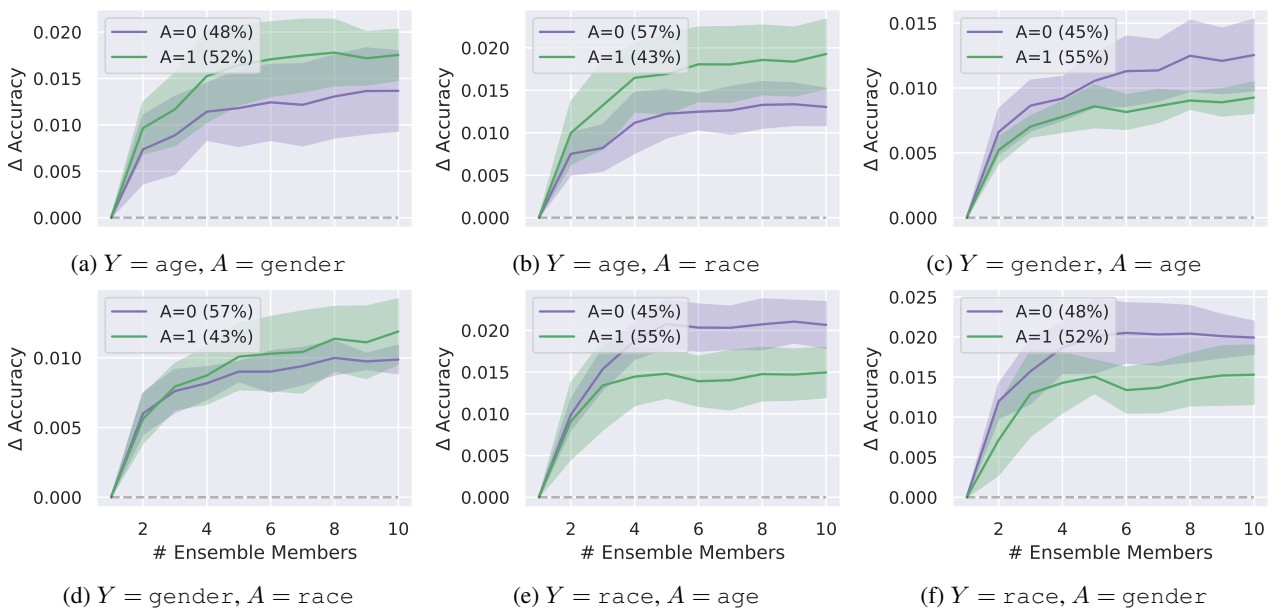

Figure 19: Change in Accuracy for a Deep Ensemble (10 members) on the UTK dataset, Statistics are computed based on five independent runs.

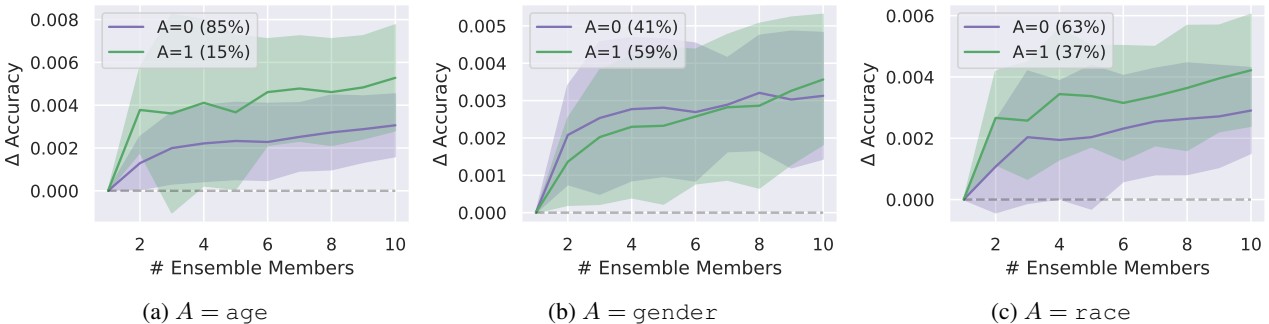

Figure 20: Change in Accuracy for a Deep Ensemble (10 members) on the CX dataset, Statistics are computed based on five independent runs.

shown in Fig. 21 - 23. As seen in the Figures, in the majority of cases the performance gains due to ensembling slightly increase for larger model classes. The fairness violations however increase to a larger degree, see *e.g.* Fig. 21 (a) and (b), Fig. 22 (a), (b) and (c) as well as Fig. 23 (a). Generally, we observe an increase in the magnitude of the change in fairness violations with larger model classes for all tasks that exhibit significant disparate benefits (*cf.* Tab. 1).

### F.3. Model Architecture

In this section we investigate the role of the specific model architecture on the existence and strength of the disparate benefits effect. The results are shown in Fig. 24 - 26. In the majority of cases, disparate benefits occur throughout all considered model architectures. Especially for EfficientNetV2-S we observe significant disparate benefits for some cases where we do not observe them in the main investigation based on ResNet50. For example for UTK, target race, group age under AOD (Fig. 25e) or CX, group race under EOD and AOD. Overall, we do not find a systematic difference of the results for different model architectures.

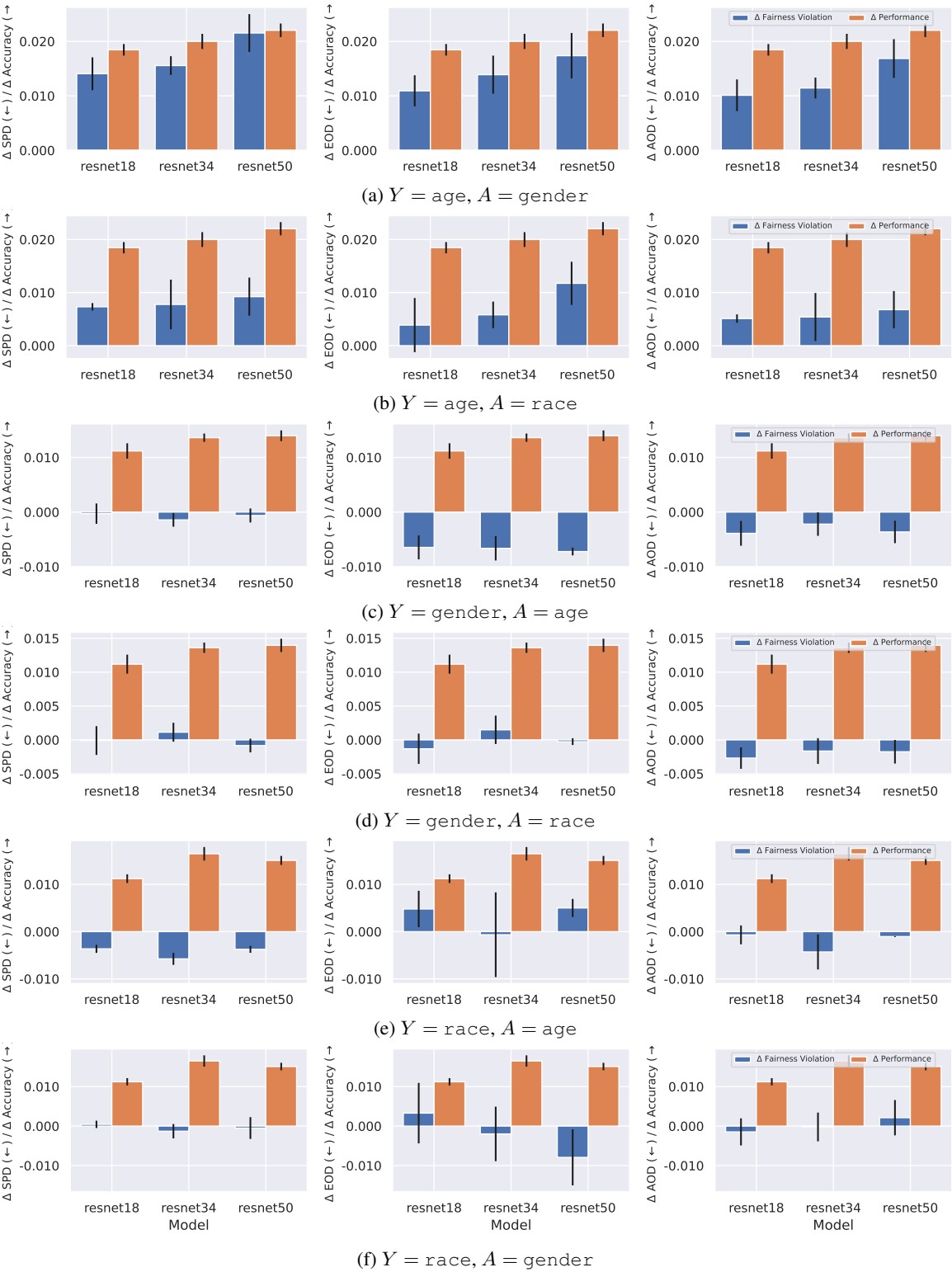

Figure 21: The disparate benefits effect of Deep Ensembles for different model sizes. Models are trained and evaluated on the FF dataset. Statistics are computed based on five independent runs.

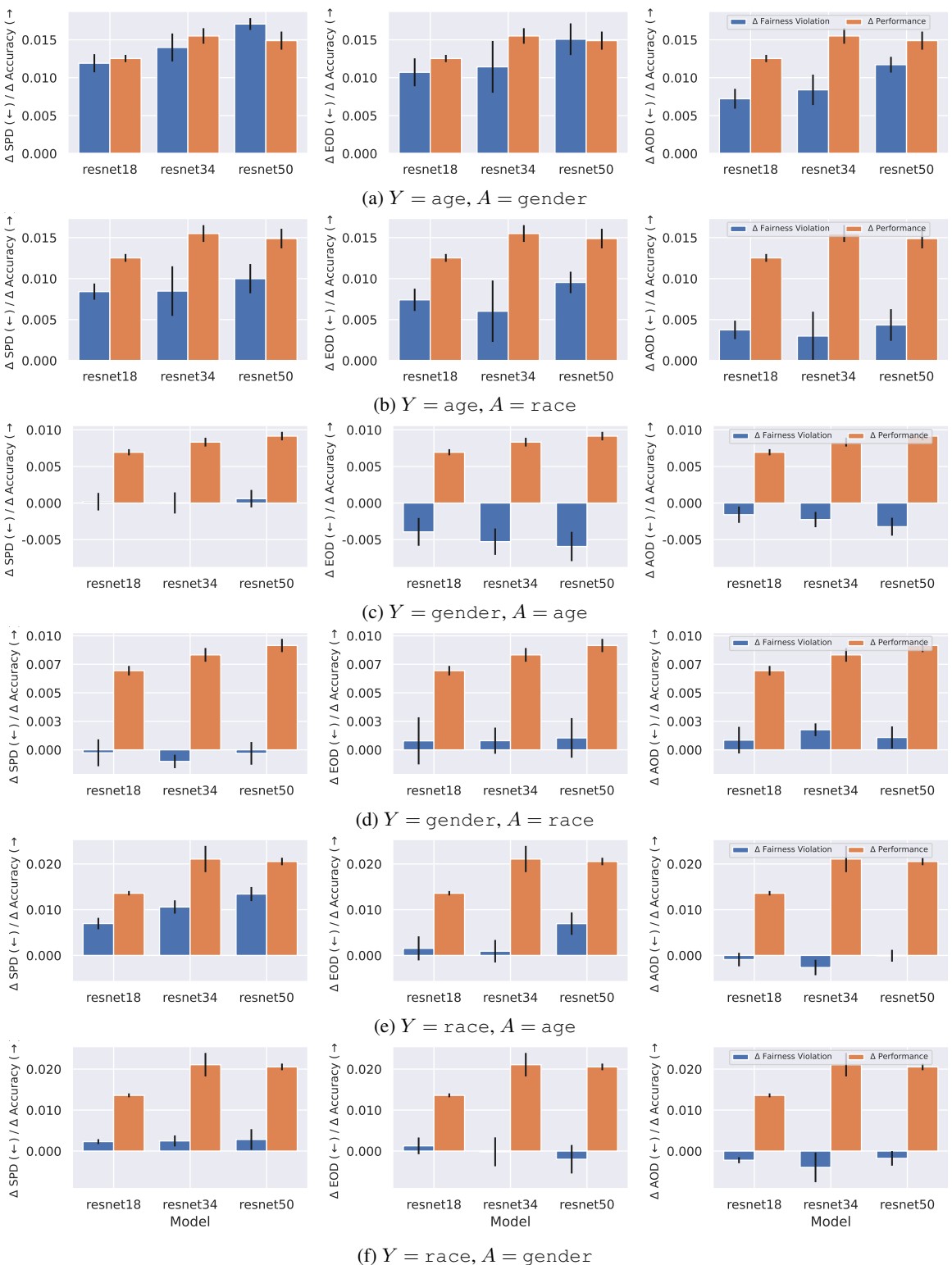

Figure 22: The disparate benefits effect of Deep Ensembles for different model sizes. Models are trained and evaluated on the UTK dataset. Statistics are computed based on five independent runs.

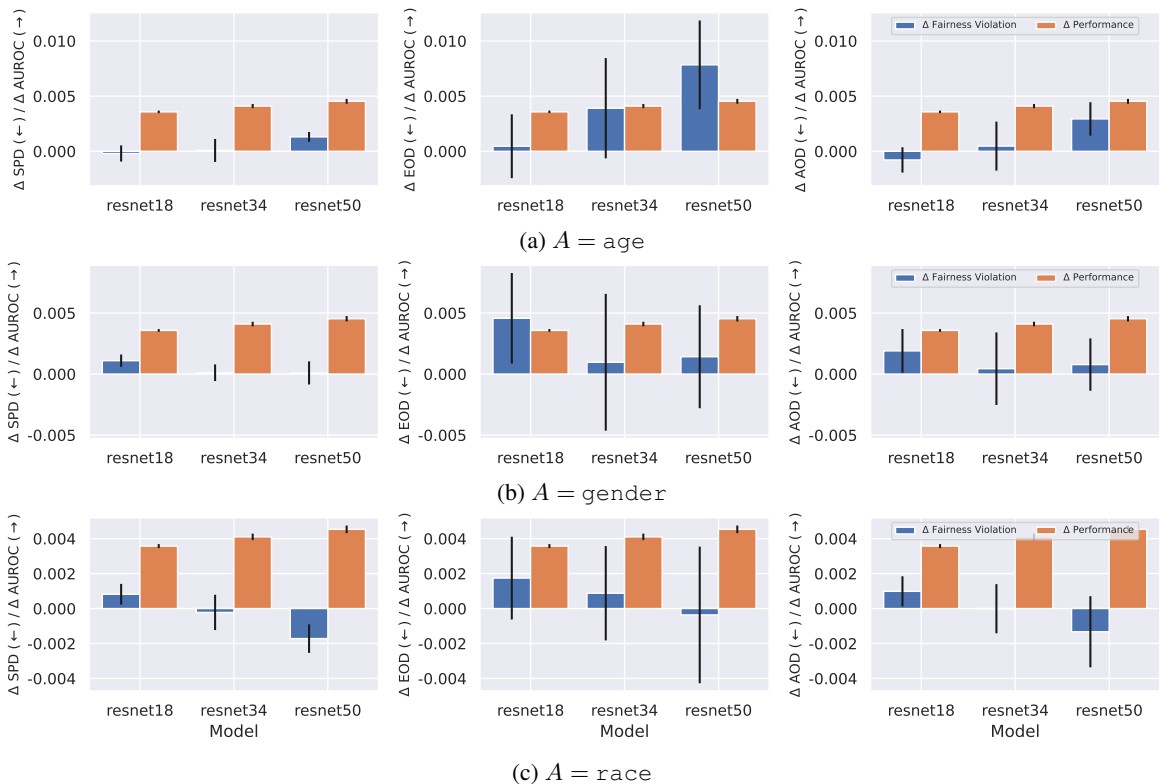

Figure 23: The disparate benefits effect of Deep Ensembles for different model sizes. Models are trained and evaluated on the CX dataset. Statistics are computed based on five independent runs.

### F.4. Heterogenous Ensembles

The results presented in Fig. 1 in the main paper are obtained from a homogeneous Deep Ensemble composed of ResNet50 models. The results presented in Fig. 27 consider the same target / protected group combinations for the same datasets using a heterogeneous Deep Ensemble of ResNet18/34/50 models. We observe the disparate benefits effect for heterogeneous ensembling to a similar extent than for homogeneous ensembling.

### F.5. Deep Ensemble Weighting

In this section, we study whether there exist weightings to combine the individual models in the Deep Ensemble that perform better than a standard uniform averaging as in Eq. (1). The approximation in Eq. (1) thus changes to

$$p_{\boldsymbol{\lambda}}(y \mid \boldsymbol{x}, \mathcal{D}) \approx \sum_{n=1}^{N} \lambda_n \, p(y \mid \boldsymbol{x}, \boldsymbol{w}_n) \,. \tag{16}$$

$\boldsymbol{\lambda}$ satisfies $\sum_{n=1}^{N} \lambda_n = 1$ and $\lambda_n \geq 0 \; \forall n$. Note that Eq. (16) results in Eq. (1) if $\lambda_n = 1/N \; \forall n$. We consider $\boldsymbol{\lambda} \sim \text{Dir}(\alpha_1, ..., \alpha_N)$ with $\alpha_n = 1 \; \forall n$. Weightings are thus drawn uniformly at random from a $N - 1$ dimensional probability simplex. In our empirical investigation, we sampled 2,000 weightings $\boldsymbol{\lambda}$ and evaluated the resulting ensembles on the three tasks. The results are given in Fig. 28, showing individual members and the different resulting ensembles, as well as their convex hull. In the case of the FF and UTK datasets, there apprears to be a strong correlation between fairness violations and performance, and the weights hardly provide more Pareto optimal models. However, regarding the CX dataset, we observe that there are many weightings that would yield a more favorable outcome than uniform averaging as generally done by Deep Ensembles. In the following, we outline two methods to choose such a weighting. However, both methods did no lead to a significantly better outcome than uniform averaging. Nevertheless, we include a qualitative discription of our experiments as guidance for future research.

**Weight selection based on the validation set.** The simplest approach to identify a more favorable set of weights consists of

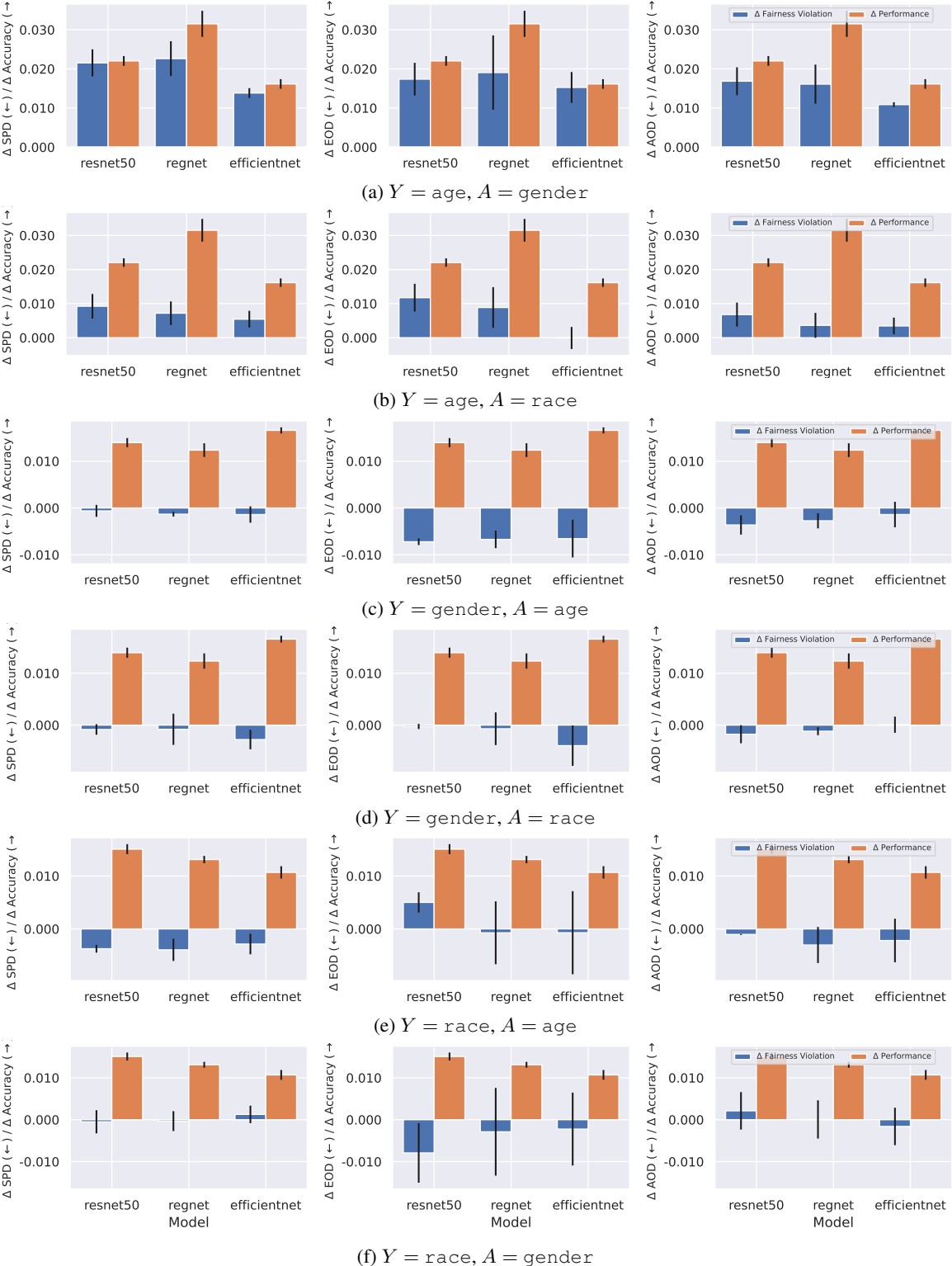

(a) $Y = $ age, $A = $ gender

(b) $Y = $ age, $A = $ race

(c) $Y = $ gender, $A = $ age

(d) $Y = $ gender, $A = $ race

(e) $Y = $ race, $A = $ age

(f) $Y = $ race, $A = $ gender

Figure 24: The disparate benefits effect of Deep Ensembles for different model architectures. Models are trained and evaluated on the FF dataset. Statistics are computed based on five independent runs.

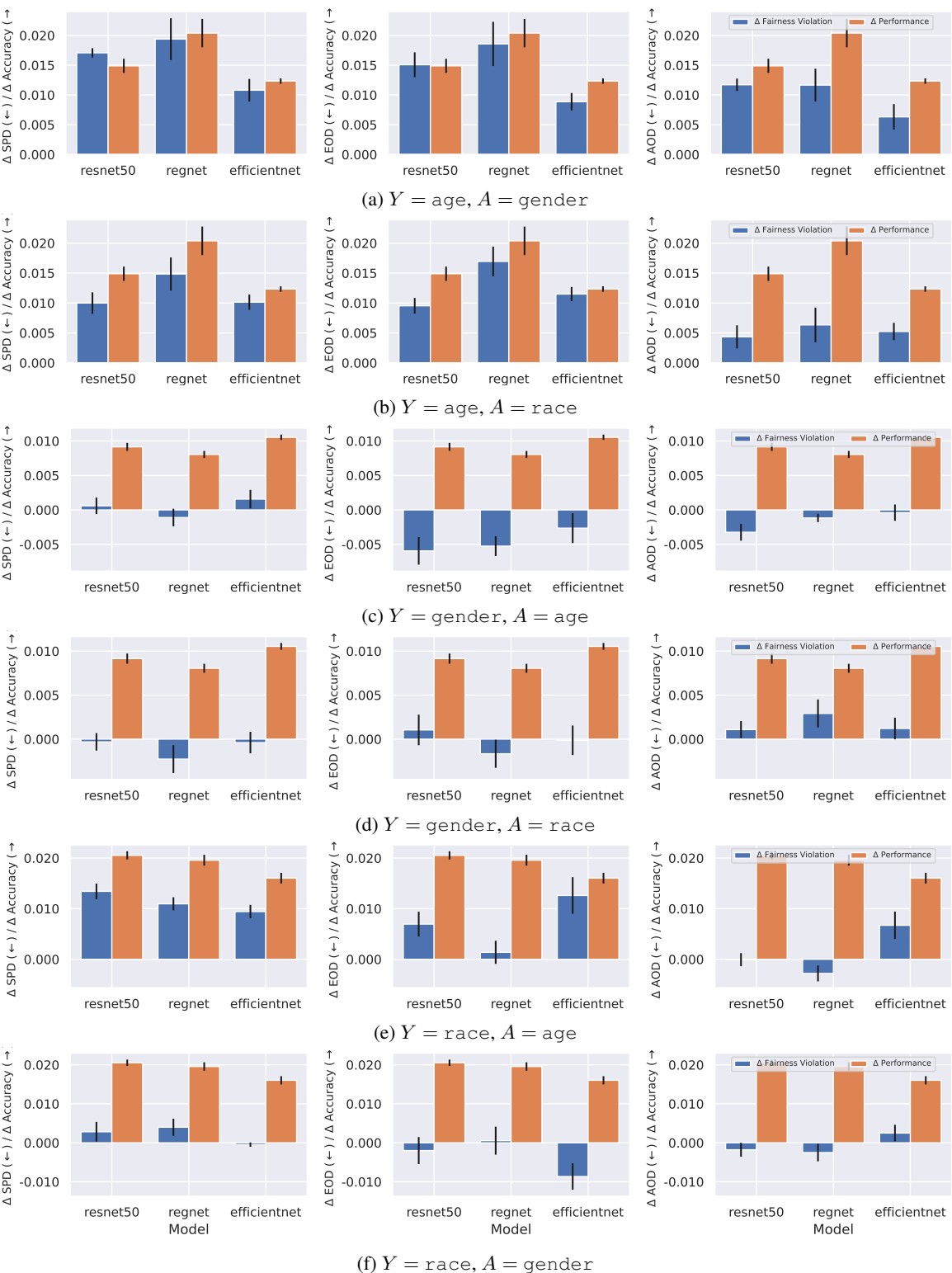

Figure 25: The disparate benefits effect of Deep Ensembles for different model architectures. Models are trained and evaluated on the UTK dataset. Statistics are computed based on five independent runs.

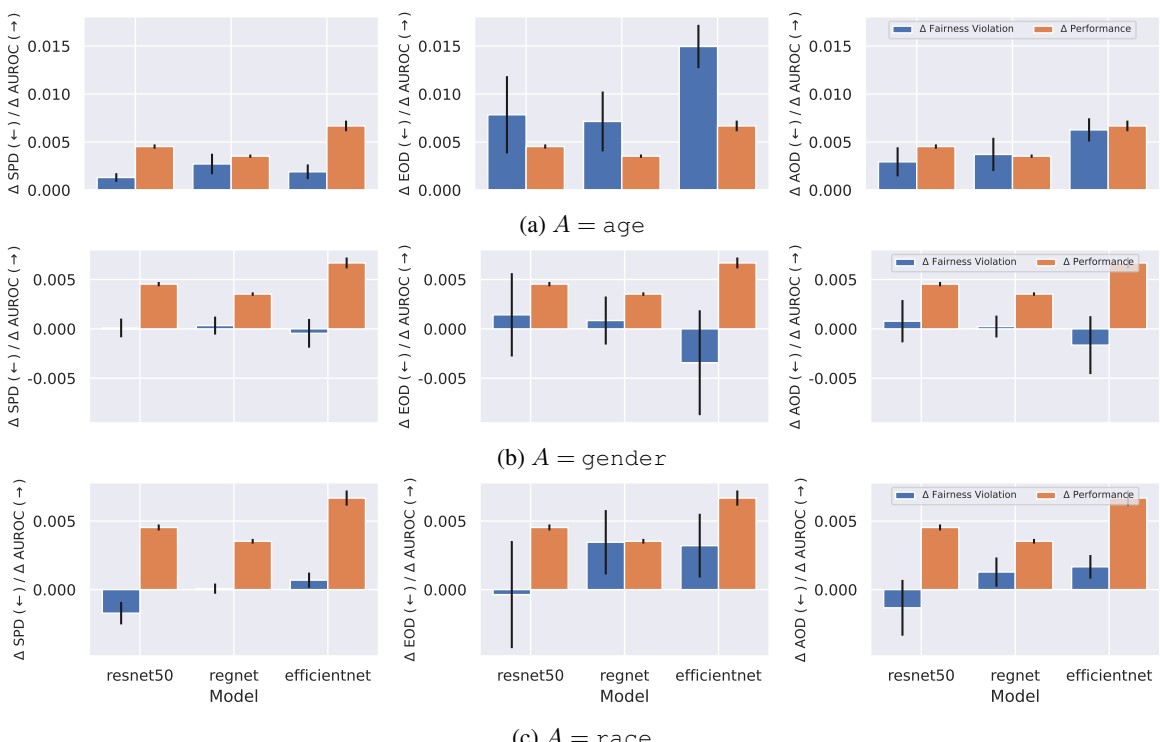

(a) $A = \texttt{age}$

(b) $A = \texttt{gender}$

(c) $A = \texttt{race}$

Figure 26: The disparate benefits effect of Deep Ensembles for different model architectures. Models are trained and evaluated on the CX dataset. Statistics are computed based on five independent runs.

selecting it as a hyperparameter. In our experiments, we sampled $\lambda$ uniformly at random as described before and selected the Pareto optimal weighting on the validation set. However, we found that the selected weights did not improve performance on the test dataset, neither for the UTK dataset - where it could expected due to the distribution shift - nor on the FF and CX datasets, where the validation and test datasets are drawn from the same distribution. Notably, the selected solutions were very close to the uniform weighting that is usually used in Deep Ensembles.

**Fairness-based weighting.** Furthermore, we leveraged the information about the fairness violation of the individual members to define the weights and yield a fairer ensembling. This is similar to the approach reported in Kenfack et al. (2021), yet for neural networks as base models. Given a fairness violation measure $F_n \in [0,1]$ for each ensemble member, we define the weighting factor

$$\lambda_n = \frac{\exp\{-F_n/\tau\}}{\sum_{j=1}^{N} \exp\{-F_j/\tau\}} , \tag{17}$$

for Eq. (16), where $\tau \in \mathbb{R}_+$ is a temperature hyperparameter. For high values of the temperature parameter $\tau \to \infty$, Eq. (16) becomes equivalent to Eq. (1). For low values of the temperature parameter $\tau \to 0$, the fairness-weighted predictive distribution given by Eq. (16) approaches the predictive distribution of the model with lowest fairness violation. We calculated the fairness measure on an additional held out "fairness" dataset. The temperature parameter was selected on the validation dataset. In our experiments, the proposed fairness-weighted Deep Ensemble was not significantly Pareto dominant to the uniform weighting. Notably, the selected solutions were either close to the individual models or to uniform averaging, thus exhibiting extremely high variance. In further analysis, we found that performance and fairness violations are extremely dependent on the selected temperature, both being non-smooth functions of the temperature. On the considered datasets and models, the best temperatures were usually found around 1e-2.

### F.6. Calibration and Threshold Selection

As elaborated in the main part of the paper, we find that the Deep Ensemble is better calibrated than individual members (Fig. 7a). Here we provide a more detailed analysis that looks into the decrease in ECE per protected group for each target / protected group attribute pair (task) we consider throughout our experiments. The results are provided in Fig. 29, showing

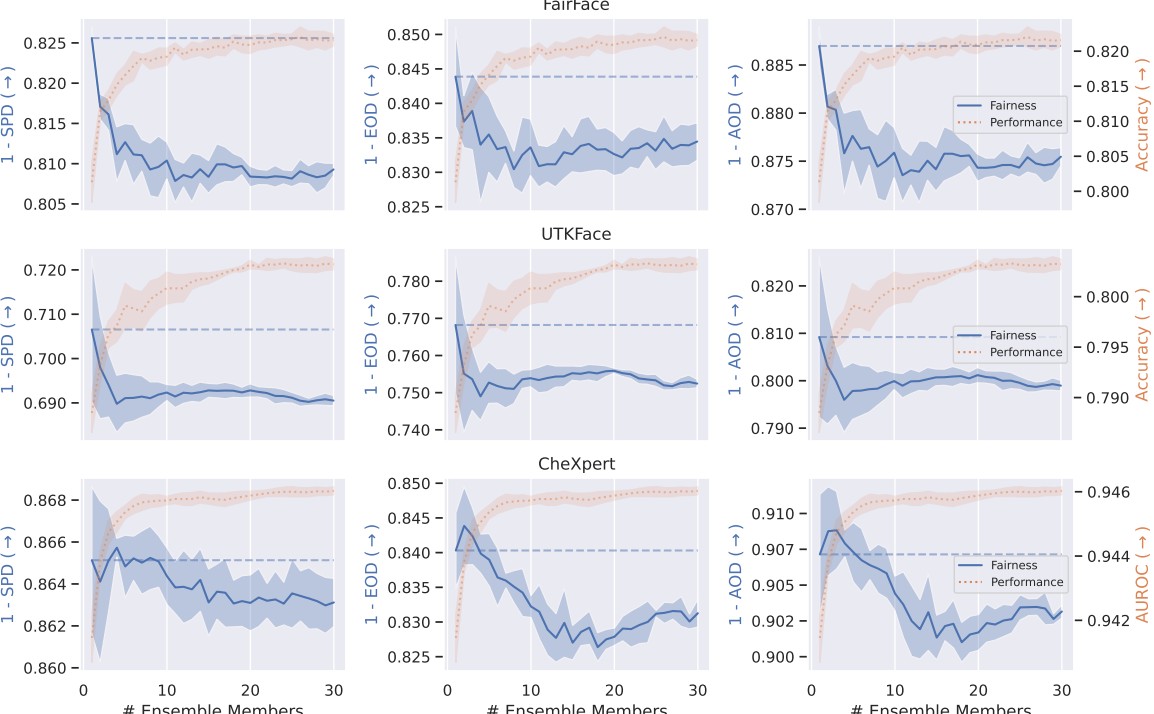

Figure 27: The dangers of the *disparate benefits* effect for heterogeneous (ResNet18/34/50) Deep Ensembles. The performance increases, but the fairness decreases when adding members to the ensemble. The models evaluated on the FairFace test dataset and UTKFace dataset are trained to predict `age` as the target variable and are evaluated using `gender` (male / female) as the protected attribute to define the groups. CheXpert models are trained to predict whether there was a finding regarding a set of medical conditions or not and are evaluated using `age` (young / old) as the protected attribute to define the groups. Statistics are obtained from five independent runs.

that for some tasks, the ECE significantly differs per group, but the Deep Ensemble is more calibrated than individual members, regardless of the protected group attribute.

Finally, we report the results of analyzing the dependency of the Deep Ensemble and individual ensemble members on selecting the threshold for prediction. When using the usual $\mathrm{argmax}$, implicitly a threshold of 0.5 is used. In the post-processing experiments we found that applying the method even under an additional fairness constraint can improve the performance. We evaluated all trained models on their respective validation datasets. Results are depicted in Fig. 30. The results show that the Deep Ensemble is more sensitive to the threshold on the FF dataset, especially for target variable `age`. Regarding the CX dataset, the balanced accuracy exhibits roughly the same behavior under varying thresholds for the Deep Ensemble than for individual members. However, the spread of the optimal threshold is much smaller throughout all experiments.

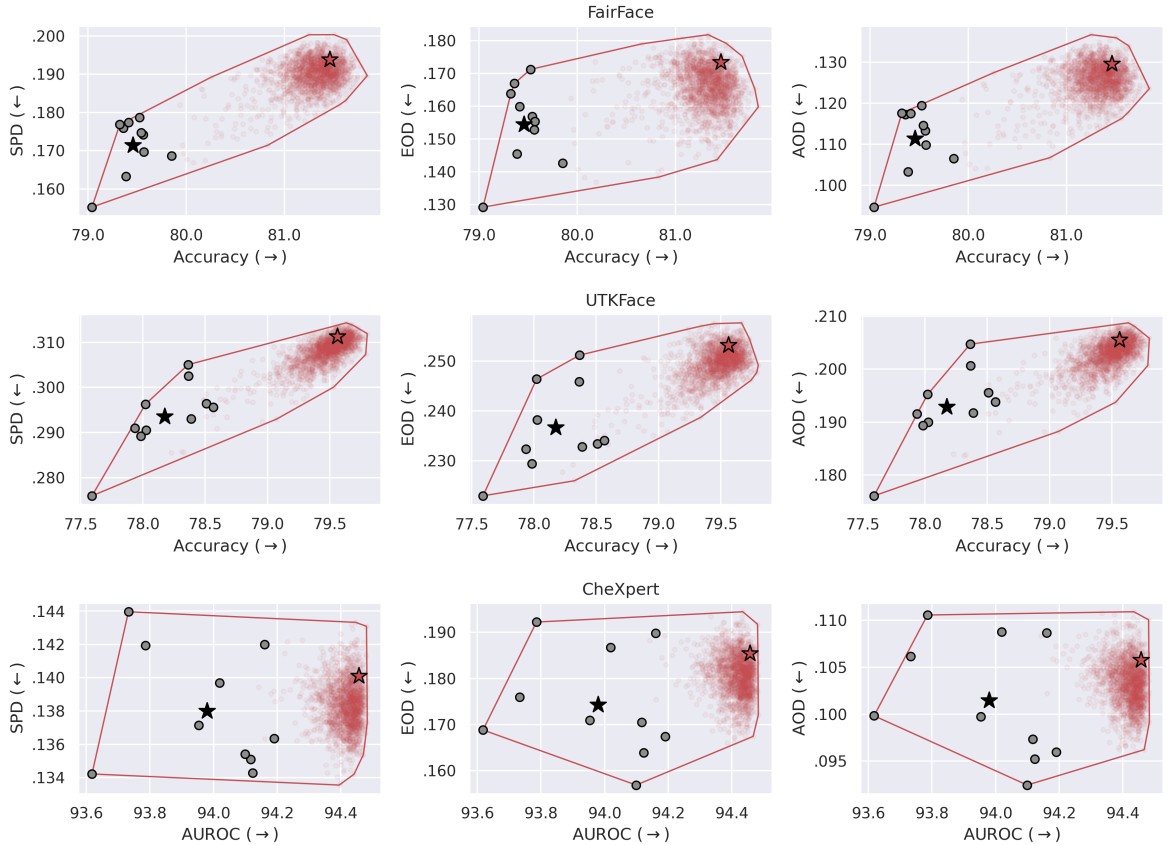

Figure 28: Convex hull of performance and fairness violations for possible weightings to aggregate members of the Deep Ensemble. Ensemble weights are drawn uniformly at random from a $N - 1$ dimensional simplex. Grey points represent individual models, the black star corresponds to their average performance and fairness violation. The red star represents the standard Deep Ensemble with uniform weighting.

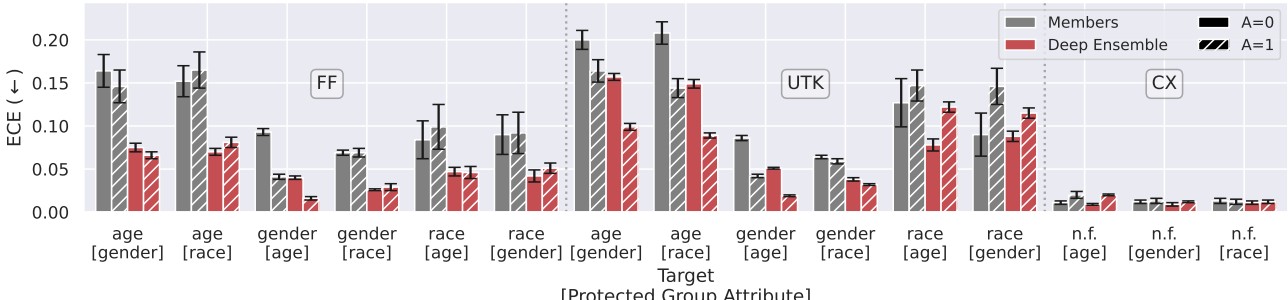

Figure 29: Expected Calibration Error (ECE) per group (group denoted by the hatches) for individual ensemble members and the Deep Ensemble for all considered target protected attribute combinations. Statistics are computed based on five independent runs.

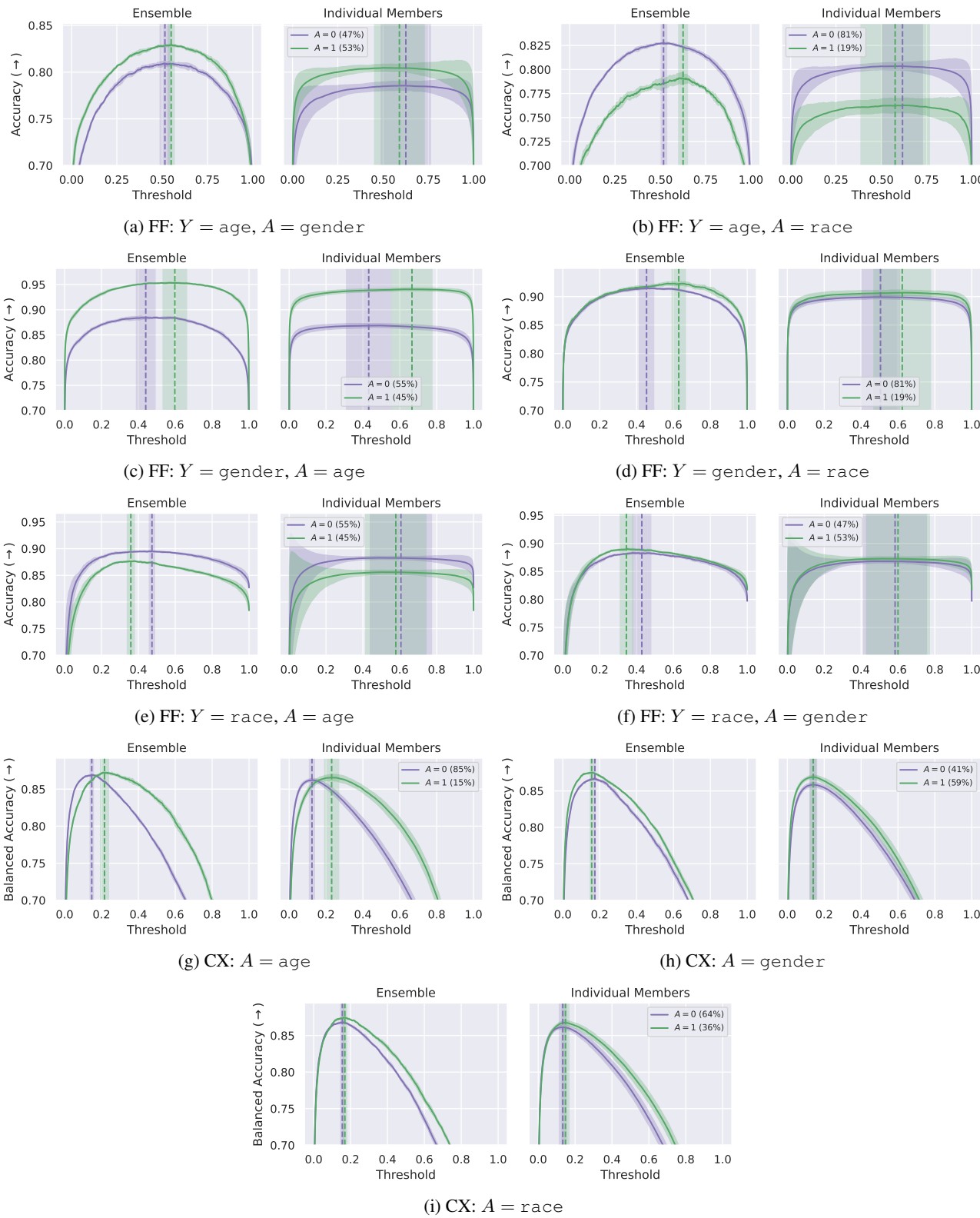

Figure 30: (Balanced) Accuracy depending on the chosen threshold for the FF and CX validation datasets. Vertical lines and shading denote optimal threshold per protected group. Statistics are computed based on five independent runs.

