# OpenReview forum: "The Disparate Benefits of Deep Ensembles"
_ICML.cc/2025/Conference — ICML 2025 poster_

### Official Review · Reviewer_mL9u · 2025-03-10

**Overall Recommendation:** 1

**Summary:**

This paper presents an empirical study of the fairness properties of deep ensembles. On several image datasets, the authors explore the fairness/accuracy tradeoff of deep ensembles at varying numbers of members, and explore a hypothesis which relates decreasing fairness metrics with varying amounts of predictive diversity between groups. They demonstrate that common post-processing methods can be helpful for improving the fairness of the ensemble.


###########
UPDATE AFTER REBUTTAL:
Thanks for the rebuttal. I'm not totally sure what to do with this paper as I still have some reservations about the central argument and I see I'm the main holdout here. I'm also not totally sure what to do with the extra table in the last comment - I'm looking at the left column which corresponds to the distribution over "Y=0,A=0 / Y=0,A=1 / Y=1,A=0 / Y=1,A=1" - however it seems like the base rates are equal between groups at all values in the left hand column: e.g.  "0.22 / 0.22 / 0.28 / 0.28" yields a 0.28/0.22 ratio in both group A=0 and 1. I tried flipping around the A and Y values to make sense but I don't quite see where the base rates variation comes in.

I'm going to keep my score as is but acknowledge that I may just be missing something here given the scores of the other reviewers - apologies if I didn't understand something obvious about the paper.

**Claims And Evidence:**

The central thesis of this paper is that deep ensembles "unevenly favor different groups", which the authors call the disparate benefits effect. The key evidence relies on demonstrating that as models are added to an ensemble, we see accuracy rise and fairness get worse.

I am not sure I am convinced of this thesis by the arguments in this paper. In particular, I don't think the authors dispel the possibility that this is simply a property of improving model (read: ensemble) accuracy. We frequently observe such a fairness/accuracy tradeoff across many types of models, in particular if group base rates differ (in this case, we must observe such a tradeoff for a metric like SPD).

Various pieces of evidence are proposed, which I do not think necessarily demonstrate the argued conclusion:
- argument around average predictive diversity: the argument here is that if diversity differs between groups then the benefits of ensembling would mostly accrue to that group. However, this seems to lead more naturally to an argument about subgroup accuracy than equalized odds type metrics, and certainly wouldn't apply to the SPD pattern seen here
- Fig 4: this synthetic example does not necessarily demonstrate the desired pattern - it just constructs a case where there is differing DIV and disparate benefits. To show the pattern, one would need to also construct the counterfactual faces where there is equal DIV and equal benefits. Current data is consistent with the hypothesis  that there is nothing special about ensembles e.g. difference in base rates is driving the disparity
- Fig 3 does demonstrate the desired pattern, however it is just observed for a single model and I would want to see this pattern shown more systematically to support the claim
- The fact that Hardt post-processing is successful is also concordant with the view that ensembles are just like any other classifier, and can have their fairness properties improved with smart post-processing. The point about thresholding in 5b is interesting but I'm not sure it distinguishes ensemble fairness that clearly; in fact, Fig 6 itself shows that post processing applies to individual models helps just as much as applied to the ensemble!

**Essential References Not Discussed:**

n/a

**Experimental Designs Or Analyses:**

see first section

**Methods And Evaluation Criteria:**

I would be interested to know what the underlying base rates are on each dataset

I think the OOD-style experiment using UTKFace is an interesting idea, enjoyed seeing that included

**Other Comments Or Suggestions:**

Fig 6: it's not totally clear to me why there are 2 red dots downstream of the starting point on each plot

**Other Strengths And Weaknesses:**

As stated previously, I find the overall argument of the paper to not be particularly convincing. However, I do think the direction here is an interesting place to explore, and I found the overall clarity of the writing to be pretty good.

**Questions For Authors:**

I would like to see evidence which tells me specifically how the fairness properties of ensembles behave differently from the fairness properties of individual models - e.g. in the DIV hypothesis, a more fully fleshed out series of observations along the lines of what's in Fig 3, or a clearer case where you construct a counterfactual example synthetically with/without high DIV that results in different levels of benefit disparity

**Relation To Broader Scientific Literature:**

connection to fairness + ensemble literature is fine

**Theoretical Claims:**

n/a

---

> ### Author Rebuttal · Authors · 2025-03-27
>
> We thank the reviewer for their critical assessment of our work and the insightful questions. We first address the main question, followed by responses to the specific concerns outlined in the review.
>
> ## Main Question
> We appreciate the request for clearer evidence linking predictive diversity to the disparate benefits effect. As you correctly noted, with real-world datasets we cannot directly control predictive diversity. This limits us in Fig 3 to comparing tasks where the disparate benefits effect is observed (top row) versus not observed (bottom row) as per the results in Tab 1.
>
> However, believe the experiment on synthetic data reported in Apx F.1. aligns with your suggestion. There, we systematically varied the level of predictive diversity using a parameter $\alpha$, while keeping all other factors constant. As shown in Fig 16 and Tab 19, the disparate benefits effect increases with higher diversity and vanishes entirely when $\alpha = 0$. Importantly, this setup uses equal base rates ($p(Y=y, A=a) = 0.25$ for all combinations), ruling out base rate differences as the driver of the observed disparities.
> Based on your feedback, we will move this experiment into the main paper in the camera-ready version to strengthen our core argument.
>
> ## Comments
> **Base rates**
>
>  We provide the base rates in the legends of Fig 3 (see Fig 13-15 for all tasks), as well as Fig 4c for the synthetic experiment. Notably, there are some tasks (e.g. FF Y=age, A=gender) with very balanced base rates exhibiting significant levels of disparate benefits (see Tab 1). Also, we designed the synthetic task with equal base rates to eliminate this possible confounding factor.
>
> **Fairness/accuracy tradeoff**
>
> We appreciate the concern about a potential fairness/accuracy tradeoff. However, we do not believe the disparate benefits effect merely reflects such a tradeoff. In some tasks (Fig 7 and 8), both accuracy and fairness improve through ensembling. Moreover, for tasks where fairness declines, we find that applying Hardt Post-processing (HPP) to the Deep Ensemble (DE) restores fairness to the level of individual models, without sacrificing the ensemble’s accuracy gains (Fig 6, red dot directly above the dotted line). This suggests the effect is not intrinsically an accuracy/fairness tradeoff.
>
> **Single model**
>
> Thank you very much for your observation. We inspected the average predictive diversity per target and group for all 5 model variants and obtained similar results to those reported in Fig 3. Based on your feedback, we will add a new set of Figures in the revised version of the paper, summarizing these results.
>
> **Hardt post-processing**
>
> We agree to the point that DEs are not different from any other probabilistic classifier regarding HPP. In essence, HPP takes any predictive distribution and group attribute as input independent of how this distribution was obtained. However, we believe that our finding regarding the improved calibration of the ensemble and thus its heightened sensitivity to the threshold is an interesting and novel finding.
>
> **Fig 6**
>
> We thank the reviewer for pointing out the lack of clarity regarding the two red dots in Fig 6, which correspond to different levels of fairness tolerance to be achieved by the post-processing of the DE.
>
> * The first level of fairness tolerance is indicated by the dotted line and corresponds to the average fairness of individual ensemble members (gray dot next to the dotted line). The red dot closest to the dotted line corresponds to applying HPP to the DE to achieve the same level of fairness as the individual members (dotted line). Note how this is achieved while increasing accuracy. Thus, it illustrates that the DE remains Pareto-dominant by increasing both fairness and accuracy.
> * The second level of fairness tolerance is marked by the dashed line and it corresponds to 0.05, which is a commonly used value in the literature. In this case, the DE is also Pareto-dominant when compared to the individual models after post-processing both to the same tolerance level.
>
> We acknowledge that Fig 6 contains a lot of information and will revise its caption and description for clarity.
>
>
> We believe these revisions and clarifications directly address the reviewer’s central concerns—particularly regarding predictive diversity, base rate confounding, and the distinct fairness behavior of ensembles. We hope this supports a more favorable reassessment of our work.

---

> > ### Comment · Reviewer_mL9u · 2025-04-04
> >
> > Thanks for this rebuttal - a couple responses below:
> >
> > On the main question:
> > - I think that some of the pieces of evidence you point to here are useful (Appendix F1, Fig 7/8 in Appendix E) are helpful to make the argument that what we're seeing goes beyond standard fairness/accuracy tradeoffs, but I think that this argument should be a much more fundamental piece of the paper and that relevant contrasts should be made much more centrally rather than in the Appendix. For example, while noting that equal base rates in Appendix F.1 does not remove this effect is helpful, I think that the overall argument requires a more systematic examination of how the effect changes wrt to base rates
> >
> > Thanks for the other clarifications. I do find the calibration threshold finding to be interesting. I think that my asks are fairly substantial and so this still won't be an accept from me - however, I'll consider changing my score in discussion with other reviewers.

---

> > > ### Author Response · Authors · 2025-04-07
> > >
> > > Thank you very much for your response and engaging in discussion!
> > >
> > > We fully agree that the experiment in Appendix F.1 is central to the paper. It was the last section we moved to the appendix to fit within the 8-page limit for the submission. Upon acceptance, we will use the extra page to bring this experiment back into the main paper, as we believe it strengthens the core argument.
> > >
> > >
> > > Regarding your point about base rates, we’re happy to share an additional piece of evidence. We repeated the experiment from Appendix F.1 while systematically varying the degree of imbalance in the base rates. The table below reports results for several configurations, with base rates shown for the combinations Y=0,A=0 / Y=0,A=1 / Y=1,A=0 / Y=1,A=1.
> > > Despite these variations, the effect of predictive diversity ($\alpha$) on disparate benefits remains consistent, while the changes in base rates appear to have little to no effect on the fairness metrics. This suggests that the disparate benefits effect is not simply a function of group base rate imbalance.
> > > We will include this analysis in the final version of the paper to further reinforce our argument.
> > >
> > >
> > > | Base Rates                | Setting       | Δ Accuracy (↑)     | Δ SPD (↓)         | Δ EOD (↓)         | Δ AOD (↓)         |
> > > |--------------------------|---------------|---------------------|--------------------|--------------------|--------------------|
> > > | 0.25 / 0.25 / 0.25 / 0.25 | α = 0.0       | 0.005±0.001         | 0.004±0.005        | -0.004±0.007       | -0.001±0.003       |
> > > | 0.22 / 0.22 / 0.28 / 0.28 | α = 0.0       | 0.005±0.002         | 0.004±0.002        | 0.006±0.003        | 0.001±0.003        |
> > > | 0.19 / 0.19 / 0.31 / 0.31 | α = 0.0       | 0.005±0.001         | 0.000±0.006        | 0.003±0.003        | -0.001±0.004       |
> > > | 0.14 / 0.14 / 0.36 / 0.36 | α = 0.0       | 0.005±0.002         | 0.004±0.002        | 0.006±0.003        | 0.001±0.003        |
> > > | 0.25 / 0.25 / 0.25 / 0.25 | α = 0.2       | 0.017±0.003         | 0.010±0.006        | 0.014±0.011        | 0.005±0.011        |
> > > | 0.22 / 0.22 / 0.28 / 0.28 | α = 0.2       | 0.014±0.002         | 0.003±0.004        | 0.005±0.007        | 0.003±0.008        |
> > > | 0.19 / 0.19 / 0.31 / 0.31 | α = 0.2       | 0.013±0.003         | -0.003±0.002       | 0.007±0.004        | 0.010±0.003        |
> > > | 0.14 / 0.14 / 0.36 / 0.36 | α = 0.2       | 0.014±0.002         | 0.003±0.004        | 0.005±0.007        | 0.003±0.008        |
> > > | 0.25 / 0.25 / 0.25 / 0.25 | α = 0.4       | 0.025±0.002         | 0.015±0.009        | 0.037±0.011        | 0.024±0.006        |
> > > | 0.22 / 0.22 / 0.28 / 0.28 | α = 0.4       | 0.021±0.002         | 0.012±0.003        | 0.040±0.008        | 0.029±0.005        |
> > > | 0.19 / 0.19 / 0.31 / 0.31 | α = 0.4       | 0.023±0.002         | 0.007±0.007        | 0.037±0.009        | 0.031±0.004        |
> > > | 0.14 / 0.14 / 0.36 / 0.36 | α = 0.4       | 0.021±0.002         | 0.012±0.003        | 0.040±0.008        | 0.029±0.005        |
> > > | 0.25 / 0.25 / 0.25 / 0.25 | α = 1.0       | 0.030±0.002         | 0.017±0.009        | 0.055±0.012        | 0.040±0.004        |
> > > | 0.22 / 0.22 / 0.28 / 0.28 | α = 1.0       | 0.030±0.001         | 0.016±0.004        | 0.061±0.005        | 0.046±0.006        |
> > > | 0.19 / 0.19 / 0.31 / 0.31 | α = 1.0       | 0.029±0.002         | 0.010±0.007        | 0.053±0.012        | 0.043±0.005        |
> > > | 0.14 / 0.14 / 0.36 / 0.36 | α = 1.0       | 0.030±0.001         | 0.016±0.004        | 0.061±0.005        | 0.046±0.006        |
> > >
> > > We believe this additional analysis directly addresses the reviewer’s concern by demonstrating that changes in base rates do not account for the observed disparities. The results consistently point to predictive diversity as the primary factor driving the disparate benefits effect.

---

### Official Review · Reviewer_rDWq · 2025-03-11

**Overall Recommendation:** 2

**Summary:**

This paper studies the fairness of deep ensembles on three image datasets. They find that ensembling tends to reinforce unfair behavior of ensemble members. The authors provide an explanation in that the predictive diversity causes this effect, which they call disparate benefits effect and give empirical evidence. Finally, the authors suggest some strategies to mitigate the effect.

## Update after rebuttal

I found the arguments risen by reviewer mL9u very reasonable and compelling. I hadn't seen it from that viewpoint. I still think that the paper makes interesting points, but will adjust my scoring below the acceptance threshold.

**Claims And Evidence:**

The paper claims that ensembling neural networks tends to have a negative impact on fairness. A well-designed and described set of experiments on three datasets sufficiently supports the claim. They further claim that this effect is caused by predictive diversity, for which they do NOT provide evidence, since they only show that the two phenomena have a significant correlation in the studied data (but both could have a common cause and be independent given that cause).

**Essential References Not Discussed:**

None I am aware of.

**Experimental Designs Or Analyses:**

I already commented on this above.

**Methods And Evaluation Criteria:**

Several evaluations are conducted throughout the study, all of them properly designed and executed except maybe the sample size of 5 and the statistical significance test using a t-test. The evaluation criteria are reasonable to me.

**Other Comments Or Suggestions:**

- I think that while Table 1 is interesting, it would be important to have a scatter plot with the absolute violations of a single member (x-axis) and the ensemble (y-axis).
- The paper coins the term "disparate benefits effect" but fails to properly introduce and maybe formalize it. I searched with Ctrl + F and found that curiously the abstract (not the introduction) gives the best intuition.
- Also, while the results show a consistent tendency to reduce fairness by ensembling, the effect is not very pronounced. Statistical significance doesn't mean that it is particularly relevant. I mean even the highest difference is only 0.022, not particularly high.

**Other Strengths And Weaknesses:**

None

**Questions For Authors:**

- As far as I can see, you have only five ensembles (seeds) for each setup; I doubt that (i) the t-test is a proper way of measuring statistical significance here (why would you believe that the assumptions are satisfies?).
- In Fig. 3, is the disparate benefits effect visible in the Figure? Or do we only know this from the selection based on table 1? I am inclined to the second option but would like to confirm this.

**Relation To Broader Scientific Literature:**

Excellent

**Theoretical Claims:**

No theoretical claims are being made.

---

> ### Author Rebuttal · Authors · 2025-03-27
>
> We thank the reviewer for their very positive assessment of our work. We are pleased that they consider our experiments to be well designed, described and sufficient to support our claim that ensembling can have a negative impact on fairness.
>
> ## Questions
> **Q1**
>
> We acknowledge that the sample size (5 ensembles per task) is relatively small, primarily due to the computational cost of training large, fully independent ensembles (e.g., 5 seeds × 10 models = 50 models, totaling ~3000 GPU hours for our experiments). This way we can assure independence of individual samples. To give more insight into the normality condition, we performed the Shapiro-Wilk test. While we acknowledge that normality is difficult to verify with small samples, the results generally support the use of the t-test. Noteworthy, there is one case rejecting normality at p < 0.05: SPD on CX (age). We will make the respective entry in Tab 1 non-bold and provide the following additional table in the final version of the paper.
>
> **Shapiro-Wilk test p-value**
> |  | Accuracy | SPD | EOD | AOD |
> |-------|-------|-------|-------|-------|
> | FF (a/g) | 0.44  | 0.62  | 0.82  | 0.20  |
> | FF (a/r) | 0.44  | 0.28  | 0.97  | 0.35  |
> | FF (g/a) | 0.76  | 0.96  | 0.85  | 0.55  |
> | FF (g/r) | 0.76  | 0.31  | 0.48  | 0.63  |
> | FF (r/a) | 0.86  | 0.95  | 0.82  | 0.28  |
> | FF (r/g) | 0.86  | 0.28  | 0.30  | 0.73  |
> | UTK (a/g) | 0.30  | 0.21  | 0.07  | 0.15  |
> | UTK (a/r) | 0.30  | 0.88  | 0.20  | 0.90  |
> | UTK (g/a) | 0.64  | 0.40  | 0.31  | 0.57  |
> | UTK (g/r) | 0.64  | 0.17  | 0.76  | 0.21  |
> | UTK (r/a) | 0.22  | 0.32  | 0.06  | 0.82  |
> | UTK (r/g) | 0.22  | 0.40  | 0.61  | 0.58  |
> | CX (a) | 0.47  | **0.01**  | 0.18  | 0.22  |
> | CX (g) | 0.47  | 0.29  | 0.33  | 0.14  |
> | CX (r) | 0.47  | 0.10  | 0.68  | 0.59  |
>
> We also considered increasing the sample size by combining members across seeds, but this would introduce dependencies between ensemble members, which we believe would compromise the statistical validity more than the limited sample size does. That said, we are open to alternative suggestions for robust significance testing if the reviewer has a preferred method in mind.
>
> **Q2**
>
> Yes, we selected according to Tab 1 as the disparate benefits effect is not directly visible in Fig 3. The purpose of Fig 3 is to illustrate how the disparate benefits effect emerges when there are differences in the predictive diversity between groups (top row vs bottom row in Fig 3). These differences are highlighted by the black arrows in the figure, showing greater disparities in predictive diversity precisely in the tasks where we have found disparate benefits as per Tab 1.
>
> For clarity, as explained in the caption of Fig 3, the top and bottom rows show examples of datasets where disparate benefits are and are not present, respectively. In the top row, we observe significant differences in predictive diversity, while in the bottom row, there are no or residual differences. This contrast demonstrates how predictive diversity differences correlate with the presence or absence of the disparate benefits effect. We will revise the caption and the accompanying text to clearly explain the distinction between the effect itself (shown in Tab 1) and its explanation (shown in Fig 3).
>
> ## Remarks
> **Causation between predictive diversity and disparate benefits**
>
> We fully agree that causality is very hard to establish. The real world experiments are observational in nature. Therefore, we conducted the interventional synthetic experiment. Note that we provide an extension of the experiment in Apx F.1., where we change the feature diversity per group (the $\alpha$ parameter). There we see strong correlations between the differences in predictive diversity and the disparate benefits effect (see Tab 19 and Fig 16).
>
> **Definition of "disparate benefits effect"**
>
> Thank you very much for this feedback. We agree with your assessment that the definition of the disparate benefits effect in the abstract is the best one and will adapt the description in the introduction accordingly, providing a more formal definition.
>
> **Effect size**
>
> The disparate benefits effect depends on the unfairness of the base models. For example, the absolute increase shown in Table 1 of SPD in 2.2% for FF (age/gender) corresponds to a 10% relative increase in unfairness. Furthermore, the change in unfairness is closely linked to the increase in accuracy due to ensembling, which is in the range of 1-3% on our considered tasks. Therefore, the absolute effect size of the disparate benefits can not be expected to be substantially larger than the effect of ensembling itself.
>
> **Scatter plot**
>
> Thank you for this suggestion, we will provide such plots in the final version of the paper.
>
> We thank the reviewer again for this thoughtful and constructive feedback. We hope to have properly addressed the questions and concerns.

---

### Official Review · Reviewer_iGve · 2025-03-14

**Overall Recommendation:** 4

**Summary:**

The paper discusses two relations between deep ensembles and fairness notions. The first relation is the effect of the number of members on the overall fairness performance of the system. The second is what the authors call 'disparate benefit', which is the difference in fairness violation between the ensemble and the average fairness violation of the members. This is achieved by conducting several large scale experiments on three datasets. The results of the initial experiments are further analyzed by conducting new experiments investigating the relation between the disparate benefit phenomenon and the predictive diversity and expected calibration error.

**Claims And Evidence:**

The claims made in the paper are supported by experiments. The main body of the paper only provides results for one architecture, however more results are provided in the appendix.

**Essential References Not Discussed:**

The reviewer is not aware of any contradacting research results.

**Experimental Designs Or Analyses:**

The overal experimental design is sound with many variables and several random seeds to account for variance in the results. The analyses are elaborate enough when also accounting for the results provided in the appendix. The reporting of the figures would benefit from
equalizing the ranges on the axes. For example in Figure 3 the reporting would improve if the y-axes were the same for the plots as the difference in length of the arrows between the top and bottom row is requied for this figure. The controlled experiment is a nice touch to validate the authors' suspicion with regard to the role that predictive diversity plays in affecting fairness violation behaviour.

**Methods And Evaluation Criteria:**

As concluded by the authors, the scope of the current experiments is rather limited, as the only use case is on image datasets and only one type of bias mitigation method is applied. The datasets used for the experiments are appropriate for this type of fairness research and the Hardt method is one of the strongest. The paper obscures quite some information with regards to fairness by only considering binary sensitive attributes (numerical and categorical attributes are mapped to be binary). This limits the scope of the analysis significantly and this choice could be communicated at an earlier stage of the paper.

**Other Comments Or Suggestions:**

/

**Other Strengths And Weaknesses:**

This is an overall well-written research paper that focuses on the behaviour of fairness notions in the context of fairness violations. The results are abundantly communicated through the figures and also discussed in the main body of the paper.  The scope of datasets and bias mitigation methods applied limit the validity of the paper, which was also acknowledged by the authors.

**Questions For Authors:**

/

**Relation To Broader Scientific Literature:**

The field of fairness research has in recent years started to investigate ensembles as an ML framework which might have some significant influences on the fairness of a system. The paper itself does not discuss many other fairness papers that focus on ensembles specifically. The only comparison seems to be made to the work of Ko et al. (2023). The main bulk of fairness research cited by the paper are
more general papers that are relevant for most fairness research. Examples of fairness research on ensembles includes the works of Gohar et al. (2023), Kenfack et al. (2021), Mishra and Kumar. (2023), and Tayebi and Garibay (2023).


Usman Gohar, Sumon Biswas, Hridesh Rajan. Towards Understanding Fairness and its Composition in Ensemble Machine Learning (2023). ICSE '23

Patrik Joslin Kenfack, Adil Mehmood Khan, S.M. Ahsan Kazmi, Rasheed Hussain, Alma Oracevic, Asad Masood Khattak. Impact of Model Ensemble On the Fairness of Classifiers in Machine Learning. ICAPAI '21.

Gargi Mishra, Rajeev Kumar. An individual fairness based outlier detection ensemble. (2023). Pattern Recognition Letters

Aida Tayebi, Ozlem Ozmen Garibay. Improving Fairness via Deep Ensemble Framework Using Preprocessing Interventions. (2023). HCII '23

**Theoretical Claims:**

No theoretical claims were made in the paper.

---

> ### Author Rebuttal · Authors · 2025-03-27
>
> We sincerely thank the reviewer for engaging deeply with our work and for providing their thoughtful and positive assessment. We’re pleased that you found the experimental design sound, the analyses sufficiently elaborate, and the results well-supported. Your detailed comments helped us reflect on both the limitations and opportunities for improvement in the framing and positioning of our work. We would like to respond in more detail to two specific comments:
>
> > The paper obscures quite some information with regards to fairness by only considering binary sensitive attributes (numerical and categorical attributes are mapped to be binary). This limits the scope of the analysis significantly and this choice could be communicated at an earlier stage of the paper.
>
> We fully agree that explicitly stating the scope of the fairness setting—particularly the use of binary sensitive attributes and labels—is crucial for readers to assess our findings. We chose to focus on the canonical setting of binary labels and attributes to make our findings easily accessible and interpretable. However, exploring non-binary labels and group attributes (also possibly intersectional) is indeed an important direction for future work.
>
> While we currently state our setting in the opening paragraphs of Sections 3 (formal setup) and 4 (dataset description), we acknowledge that this may come too late for some readers. To make this clearer upfront, we propose updating the final paragraph on the first page to state: *“... each with multiple* **binary** *target variables and protected group attributes.”* to ensure that readers are immediately aware of our setup before engaging with the technical content.
>
> > Relation To Broader Scientific Literature
>
> Thank you for highlighting several important and relevant works on fairness in ensemble learning. While our Related Work section focused primarily on studies most closely related to the disparate benefit phenomenon—such as Ko et al. (2023)—we fully agree that placing our contribution within the broader literature on ensembles and fairness will strengthen the paper.
>
> In the revised version, we will expand the Related Work section to include the studies you suggested (Gohar et al., 2023; Kenfack et al., 2021; Mishra & Kumar, 2023; Tayebi & Garibay, 2023) as well as additional relevant works (e.g., Grgić-Hlača et al., Bhaskaruni et al.). For example:
>
> * Tayebi & Garibay apply pre-processing interventions to ensemble members of a Deep Ensemble (on tabular data), offering a complementary perspective to our work.
> * Gohar et al. investigate multiple research questions around the fairness impact of different ensembling techniques for classical models (e.g. SVMs, logistic regression, naive bayes), extending the discussion beyond deep learning.
>
> We believe these additions will further clarify our paper’s relationship to the broader fairness literature and underscore its specific contributions.
>
>
> Once again, we sincerely thank the reviewer for their valuable feedback. We appreciate your support of our work and are committed to addressing these points to improve the clarity and contextualization of our contribution in the final version. Please don’t hesitate to suggest any additional revisions that might further strengthen the paper.
>
> ---
>
> Grgić-Hlača, N., Zafar, M. B., Gummadi, K. P., & Weller, A. (2017). On fairness, diversity and randomness in algorithmic decision making. arXiv:1706.10208.
>
> Bhaskaruni, D., Hu, H., & Lan, C. (2019, November). Improving prediction fairness via model ensemble. In 2019 IEEE 31st International conference on tools with artificial intelligence (ICTAI) (pp. 1810-1814).

---

### Official Review · Reviewer_zbgH · 2025-03-17

**Overall Recommendation:** 3

**Summary:**

This paper explores the impact of Deep Ensembles on algorithmic fairness. The authors find that Deep Ensembles, while improving overall performance, can unevenly benefit different groups, a phenomenon they call the "disparate benefits effect." The paper suggests that differences in predictive diversity among ensemble members across different groups could potentially explain this effect. The paper also shows that the classical post-processing method (Hardt et al., NIPS 2016) can mitigate the disparate benefits effect by utilizing the better-calibrated predictive distributions of Deep Ensembles.

**Claims And Evidence:**

The paper's central finding is the potential for unfairness in deep ensembles. Empirical results highlight this issue, though the reliance on vision datasets may limit the generalizability of these findings.

**Essential References Not Discussed:**

Mostly well-discussed.

**Experimental Designs Or Analyses:**

While the paper's diverse experimental results offer various insights, it would benefit from the inclusion of different dataset types (e.g., non-image) and a more comprehensive set of fairness algorithms, as discussed above.

**Methods And Evaluation Criteria:**

The paper shows that an existing post-processing method (Hardt et al., NIPS 2016) can mitigate unfairness in deep ensembles, but it does not introduce a novel algorithm for this purpose. Comparing more diverse fairness algorithms would enhance the paper by providing a deeper understanding of different fairness approaches within ensemble learning.

**Other Comments Or Suggestions:**

NA

**Other Strengths And Weaknesses:**

Strengths:
- The paper is well-written, and the overall results provide valuable insights.
- The paper offers an interesting intuition into unfairness in deep ensemble learning.

Weaknesses:
- Although the paper provides diverse experimental results, its reliance on vision datasets may limit the generalizability of the findings.
- Additionally, the paper does not introduce its own fairness algorithm. While Hardt post-processing is a good existing algorithm that can mitigate this issue, it is not specifically designed for ensembling. This also weakens the connection between the intuition in Section 6 and the mitigation strategy.
- Furthermore, a broader comparison with various types of fairness algorithms would be beneficial.

**Questions For Authors:**

Questions are included in the above sections.

**Relation To Broader Scientific Literature:**

This paper focuses on potential unfairness issues in deep ensemble learning, which may offer some insights for applications of ensemble methods in scientific literature.

**Theoretical Claims:**

The paper does not contain theoretical results.

---

> ### Author Rebuttal · Authors · 2025-03-27
>
> We thank the reviewer for their thorough assessment of our work.  We are pleased that it was found to be well written and to offer valuable insights. In response to the questions raised:
>
> > Limited generalizability of findings due to focus on vision datasets
>
> We agree with your observation and acknowledge this limitation in the final paragraph of the paper:  *“The main limitations of our study are that we focus on vision tasks, and hence on ensembles of convolutional DNNs. [...]”*. That said, we aim to partially mitigate this limitation by including a diverse range of vision tasks spanning facial analysis (including under distribution shift) and medical imaging. These are application domains where Deep Neural Networks and Deep Ensembles are commonly deployed, making them highly relevant for studying their fairness implications. Moreover, we note that Deep Ensembles are rarely used in the tabular settings often featured in fairness research, where tree-based models remain dominant (Shwartz-Ziv, Ravid, and Amitai Armon. "Tabular data: Deep learning is not all you need." Information Fusion 81 (2022): 84-90). This context motivated our focus on vision data, where Deep Ensembles are widely considered.
>
> > The paper does not introduce its own fairness algorithm
>
> We agree that our paper does not propose a new fairness algorithm—this was not our objective. The main contribution of our paper is to show the existence of the disparate benefits effect of Deep Ensembles (Section 5) and to investigate its underlying causes (Section 6). While we do not claim to introduce a novel fairness method, we believe that presenting this new challenge without discussing potential remedies would leave the analysis incomplete. Therefore, in Section 7, we explore ways to mitigate the negative consequences of disparate benefits. We focus on post-processing methods that can be applied on the already trained individual ensemble members for practicality. Specifically: (1) we show that weighting members non-uniformly does not improve the performance-fairness trade-off whereas (2) Hardt Post-processing (HPP) is very effective in improving fairness while maintaining the utility of Deep Ensembles; and (3) we provide insights into why this is the case (improved calibration / threshold sensitivity).
>
> > [Hardt Post-processing] is not specifically designed for ensembling. This also weakens the connection between the intuition in Section 6 and the mitigation strategy.
>
> Indeed, HPP is not specifically designed to operate on ensembles.  However, its model-agnostic nature allows it to operate on any predictive distribution, making it broadly applicable. Since Deep Ensembles produce predictions by averaging the output probabilities of individual models (Eq. 1), HPP can be applied in exactly the same way as with individual ensemble members.
>
> Regarding “the intuition in Section 6”, we note that predictive diversity is the reason that ensembles predict differently than individual models - if there were zero predictive diversity, all models would agree and the ensemble would predict exactly the same as every ensemble member.  The averaging of diverse, and sometimes conflicting, predictive distributions leads to improved calibration (as shown in Figure 5). We show that HPP can effectively leverage this property: Deep Ensembles are more sensitive to thresholding decisions than individual models, and HPP, which relies on optimizing group-specific thresholds under fairness constraints, is particularly well-suited to take advantage of this threshold sensitivity. Thus, the improved calibration and diversity in ensembles directly support the effectiveness of HPP as a mitigation strategy.
>
> > Comparison with various types of fairness algorithms
>
> We agree that this is an important direction for future work. We explicitly acknowledge this in the final sentence of the paper: *“Furthermore, we intend to investigate the disparate benefits effect for Deep Ensembles where pre- or in-processing fairness methods have been applied to individual ensemble members.”* Given the wide variety of fairness interventions available, we believe this direction warrants a dedicated study. Expanding the scope further within the 8-page limit would have come at the expense of depth in our current analysis.
>
> We thank the reviewer again for their valuable feedback and thoughtful remarks. We hope that our rebuttal provides a more complete perspective on the contributions and scope of our study, and kindly ask the reviewer to consider a reassessment of their evaluation in light of these points.

---

### Decision · Program_Chairs · 2025-05-01

**Decision:**

Accept (poster)

**Comment:**

The paper presents an empirical study about the fairness impacts of deep ensembles. Experimental results show that deep ensembles unevenly helps boost performance of groups, which furthers the parity in prediction accuracy between groups. The authors explain this may be linked to predictive diversity for each group. The paper received mixed ratings (1,2,3,4). While the reviewers appreciated the empirical findings and potential underlying causes, they also had concerns about the validity of the claim and questioned whether the provided evidence can be tightly connected to the main argument. While the concerns exist, the paper's findings can provide a new insight about the property of deep ensembles.